# Mechanical force regulates the inhibitory function of PD-1

Hui Chen [1,2,9 ✉], Yong Zhang[1,2,9], Lei Cui[1,2,9], Juan Fan[3,4], Huaying Zhu [3,5], Songfang Wu[1], Hang Zhou[1,2], Yanruo Zhang[1,2], Guangtao Song[1,2], Ning Jiang[6], Mingzhao Zhu [1,2], Changjie Lou [7 ✉], Wei Chen [3,8 ✉] & Jizhong Lou [1,2 ✉]

## Abstract

**The immune checkpoint molecule, programmed cell death 1 (PD-1), critically regulates T-cell activation upon binding PD-L1 or PD-L2, making it a key target in cancer immunotherapy. Although extensively studied, the molecular mechanism of the inhibitory function of PD-1 remains incompletely understood. Using the biomembrane force probe (BFP), we measure catch-slip bond behavior between PD-1 and PD-L1/PD-L2 under force. Steered molecular dynamics (SMD) simulation reveals a force-induced bound state distinct from the force-free state observed in solved complex structures. Disrupting interactions that stabilize either state weakens the catch bond, and diminishes the inhibitory function of PD-1. Interestingly, soluble forms of PD-L1/PD-L2 compete with their surface-bound counterparts and attenuate PD-1-mediated T-cell inhibition, suggesting that soluble PD-1 ligands could potentially serve as anti-PD-1 drugs. Tumor growth studies using a gain of function mutant based on the catch-bond mechanism confirm the anti-cancer activity of soluble PD-L1. Our findings highlight that mechanical force governs the inhibitory function of PD-1 and suggest that PD-1 acts as a mechanical sensor in T-cell suppression. Thus, mechanical regulation should be considered when designing PD-1 blocking therapies.**

**Keywords** PD-1; T Cells; Mechanical Force; Immunotherapy; Immunological Synapse
**Subject Categories** Cancer; Immunology; Signal Transduction

## Introduction

Checkpoint immunotherapies targeting co-inhibitory molecules have revolutionized the treatment of various tumors (Baumeister et al, 2016; He and Xu, 2020). Among these, programmed cell death

1 (PD-1) stands out as a critical target. Therapies blocking PD-1 or its ligands have achieved remarkable success, illuminating new possibilities for cancer treatment (Chamoto et al, 2023). Despite several approved drugs for inhibiting PD-1 or its ligands, hundreds more clinical trials are currently ongoing to explore their full potential (Topalian et al, 2020). However, a significant challenge remains: many cancer patients exhibit limited response to PD-1 blockade, even for cancers previously shown to be treatable with this therapy (Wei et al, 2018). This limitation underscores the need for a deeper understanding of the inhibitory mechanism of PD-1.

The interactions between PD-1 and its ligands have been explored using various biochemical and biophysical approaches (Na et al, 2017; Wang et al, 2003; Zak et al, 2017). Antibodies developed against PD-1 or its ligands aim to block PD-1/ligand interactions and diminish the inhibitory function of PD-1 (Lee et al, 2016; Tan et al, 2016). Some of these antibodies have been approved as cancer treatment drugs. However, PD-1/antibody interactions can lead to divergent outcomes (Suzuki et al, 2023), highlighting the importance of precise mechanisms governing PD-1/ligand interactions and how they correlate with PD-1 function.

PD-1 is inducibly expressed on the T-cell surface post-activation. It consists of an extracellular globular domain, a connecting peptide, a transmembrane domain, and an intrinsically disordered intracellular region. Recent evidences indicate that PD-1 can form dimers through transmembrane domain interactions, which are linked to its inhibitory function (Chamoto et al, 2023; Philips et al, 2024). The intracellular region contains two signaling motifs, ITSM and ITIM, that are essential for PD-1 activity. Upon binding to its natural ligands, PD-L1 or PD-L2, via the extracellular domain (Freeman et al, 2000; Latchman et al, 2001), tyrosine residues within ITSM and ITIM become phosphorylated. This phosphorylation recruits phosphatase SHP-2, which dephosphorylates TCR/CD3, CD28 and other downstream molecules, thereby suppressing T-cell activation (Chemnitz et al, 2004; Hui et al, 2017). Additionally, PD-1 has been shown to inhibit T cell activation by disrupting TCR-CD8 cooperativity (Li et al, 2021).

[1]State Key Laboratory of Epigenetic Regulation and Intervention, CAS Center for Excellence in Biomacromolecules, Institute of Biophysics, Chinese Academy of Sciences, Beijing 100101, China. [2]University of Chinese Academy of Sciences, Beijing, China. [3]Collaborative Innovation Center for Diagnosis and Treatment of Infectious Diseases, the MOE Frontier Science Center for Brain Science & Brain-machine Integration, State Key Laboratory for Modern Optical Instrumentation Key Laboratory for Biomedical Engineering of the Ministry of Education, College of Biomedical Engineering and Instrument Science, Zhejiang University, Hangzhou, Zhejiang, China. [4]School of Basic Medical Sciences, Henan University, Zhengzhou, Henan, China. [5]China Resources Sanjiu Medical & Pharmaceutical Co., Ltd., Shenzhen 518000, China. [6]Department of Bioengineering, University of Pennsylvania, Philadelphia, PA 19104, USA. [7]Department of Gastrointestinal Medical Oncology, Harbin Medical University Cancer Hospital, Harbin 150001 Heilongjiang, China. [8]Zhejiang Laboratory for Systems and Precision Medicine, Zhejiang University Medical Center, Hangzhou, Zhejiang, China. [9]These authors contributed equally: Hui Chen, Yong Zhang, Lei Cui. ✉E-mail: cdchenhui@ibp.ac.cn; 601245@hrbmu.edu.cn; jackweichen@zju.edu.cn; jlou@ibp.ac.cn

Multiple studies confirm that mechanical forces from the intracellular cytoskeleton and intercellular motions influence the formation and context of the immunological synapse (Dustin and Depoil, 2011; Hosseini et al, 2009; Roumier et al, 2001). These forces inevitably affect the function of the multiple receptor/ligand interactions within the immune synapse. It has been demonstrated that TCR functions to distinguish non-self from self-antigens with the aid of mechanical force (Liu et al, 2014; Wu et al, 2019). Studies have also shown that CD28 co-stimulation could augment CD3-generated traction forces during T cell activation (Bashour et al, 2014). Thus, it is reasonable to conclude that PD-1/ligand interactions may also be influenced by the action of force, similar to TCR or CD28. Indeed, studies had confirmed that PD-1 transmits piconewton (pN) forces to its ligands upon surface engagement (Ma et al, 2019), and PD-1 could form catch bond with its ligand in mouse system (Li et al, 2024). This raises two critical questions: how does mechanical force affect PD-1/ligand interactions affected by the mechanical force, and whether this modulation influences the inhibitory function of PD-1?

In the present study, we used an integrated approach to investigate the interactions between PD1 and its natural ligands. We found that PD-1/PD-L1 and PD-1/PD-L2 interactions exhibit catch bond behavior, where bond lifetimes increase in the low force regime. This force dependence likely promotes PD-1 engagement within the immune synapse, aids in T cell suppression. We also found that soluble and immobilized ligands have different effects on the inhibitory function of PD-1. Considering that soluble ligands can bind PD-1 but are unable to provide an external force, the result indicated that force might play a role in triggering the inhibitory function of PD-1. We demonstrated that the soluble ectodomain of PD-L1 can compete with PD-1 ligands expressed on the surface of APCs for binding PD-1, and serve as a potential PD-1 blocking drug with antitumor activity. Conversely, surface-immobilized ligands enable the formation of PD-1 clusters, which protect PD-1 from dephosphorylation and maintain SHP-2 recruitment. This process attenuates T-cell activation by dephosphorylating TCR/CD3 and CD28 in the vicinity of PD-1. Our results suggest that PD-1 may also function as a mechano-sensor. Understanding the mechanical regulation of PD-1 function could provide new avenues for designing PD-1-based immunotherapies for cancer.

## Results

### PD-1 forms catch bonds with PD-L1

The interaction of PD-1 and its ligand in the mouse system have been studied, and catch-slip bond behavior is observed (Li et al, 2024). To investigate whether mechanical force affects the interaction between human PD-1 and its ligands, we use a biomembrane force probe (BFP) to measure the force dependence of bond lifetimes between human PD-1 and human PD-L1 at the single-molecule level (Table EV1). BFP is particularly suitable for studying interactions between surface receptors and ligands (Fan et al, 2022; Liu et al, 2014; Zhao et al, 2022). In our experiments, human PD-1 (hPD1) expressing Jurkat cells or mouse PD-1 (mPD-1) expressing EL4 cells were repeatedly contacted and detached against beads coated with human or mouse PD-L1 (Fig. 1A). We

controlled the site density of PD-L1 on the beads to maintain an adhesion frequency below 25%, ensuring predominantly single-bond interactions between PD-L1 and PD-1 on the cell surface (Fig. EV1A,B).

While previous reports suggest that PD-L1 can bind CD80 on T cells (Butte et al, 2007), recent studies increasingly support PD-L1 primarily binding CD80 in cis on the same cell surface (Chaudhri et al, 2018; Sugiura et al, 2019; Zhao et al, 2019). Our data aligns with these findings. Bond lifetimes were collected from PD-1/PD-L1 dissociation events at preset constant force (Fig. EV1C), revealing that human PD-1 also forms catch bonds with human PD-L1 under low force, similar to the mouse systems (Fig. 1B,C). Notably, differences can be observed between human and mouse PD-1/PD-L1 dissociation: human PD-1/PD-L1 bonds exhibited longer lifetimes, peaking at ~7 pN, compared to mouse PD-1/PD-L1 bonds, which peaked at around 10 pN (Fig. 1B,C). These results suggest that human PD-1/PD-L1 interaction may produce a stronger inhibitory effect compared to mouse (Masubuchi et al, 2025). Collectively, we found that PD-1/PD-L1 interactions are actually regulated by force, and the force-dependent behavior differs between human and mouse.

To elucidate the molecular mechanism of PD-1/PD-L1 catch bonds, we employed steered molecular dynamics (SMD) simulations to model the force-driven dissociation of PD-1 and PD-L1 using both constant-velocity (cv-SMD) and constant-force (cf-SMD) modes. The CV-SMD simulations revealed two distinct binding states (states I and II) during the dissociation process (Fig. 1D–F). In state I, the inter-domain angles between PD-1 and PD-L1 ectodomains are ~87°and the extensions between their C-terminal Cα atoms are around 49 Å, closely resembling the crystal structures (Fig. 1D,G,H). In state II, the inter-domain angles are approximately 157° and the extensions are around 77 Å, indicating a novel extended conformation (Fig. 1D,G,H). These two similar binding states were also observed in constant-force SMD simulations (Figs. 1I,J and EV1D,E). Interestingly, removing force from state II snapshots reverted the conformation back to state I (Fig. EV1F,G), suggesting reversible conformational changes between states depending on force application. The switching between state I and state II under force likely enhances the stability of PD-1/PD-L1 interactions, resulting in catch bonds.

### Force-induced PD-1/PD-L1 conformational changes are related to its inhibitory function

To test our hypotheses and further confirm the molecular mechanism of force-regulated PD-1/PD-L1 interactions, we introduced point mutations to PD-1 and/or PD-L1 based on the simulation results.

First, we found that residue E136 of PD-1 can form salt bridges with residues R113 and R125 of PD-L1 in state I, but not in state II (Figs. 2A–C and EV2A,B). This interaction thus likely stabilizes state I. This was confirmed by BFP experiments showing that mutations of these residues (R113S or R125S in PD-L1, and E136R in PD-1) weakened the catch bond, while swapping mutations (E136R in PD-1 and R113E or R125E in PD-L1) rescued it (Fig. 2D,E). Co-inhibition assays demonstrated that the mutations weakening the catch bond increased IL-2 secretion, suggesting a reduction in the inhibitory function of PD-1 (Fig. 2F).

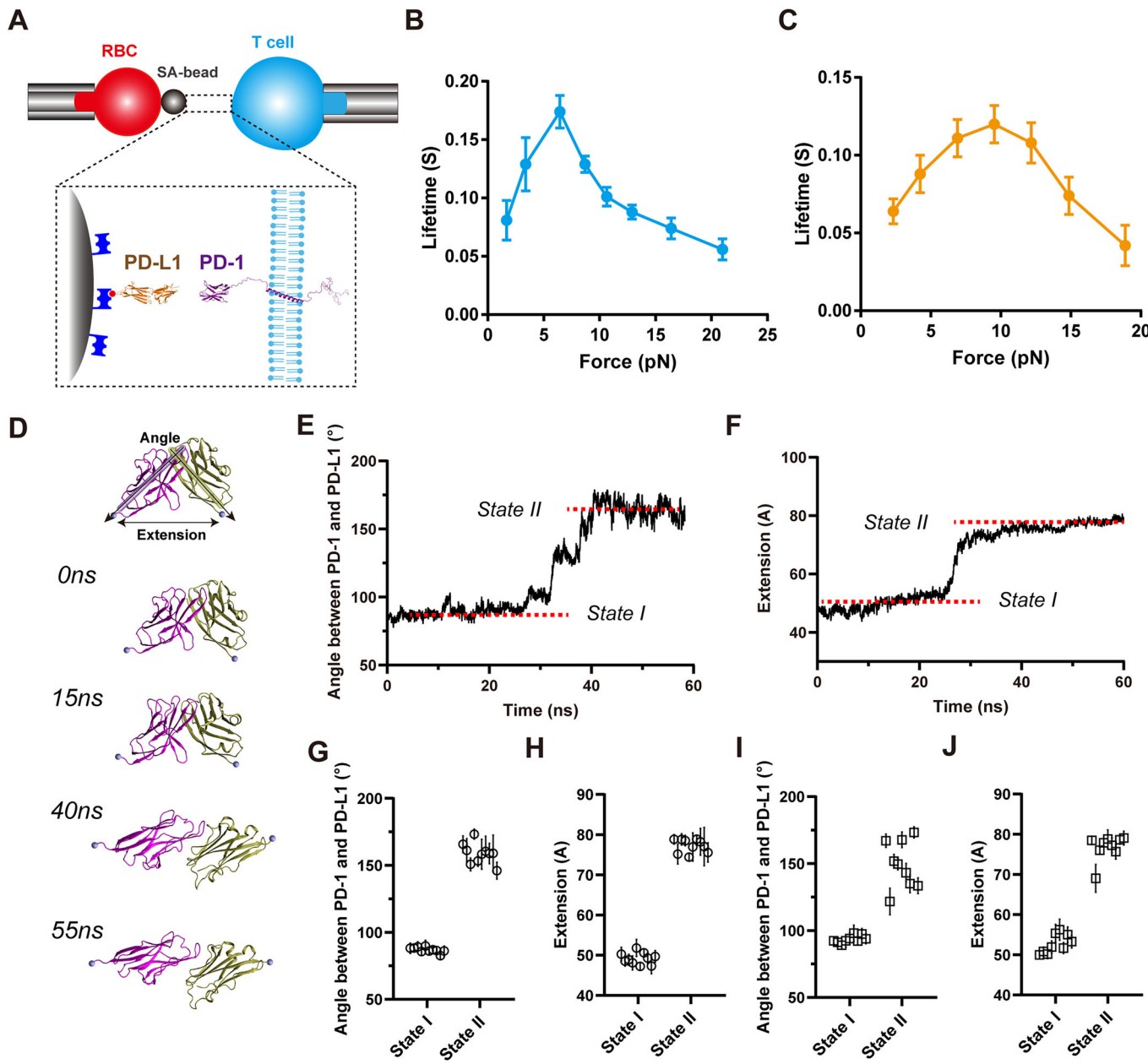

**Figure 1. PD-1 forms catch bonds with PD-L1 under force.**

(A) Schematic of BFP experiments. PD-1 on a living cell (Jurkat cell for human PD-1 or EL4 cell for mouse PD-1) were driven to interact with human or mouse PD-L1 immobilized on a SA bead; (B, C) Mean bond lifetime dependence on mechanical force of human (B) and mouse (C) PD-1/PD-L1 interactions, data are shown as Mean ± SEM; (D) Sequential snapshots at indicated simulation times of cv-SMD simulations showing PD-L1 dissociation from the PD-1; (E) Representative time-course of the inter-domain angle between PD-1 and PD-L1 in cv-SMD simulations, distinguishing two different binding states; (F) Representative time-course of the C-terminus to C-terminus (CT-CT) distances between PD-1 and PD-L1 in cv-SMD simulations, showing two different binding states; (G, H) Statistics of the inter-domain angle (G) and CT-CT distance (H) between PD-1 and PD-L1 for the two different binding states from ten independent cv-SMD simulations, each point represents an average value of angle on either state in a MD trajectory, n = 10, data are shown as Mean ± SD; (I, J) Statistics of the inter-domain angle (I) and CT-CT distance (J) between PD-1 and PD-L1 for the two different binding states from nine independent cf-SMD simulations, each point represents an average value of angle on either state in a MD trajectory, n = 9, data are shown as Mean ± SD. Data information: The force-lifetime curves for the interaction between wild-type PD-1 and PD-L1 (shown in Fig. 1A, B) were reused as a reference to compare with the various mutant groups. Source data are available online for this figure.

Next, we observed that residue L128 of PD-1 forms hydrophobic packing with a hydrophobic patch on PD-L1 composed of residues I54, Y56, V68, and M115 in state I, with a contact area ~55 Å². This packing is enhanced in state II, where the contact area is increased to about 80 Å² (Figs. 2G–I and EV2C,D). Mutations disrupting this hydrophobic interaction (L128A in PD-1 or I54A in PD-L1) also weakened the catch bond, supporting our hypothesis (Fig. 2J). MD Simulations also showed that residue E61 of PD-1 interacts with

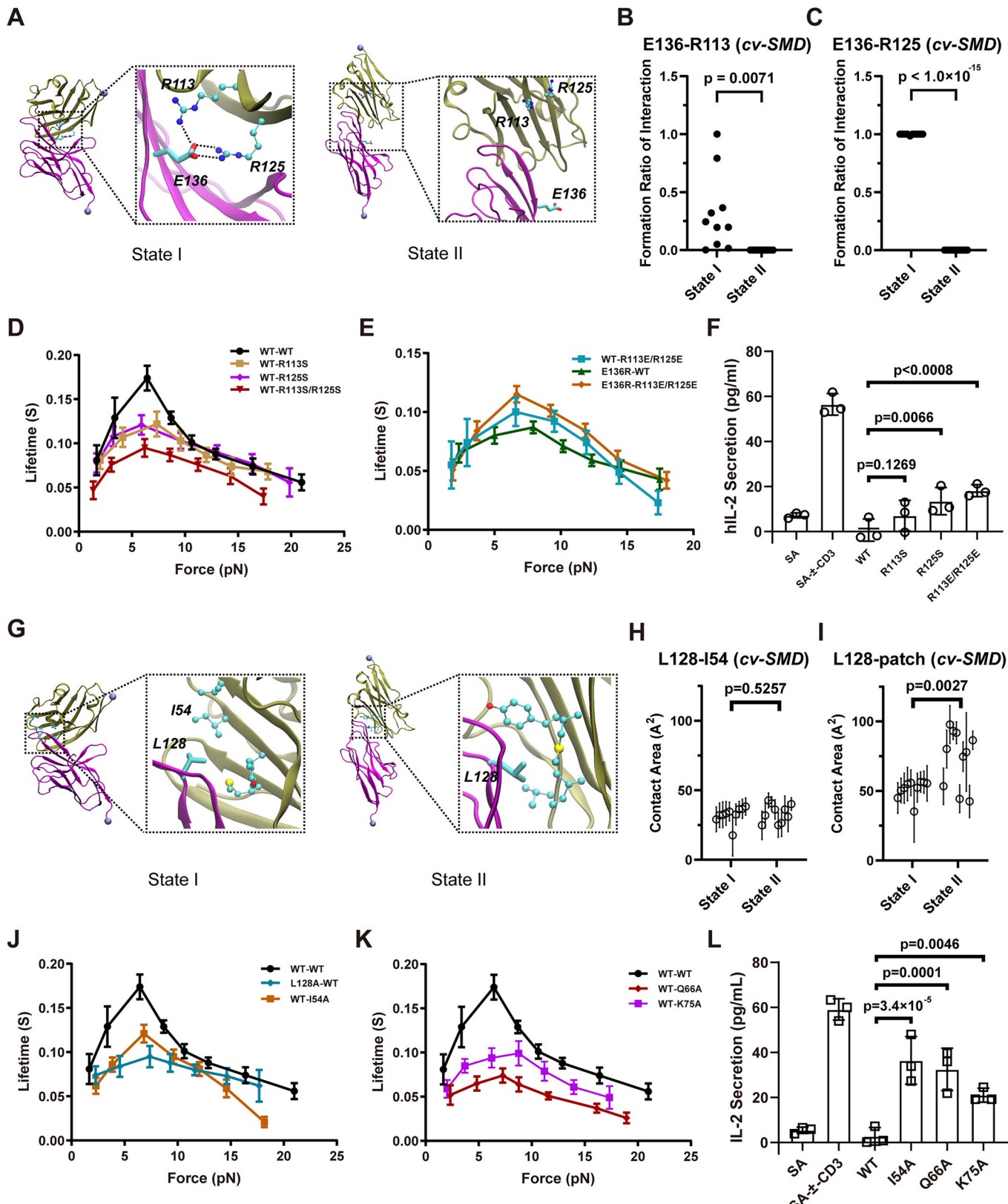

◀ **Figure 2. Molecular mechanisms of PD-1/PD-L1 catch bonds.**

(A) Representative snapshots showing the interaction between E136 (PD-1) and R113/R125 (PD-L1) in two different binding states; (B, C) Probabilities of salt bridge formation between E136 and R113 (B) or R125 (C) in cv-SMD simulations ($n = 10$); (D, E) Mean bond lifetime dependence on force of human wildtype or mutated (E136R) PD-1 interacting with human wildtype or mutated PD-L1 (R113S, R125S, and R113S/R125S in (D), R113E/R125E in (E), data were shown as Mean ± SEM; (F) IL-2 secretion of Jurkat cells expressing human PD-1 stimulated with CD3ε antibody in the presence or absence of widetype or mutated PD-L1, data were shown as Mean ± SD, $n = 3$; (G) Representative snapshots showing the interaction between L128 (PD-1) and the hydrophobic patch of PD-L1 in two different binding states as indicated; (H, I) Contact area between L128 (PD-1) and I54 (H) or hydrophobic patch (I) of PD-L1 in cv-SMD simulations ($n = 10$); (J, K) Mean bond lifetime of wildtype or L128A mutated PD-1 interacting with wildtype or mutated PD-L1, data were shown as Mean ± SEM; (L) PD-L1 mutations impaired PD-1 mediated inhibition of IL-2 secretion, data were shown as Mean ± SD, $n = 3$. Data information: In panel (D, J, K), the force-lifetime curves of wildtype human PD-1 interact with wildtype human PD-L1 were reused from Fig. 1B. T-test is used to produce $p$ value in Fig. 1B, C, H, I. One-way Anova is used to quantify the difference between wildtype and mutated PD-L1 in Fig. 1F, L. all the experiments were technically repeated three times. Source data are available online for this figure.

residue K75 of PD-L1 in state II, but not in state I (Fig. EV2E–G). Additionally, residue Q66 of PD-L1 interacts with different backbone atoms of PD-1 in both states (Fig. EV2H–N). As expected, both Q66A and K75A mutations on PD-L1 also weakened the catch bond (Fig. 2K). Our simulations indicate that instead of interacting directly with PD-1, K75 in PD-L1 indirectly stabilizes state I of the PD-L1/PD-1 complex by forming an intramolecular hydrogen bond network (K75–D73–Q66). The network enables Q66 (PD-L1) to form a direct hydrogen bond with the backbone of A132 (PD-1), an interaction known to be critical for state I stability (Fig. EV2H). Co-inhibition experiments confirmed that weakening PD-1/PD-L1 catch bond via these mutations also led to increased IL-2 secretion, indicating a decrease in the inhibitory function of PD-1 on T cell activation (Fig. 2L).

To rule out potential interference from protein glycosylation, which has been shown to affect PD1/PD-L1 interactions (Li et al, 2018; Okada et al, 2017), we also performed bond lifetime measurements using PD-1/PD-L1 proteins purified from 293F cells. Although the lifetime values were slightly different, all wild-type and mutant PD-L1 proteins exhibited similar catch/slip bond behavior compared to PD-L1 purified from *E. Coli* (Fig. EV2O,P), and the interaction between purified PD-1 and purified PD-L1 also showed catch/slip bond behavior (Fig. EV2Q). This suggests that glycosylation and cellular context have minimal impact on PD-1/PD-L1 interactions, compared to the effects of the introduced mutations.

Thus, our SMD simulations, single-molecule force spectroscopy (SMFS), and cell functional experiments show that mutations to destabilize either PD-1/PD-L1 binding state impair catch bond behavior and simultaneously reduce the inhibitory function of PD-1.

## PD-L2 forms a weaker catch bond with PD-1, correlating with reduced inhibitory effects

PD-L2, another natural ligand of PD-1, also induces the inhibitory function of PD-1. However, previous studies have demonstrated differences in PD-1 binding to PD-L2 compared to PD-L1 (Lazar-Molna et al, 2008; Liang et al, 2006; Zak et al, 2017). By measuring the bond lifetime of PD-1 and PD-L2 under force, we found that the interaction between PD-1 and PD-L2 also exhibits catch bond behavior, although it is distinctly weaker than that of PD-1/PD-L1. This weakness is characterized by a lower optimal force and short bond lifetime (Fig. 3A). Consistently, the inhibitory function of PD-L2 is also weaker than that of PD-L1 (Fig. 3B). A recent study revealed that PD-1 binds PD-L2 more strongly than PD-L1 by SPR

(Masubuchi et al, 2025), which means mechanical force differently altered the stability of PD-1/PD-L interactions.

As expected, cv-SMD and cf-SMD simulations also revealed two binding states for PD-1/PD-L2 interaction under force (Figs. 3C–G and EV3A–D), similar to the PD-1/PD-L1 complex. Structural alignment showed that PD-1/PD-L1 and PD-1/PD-L2 complexes have similar compact conformation in state I, while exhibit conformational divergence in state II (Fig. 3C). In state II, the inter-domain angles between PD-1 and PD-L2 are ~135°, about 22° smaller than those observed in PD-1/PD-L1 complex (Figs. 3D,E and EV3A,B), and the distances between their C-terminal Cα atoms are ~76 Å, similar with those observed in PD-1/PD-L1 complex in both cv-SMD and cf-SMD simulations (Figs. 3F,G and EV3C,D). We propose that the conformational and catch bond differences between state II of PD-1/PD-L1 and PD-1/PD-L2 complexes under force contribute to their functional difference. Specifically, the altered conformation of the PD-1/PD-L2 complex in state II likely results in its weaker catch bond behavior and reduced inhibitory effects.

Our results here for human is different with that obtained for mouse, where PD-1 forms weaker catch bond with PD-L1 than PD-L2 (Li et al, 2024). The discrepancy confirmed the cross-species divergence of PD-1 in human and mouse, as revealed by a recent study (Masubuchi et al, 2025).

## Force enhances PD-1 function and its localization at immune synapses

To further understand the role of mechanical force in PD-1 signaling, we constructed a Tension Gauge Tether (TGT) sensor on glass beads (Fig. EV4A) (Wang and Ha, 2013). Biotinylated human CD3 antibodies were tethered to the highest tension tolerance TGT molecule (High Tension Tolerance, short as High-TT), while PD-L1 was ligated to TGT molecules with different tension tolerance (Low-TT, Medium-TT and High-TT). We found that the activation of the PD-1 signal was highly dependent on mechanical force. The Low-TT TGT molecule tethered to PD-L1 could not activate PD-1 signaling and consequently did not attenuate the IL-2 secretion in Jurkat cells (Figs. 4A and EV4B,D). This result suggests that soluble forms of PD-L1 (S-PDL1) might be ineffective in triggering PD-1 signaling due to their inability to induce mechanical activation of PD-1, a finding supported by previous studies (Wan et al, 2006) and our co-stimulation experiments (Figs. 4B and EV4E).

To further understand the role of mechanical force in PD-1 signaling, we stimulated Jurkat cell with CO-α-hPD1 beads

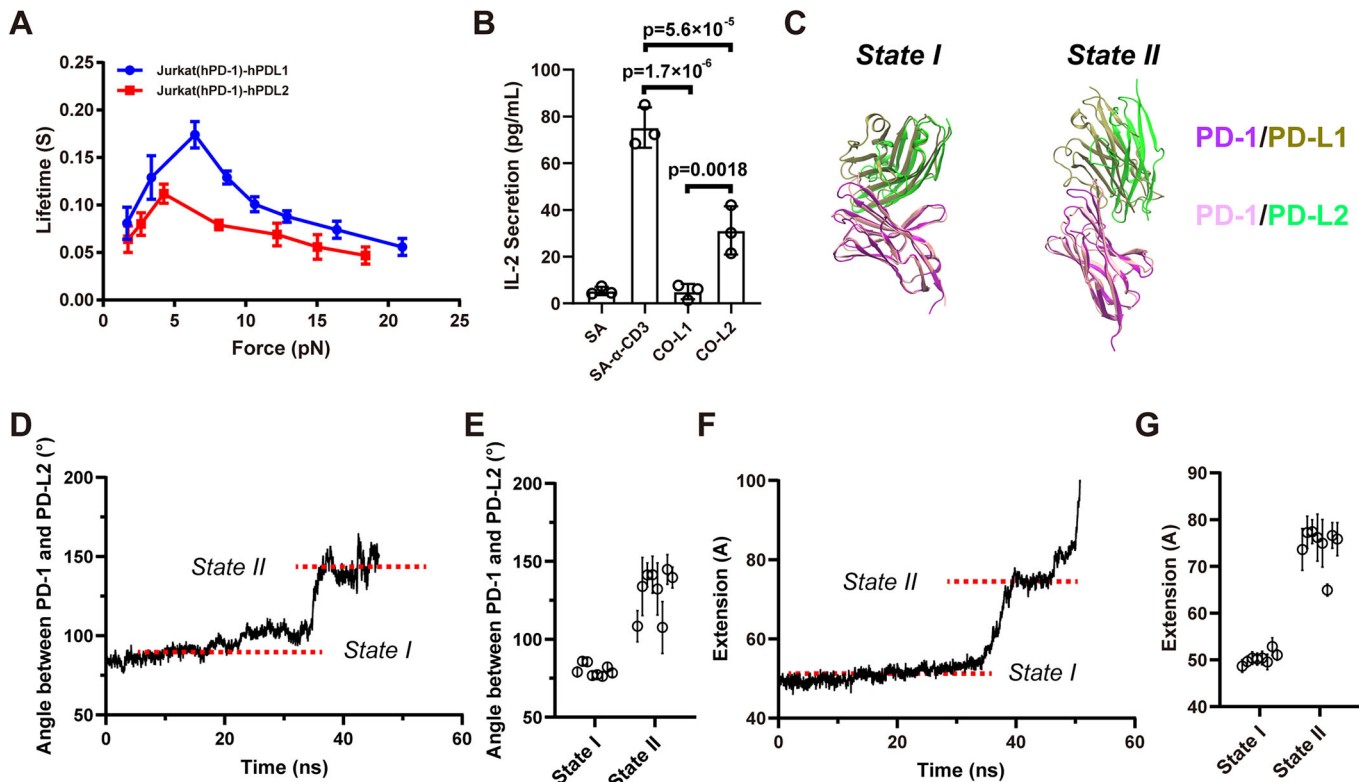

**Figure 3. PD-L2 forms weaker catch bonds with PD-1 than with PD-L1.**

(A) Mean bond lifetime dependence on force for PD-1/PD-L2 interaction, compared to PD-1/PD-L1 interaction, data were shown as Mean ± SEM; (B) IL-2 section inhibition by PD-L1 and PD-L2 in PD-1 expressed Jurkat cells, PD-L1 (CO-L1) or PD-L2 (CO-L2) is coated on SA beads together with CD3ε antibody (SA-α-CD3), data are presented as Mean ± SD, $n = 3$; (C) Structural alignment of PD-1/PD-L1 and PD-1/PD-L2 for the two different binding states obtained in SMD simulations; (D) Representative time-courses of the inter-domain angle between PD-1 and PD-L2 in cv-SMD simulations, distinguishing two different binding states; (E) Statistics of the inter-domain angle between PD-1 and PD-L2 for the two different binding states in cv-SMD simulations ($n = 8$); (F) Representative time-courses of the CT-CT distance between PD-1 and PD-L2 in cv-SMD simulations; (G) Statistics the CT-CT distance (G) between PD-1 and PD-L2 for two different binding states in cv-SMD simulations ($n = 8$). Data information: In panel (A), the force-lifetime curve of wild-type human PD-1 interact with wild-type human PD-L1 was reused from Fig. 1B. T-test is used to produce p value in Fig. 2B. Source data are available online for this figure.

(streptavidin beads simultaneously ligating human PD-1 antibody and CD3 antibody) in the presence of CD28 antibody. The results showed that immobilized PD-1 antibody significantly attenuated IL-2 secretion in Jurkat cells (Fig. 4C), aligning with previous studies where immobilized PD-1 antibody inhibited the proliferation of CD4[+] T cell (Bennett et al, 2003).

Unlike PD-L1, which inhibits IL-2 secretion in a dose-dependent manner, PD-1 antibody inhibits Jurkat cell with a constant inhibitory effect at concentrations comparable to those of a CD3 blocking antibody (Figs. 4D and EV4F). Considering that the PD-1 antibody has a much higher affinity for PD-1 compared with PD-L1, we investigated the inhibitory function of the PD-1 antibody at lower concentrations. We found that the IL-2 secretion decreased proportionally with increasing of PD-1 antibody concentration (Fig. 4E). Additionally, soluble PD-1 antibody could not inhibit IL-2 secretion in Jurkat cell (Fig. EV4G).

To monitor PD-1 phosphorylation, we constructed a phosphorylation reporter molecule by tagging the N-terminal SH2 domain of SHP-2 (SHP-2$_{SH2-SH2}$) with mCherry. PD-1 overexpressing Jurkat cells were stimulated with surface-conjugated PD-L1 and then imaged

with TIRF-SIM (Fig. 4F). We found that wild-type PD-L1 could recruit PD-1 and SHP-2 to the cell-surface contact region, while other mutated PD-L1 cannot (Fig. 4G,H). Our BFP results have shown that these PD-L1 mutations bind PD-1 with a weakened catch bond (Fig. 2E,J,K). We then asked whether strong PD1-PDL1 interaction would contribute to CD45 exclusion, thereby promoting PD-1 phosphorylation (Carbone et al, 2017). So we investigated the distribution of CD45 on PD-L1-stimulated Jurkat cells by TIRF-SIM, and found that the distribution of PD-1 was positively correlated with SHP-2 and negatively correlated with CD45 (Fig. EV4H,I). These results together indicate that mechanical force may regulate PD1-PDL1 interaction and then shape the downstream signaling of PD-1 by excluding CD45.

Moreover, successful T cell inhibition requires the co-localization of activated PD-1 with the CD28 and TCR complex (Figs. 4I and EV5A) (Bennett et al, 2003; Hui et al, 2017). Studies have shown that elongation of PD-1 ectodomain affects the co-localization with TCR (Yokosuka et al, 2012), potentially explaining why larger antibodies exhibit weaker inhibitory function compared to PD-L1 (Fig. 4C,E). Also, a previous study reported that a chimeric receptor, consisting of the mouse CD28 extracellular

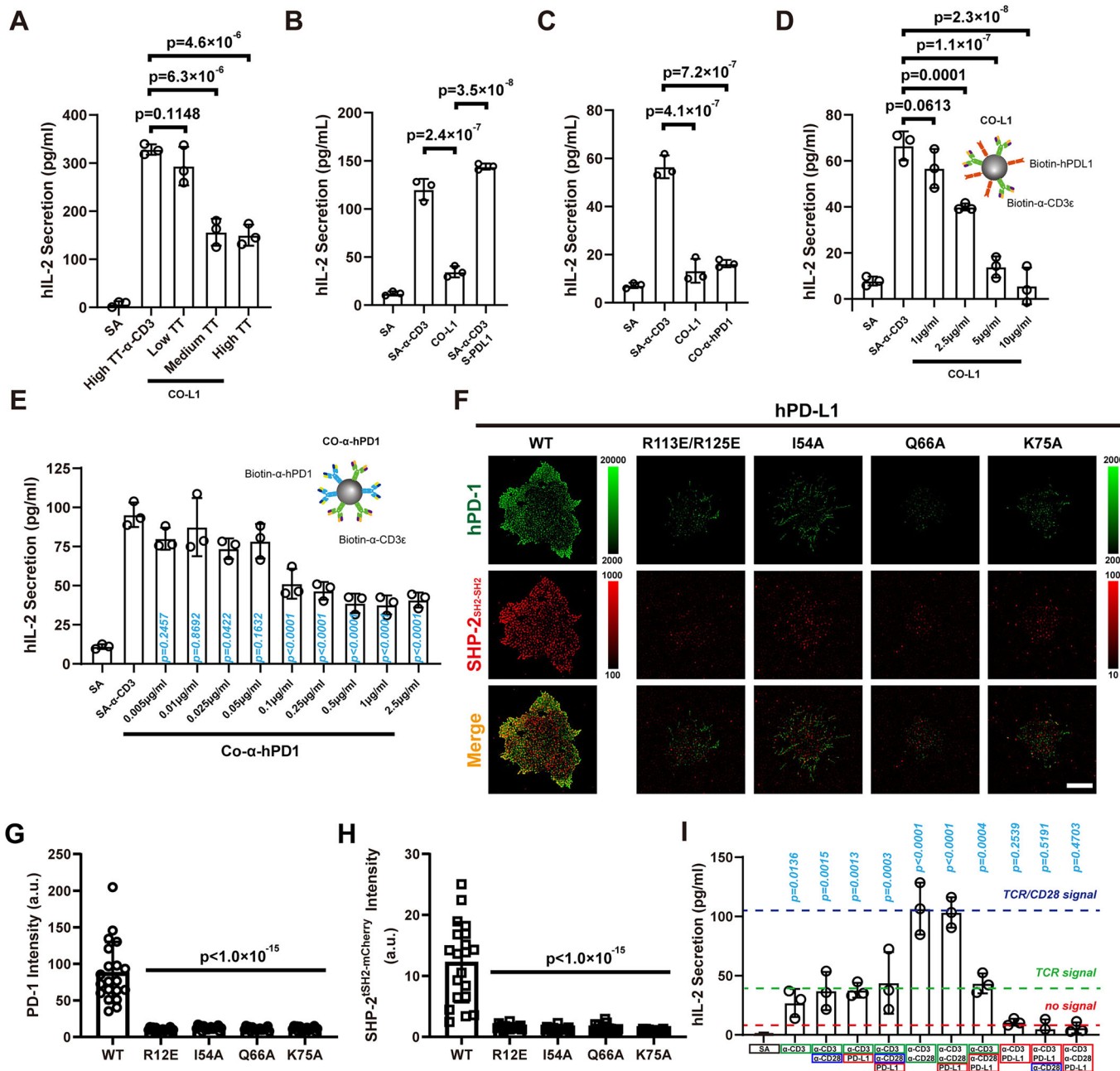

**Figure 4.  Force enhances PD-1 function and its IS localization.**

(A) IL-2 secretion of stimulated Jurkat cells under tension gauge tether (TGT) of indicated tether force, data are shown as mean ± SD, n = 3; (B, C) Jurkat cells were stimulated with beads coated with SA-α-CD3, CO-L1, and SA-α-CD3 in the presence of S-PDL1 (B) or CO-immobilized anti-hPD1 antibody (CO-α-hPD1) (C), data were shown as mean ± SD, n = 3; (D, E), IL-2 secretion of Jurkat cells stimulated with anti-hCD3 beads, CO-PDL1 (D) or CO-α-hPD1 (E) at indicated concentration, data are shown as mean ± SD, n = 3; (F–H) PD-1 phosphorylation in Jurkat cells. Jurkat cells (PD-1-mGFP and SHP2$_{SH2-SH2}$-mCherry overexpressed) were stimulated with the indicated wild-type or mutated PD-L1 for 40 min. Typical images (F) and quantification of PD-1 intensity (G) and SHP2 intensity (H) in the contact area were shown. Scale bar: 10 μm, data were shown as mean ± SD, n = 20 for WT, n = 18 for R12E, I54A, and K75A, n = 17 for Q66A; (I) IL-2 secretion of Jurkat cells stimulated with beads coated with indicated proteins, data were shown as mean ± SD, n = 3. Data information: Experiments in (A–E) were technically repeated three times in the presence of 2.5 μg/ml anti-CD28 and IL-2 secretion was quantified by ELISA, one-way Anova is used to produce p value in Fig. 4A–E, G–I. Source data are available online for this figure.

domain and the human PD-1 cytoplasmic tail, can inhibit CD4[+] T cell in the presence of co-immobilized mouse CD28 antibody (Chemnitz et al, 2004). Altogether, these findings suggest that mechanical force plays a crucial role in PD-1-mediated T-cell inhibition by regulating the activation and localization of PD-1.

## S-PDL1 serves as a potential agent for PD-1/PD-L1 blockade

The aforementioned results show that S-PDL1 cannot trigger PD-1 signaling. Considering that the density of PD-L1 on the surface

might be higher than that in solution, we assessed the function of S-PDL1 in Jurkat cell activation at a higher concentration (200 μg/ml). At this concentration, S-PDL1 slightly inhibited Jurkat cell activation, but the difference was not significant compared to the control group (Fig. EV4E). This partial inhibition could potentially result from PD-L1 dimerization at high concentration (Mahoney et al, 2019).

We then cocultured Jurkat cells with beads coated with both CD3 antibody and PD-L1 (CO-L1) in the presence of S-PDL1 at various concentrations, and found that the IL-2 secretion by Jurkat cells increased with increasing S-PDL1 concentration. Notably, S-PDL1 at 100 μg/ml was sufficient to block the function of immobilized hPD-L1, resulting in IL-2 secretion levels comparable to those stimulated with CD3 antibody alone (Fig. 5A).

To ensure that the increase in IL-2 secretion was not due to the interactions between PD-L1 and CD80, we detected the surface expression of CD80 on Jurkat cell. The result showed that Jurkat cells do not express CD80, confirming that the increase in IL-2 did not result from PDL1-CD80 interaction (Fig. EV4D). These findings were corroborated by an intact cell assay where Jurkat (hPD1[+]) cells were stimulated with SEE superantigen preloaded Raji (hPD-L1[+]) cells, producing consistent results with the beads-based co-stimulation experiments (Fig. 5B). Additionally, a cell–cell conjugate assay showed that S-PDL1 could attenuate hPD1 aggregation at the immune synapse by blocking the interaction of hPD1 and hPD-L1 (Figs. 5C and EV5B,C).

These results indicate the potential antitumor activity of S-PDL1. To confirm this potential and assess the antitumor efficacy of S-PDL1 in vivo, we established MC38 tumors in C57BL6 mice. Considering that wild-type mouse PD-L1 (mPDL1) has a much lower affinity for mouse PD-1 (mPD-1) in the absence of force loading, we constructed a mouse PD-L1 mutant (mPDL1-CR, a C113R mutation in mouse PD-L1) with a higher binding affinity for mPD-1 (Figs. 5D and EV6A). However, mPDL1-CR exhibited no inhibitory functions in either soluble or immobilized form, possibly due to the relatively short lifetime at 10 pN (Figs. 5E and EV6B–G). Surprisingly, the mouse PD-1 antibody also failed to inhibit the activation of CD3[+]CD8[+] T cells, despite its very high binding affinity for mPD-1 (Figs. 5E and EV6D). This inconsistency with the Jurkat cell co-stimulation experiments (Fig. 4C,E) may be attributed to the differences between human and mouse PD-1 systems and/or different antibody binding epitopes.

Importantly, tumor growth curves revealed that soluble mPDL1-CR, but not soluble wild-type mPDL1, could inhibit MC38 tumor growth in C57BL6 mice (Fig. 5F). These findings suggest that soluble PD-L1 has potential applications in PD-1-based antitumor immunotherapy, and that identifying PD-L1 mutants with higher PD-1 affinity could represent a novel strategy for immunotherapy.

## Discussion

Mechanical force plays a vital role in T cell activation (Liu et al, 2014; Sibener et al, 2018; Wu et al, 2019). In this study, we find that human PD-1 can also form catch bonds with PD-L1 and PD-L2 at the single-molecule level, identifying key residues on both human PD-1 and PD-L1 responsible for these force-regulated interactions. Our MD simulations and SMFS studies reveal subtle differences in the molecular mechanism of PD-1/PD-L1 interactions between human and mouse, suggesting species-specific functions of S-PDL1. TGT-based T cell co-stimulation experiments further demonstrate that mechanical loading on the PD-1/PD-L1 interaction is associated with PD-1 function. The phenomenon of mechanor-egulation of the PD-1/ligand interaction axis has been reported in the mouse system, and a series of key residues in the PD-1/PD-L2 interaction have also been revealed. Our results are in good agreement with these findings (Li et al, 2024). Collectively, these findings indicate the lifetime of PD-1/ligand bond under force (such as within the immune synapse) is longer than previously thought. This force-enhanced engagement of PD-1 correlated with its inhibitory functions. However, these results also show the variability of PD-1/PD-L interactions between human and mouse systems (Li et al, 2024; Masubuchi et al, 2025), which may be related to the tissue specificity as well as the dynamics of PD-1 ligand expression in human and mouse (Latchman et al, 2001).

Plenty of studies utilizing surface plasmon resonance (SPR) and other non-mechanical methods have demonstrated that PD-L2 binds to PD-1 with higher affinity than PD-L1 for human (Masubuchi et al, 2025; Tang and Kim, 2019). However, in contrast, studies on T cell function reveal that PD-L2 mediates weaker T cell inhibition compared to PD-L1 (Fig. 3B) (Latchman et al, 2001). We found that under mechanical force, the binding of PD-L2 to PD-1 is actually weaker than that of PD-L1 (Fig. 3A). Given that research has shown PD-1 is subjected to mechanical forces during T cell-target cell interactions (Ma et al, 2019). These findings collectively suggest that the inhibitory function of PD-1 is closely associated with the mechanical stability of its ligands, rather than their binding affinity. Moreover, additional studies have discovered that soluble PD-L2 can bind to PD-1, thereby preventing PD-L1-mediated T cell inhibition (Karunarathne et al, 2016).

Previous studies showed that PD-1 engagement at the immune synapse is PD-L1 dependent and essential for its inhibitory functions (Yokosuka et al, 2012), with PD-1 capable of transmitting pN forces to its ligand (Ma et al, 2019). This suggests a two-step process: PD-1 is initially recruited to the immune synapse and then stretched by ligand binding for downstream inhibitory function. The stretching induces a conformational change in the PD-1/PD-L1 complex, leading to extended bond lifetime (Fig. 6). The prolonged presence of PD-1 allows for its stronger phosphorylation and greater recruitment of SHP-2, which subsequently depho-sphorylates CD3 and CD28 in the vicinity of PD-1 (Fig. 6). Blocking PD-1 with antibodies or soluble PD-L1 prevents PD-1 recruitment by PD-L1 expressed on APCs (Nishi et al, 2023), while mutations in PD-1 or PD-L1 that impair catch bond behavior shorten the dwell time of PD-1 in immune synapse, thereby abrogating its inhibitory function (Fig. 6). However, whether PD-1 phosphorylation depends on the force applied by its ligand or antibody binding remains unclear.

Based on previous studies and our current results, two possible mechanisms may exist by which PD-1 induces T cell inhibition. The first, PD-1 on the cell surface, may be dynamically phosphorylated or dephosphorylated by Lck or CD45, respectively. Binding with soluble PD-1 ligand does not alter this dynamic balance, but immobilized PD-1 ligand could recruit PD-1 to the cell–cell interface, where CD45 is excluded (Carbone et al, 2017). The absence of CD45 in the vicinity makes PD-1 more prone to be phosphorylated by Lck. Moreover, PD-1 phosphorylation is also

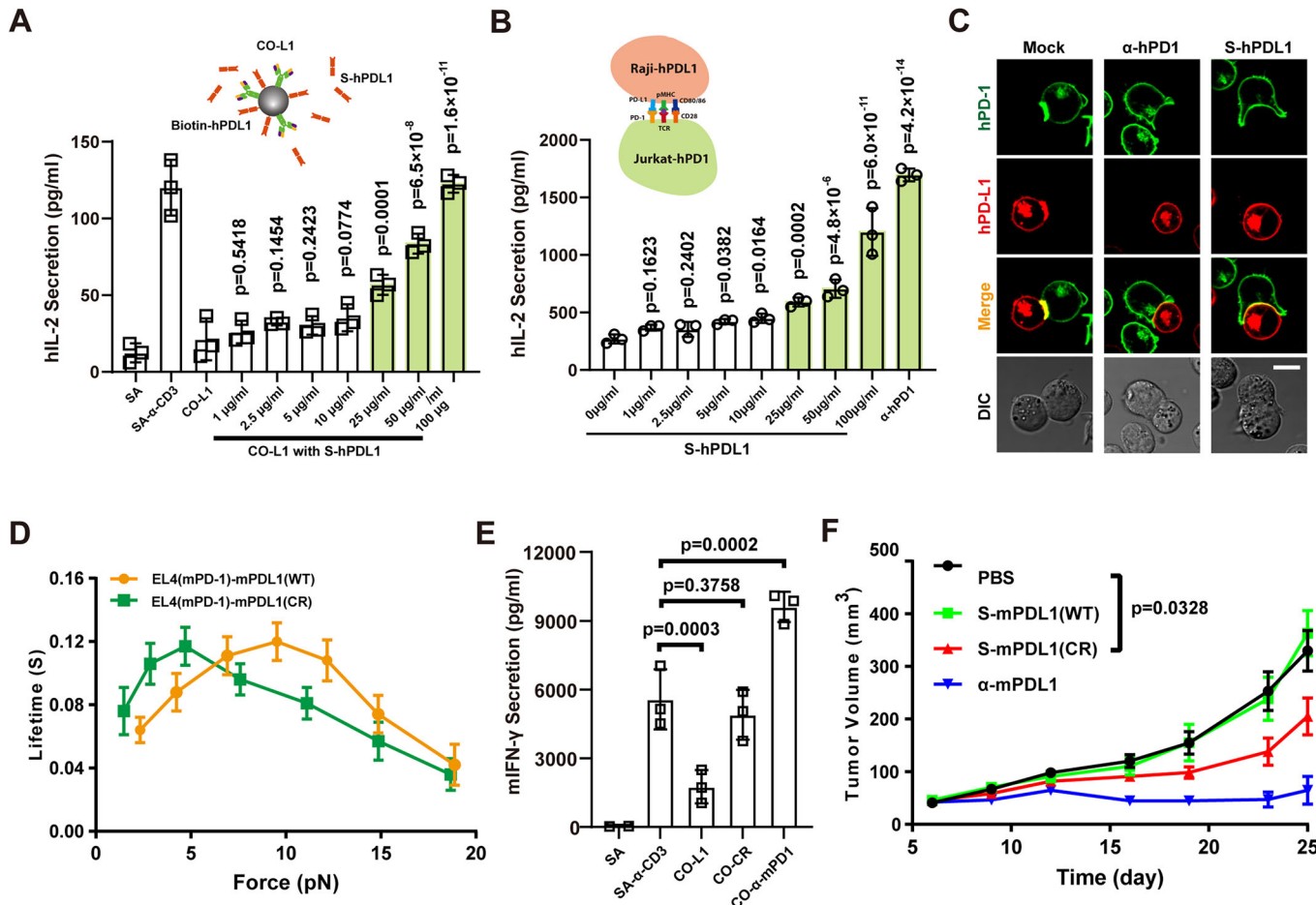

**Figure 5. S-PDL1 abrogates PD-1-mediated inhibitory function, similar to PD-1 antibodies.**

(A) IL-2 secretion of Jurkat cells stimulated with CO-L1 beads in the presence of soluble S-PDL1 at the indicated concentration, data were shown as Mean ± SD, $n = 3$; (B) IL-2 secretion of Jurkat-hPD1 cells in the presence of S-PDL1 at the indicated concentration, 50 μg/ml hPD1 antibody was used as control. (Inset) A paradigm of hPD1 expressed Jurkat cells stimulated with hPD-L1 expressed Raji cells, data were shown as Mean ± SD, $n = 3$; (C) Representative confocal images of Jurkat (hPD1[+])-Raji (hPD-L1[+]) conjugates in the presence of S-hPD-L1 or α-hPD1, Scale bar: 10 μm; (D) Force-dependent lifetime of mouse PD-1 interacting with wildtype and CR-mutated mPDL1, data were shown as Mean ± SEM; (E) Mouse primary CD8[+] T cells were stimulated with beads co-immobilized with mPDL1 or mPD-1 antibody, data were shown as Mean ± SD, $n = 3$; (F) Mouse tumor growth Curves, wildtype C57BL6 mice were inoculated with MC38 cells and then treated with mouse S-mPDL1(WT), S-mPDL1(CR), or mPDL1 antibody, data were shown as Mean ± SEM, $n = 6$. Data information: In panel (D), the force-lifetime curve of wild-type mouse PD-1 interact with wild-type mouse PD-L1 was reused from Fig. 1C. One-way Anova is used to produce *p*-value in Fig. 5A,B,E. Two-way Anova is used to determine the difference between the PBS group and S-mPDL1(CR) group in Fig. 5F. Source data are available online for this figure.

promoted by TCR and CD28 signaling, which facilitates Lck activation (Chen et al, 2023). The second, the intracellular domain (ICD) of PD-1 may adopt a conformation that protects itself from phosphorylation. Binding with soluble PD-1 ligand can hardly affect this inhibited ICD conformation, whereas surface-expressed PD-1 ligand could transmit force to PD-1 and alter this inhibited conformation, which facilitates PD-1 phosphorylation. In both models, the co-localization of PD-1 with TCR and/or CD28 is essential for PD-1-mediated T cell inhibition, and isolated immobilized PD-1 ligands may induce PD-1 phosphorylation but not necessarily T cell inhibition.

Recently, a secreted splicing variant of PD-L1 (sec-PDL1) has been reported to form dimers and inhibit T cell activation (Mahoney et al, 2019). We speculate that sec-PDL1 could simultaneously bind with CD80 on APCs and PD-1 on T cells, promoting PD-1 engagement on the cell–cell contact interface.

Considering that PD-L1 binds CD80 in-cis, soluble sec-PDL1, with its free orientation, can indeed link CD80 and PD-1. Alternatively, sec-PDL1 might induce PD-1 cross-linking via dimerized PD-1, or under the assistance of SHP-2, which has been shown to trigger PD-1 dimerization through its two SH2 domains interacting with two PD-1 molecules (Patsoukis et al, 2020).

The exact mechanism of transmembrane signal transduction of immunoreceptors, such as TCR, remains elusive. Mechanotransduction provides a new angle for in-depth investigation (Zhu et al, 2019). Our current study highlights the importance of mechanical force in PD-1 signaling by inducing a catch bond between PD-1 and its ligands, and soluble PD-L1, where mechanical force application is disabled, could be a potential agent for cancer immunotherapy. However, it remains unclear whether force alone is sufficient to trigger PD-1 phosphorylation. Further studies are required to fully address this question.

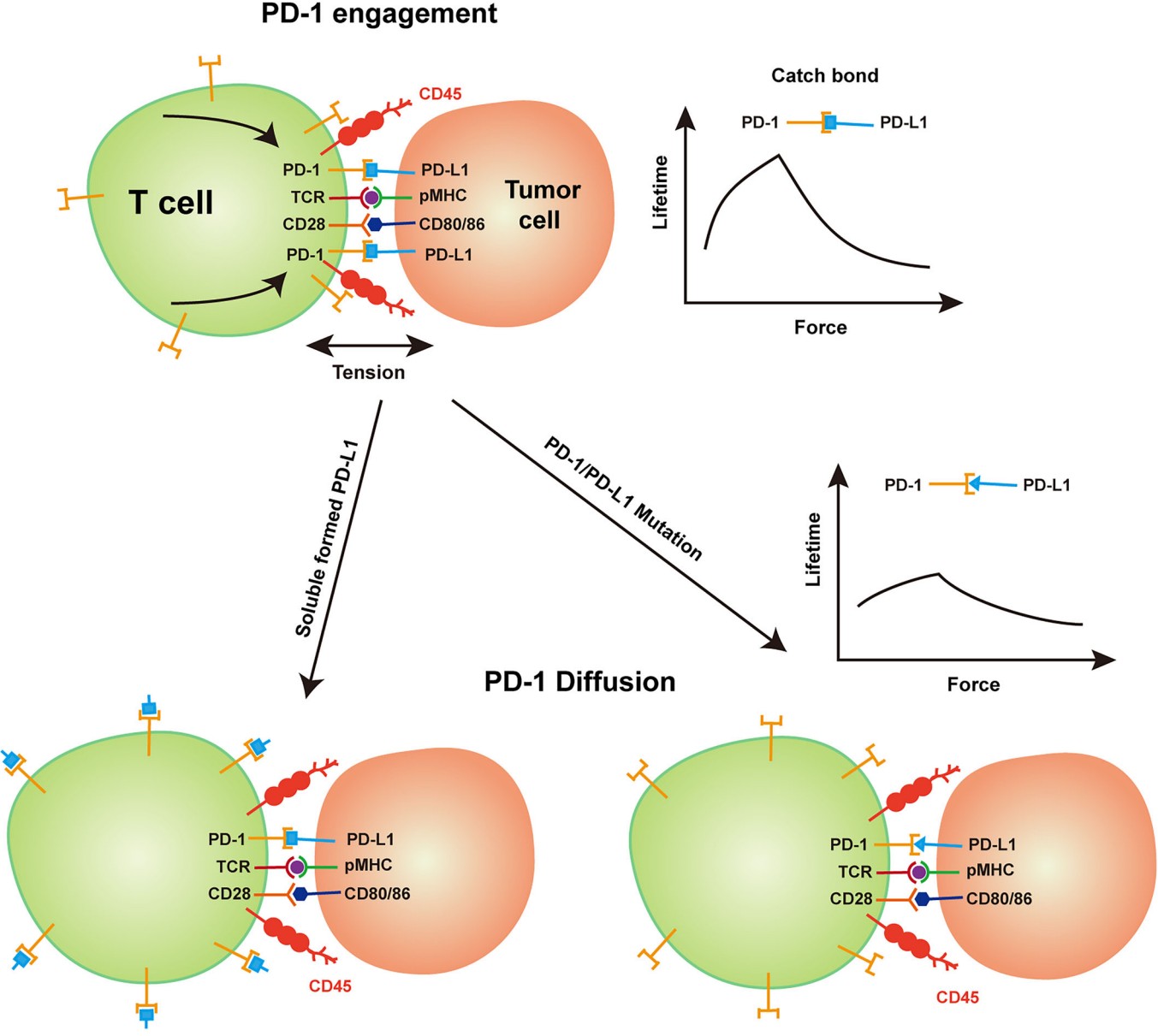

**Figure 6. Model of force-regulated PD-1/ligand interaction at the immune synapse.**

Once the T cell has recognized the target cell via TCR-pMHC interaction, both PD-1 and CD28 bind to their ligand on the target cell side. The resulting tension between the T cell and the target cell acts on the PD1/PDL1 interaction axis, resulting in a conformational change of the PD-1/PD-L1 complex that prolongs the retention time of PD-1 in the vicinity of the TCR and CD28, while simultaneously recruiting the downstream SHP-2 to inhibit its signaling. Mutations in PD-1 or PD-L1 result in a weakening of the PD-1/PD-L1 catch bond, which will affect the residency of PD-1 at the immunological synapse, which in turn will impair its inhibitory function. In the microenvironment, free PD-1 ligands can bind to PD-1 on the surface of the T-cell membrane, preventing the aggregation of PD-1 at the center of the immune synapse, blocking PD-1 signaling, and restoring the function of T cells. Source data are available online for this figure.

In summary, our study reveals that mechanical force modulates PD-1 signaling by stabilizing the PD-1/PD-L1 interaction through a catch bond mechanism under force-applying conditions. Guided by this observation, we engineered a soluble PD-L1 mutant that retains high-affinity binding to PD-1 in the absence of applied force, thereby bypassing force-dependent inhibitory signaling for the wild-type interaction.

This mutant acts as a competitive antagonist, blocking PD-1 binding to membrane-bound ligands and enhancing antitumor immune response. Together, our findings provide a mechanistic framework for the development of soluble checkpoint inhibitors optimized for force-independent blockade, suggesting a potential strategy to overcome resistance to conventional immunotherapies.

# Methods

### Reagents and tools table

| Reagent/resource | Reference or source | Identifier or catalog number |
|---|---|---|
| **Experimental models** | | |
| Jurkat cell (*H. sapiens*) | ProCell | CL-0129 |
| C57BL6 (*M. musculus*) | HFK Bioscience | N/A |
| **Recombinant DNA** | | |
| pET28a-8×His-hPD-L1-AVI | This study | N/A |
| pET28a-8×His-hPD-L1 (R113S)-AVI | This study | N/A |
| pET28a-8×His-hPD-L1 (R113E)-AVI | This study | N/A |
| pET28a-8×His-hPD-L1 (R125S)-AVI | This study | N/A |
| pET28a-8×His-hPD-L1 (R125E)-AVI | This study | N/A |
| pET28a-8×His-hPD-L1 (I54A)-AVI | This study | N/A |
| pET28a-8×His-hPD-L1 (Q66A)-AVI | This study | N/A |
| pET28a-8×His-hPD-L1 (K75A)-AVI | This study | N/A |
| pET28a-8×His-hPD-L2-AVI | This study | N/A |
| pET28a-8×His-mPDL1-AVI | This study | N/A |
| pET28a-8×His-mPDL1(CR)-AVI | This study | N/A |
| pHAGE-8×His-PD-L1-AVI | This study | N/A |
| pHAGE-8×His-PD-L1(R113S)-AVI | This study | N/A |
| pHAGE-8×His-PD-L1(R125S)-AVI | This study | N/A |
| pHAGE-8×His-PD-1-AVI | This study | N/A |
| pHAGE-PD-1-mEGFP | This study | N/A |
| pHAGE-PD-1(L128A)-mEGFP | This study | N/A |
| pHAGE-PD-1(E136R)-mEGFP | This study | N/A |
| pHAGE-SHP-2-mCherry | This study | N/A |
| **Antibodies** | | |
| Biotin-mouse PD-1 antibody | eBioscience | 13-9981-82 |
| Biotin-mouse CD3ε antibody | Abcam | ab25172 |
| APC-mouse CD3ε antibody | eBioscience | 17-0032 |
| FITC-mouse CD8 antibody | Abcam | ab112219 |
| Mouse CD28 antibody | eBioscience | 16-0281 |
| Mouse PD-1 antibody | Abcam | ab63477 |
| Mouse PD-L1 antibody | Abcam | ab80276 |
| PE-human PD-1 antibody | eBioscience | 12-2799 |
| Biotin-human PD-1 antibody | Biolegend | 329934 |
| Biotin-human CD80 antibody | Biolegend | 305204 |
| Biotin-human CD3ε antibody | Abcam | ab191112 |
| Biotin-human CD28 antibody | Biolegend | 302904 |
| APC-human CD45 antibody | Thermo Fisher Scientific | 17-0459-42 |
| **Oligonucleotides and other sequence-based reagents** | | |
| 3′-NH$_2$-CACAGCACGGAGGCA CGACAC-5′ | Sangon, Shanghai | N/A |

| Reagent/resource | Reference or source | Identifier or catalog number |
|---|---|---|
| 5′-Biotin-GTGTCGTGCCTCCGT GCTGTG-3′ | Sangon, Shanghai | N/A |
| 5′-GTG T-Biotin CGTGCCTCCGT GCTGTG-3′ | Sangon, Shanghai | N/A |
| 5′-GTGTCGTGCCTCCGTGCTGT G-Biotin-3′ | Sangon, Shanghai | N/A |
| **Chemicals, Enzymes and other reagents** | | |
| SA-FITC | eBioscience | 11-4317 |
| SA-APC | eBioscience | 17-4317 |
| SA | Sigma-Aldrich | S4762 |
| Quantibrite™ PE beads | BD Biosciences | 340495 |
| APTES | Sigma-Aldrich | 440140 |
| CFSE | eBioscience | 65-0850-84 |
| SEE | Signalway antibody | AP |
| Paraformaldehyde | VETEC | V900894 |
| jetPRIME | Sino Biological | PT-114-15 |
| DMEM high glucose | Thermo Fisher Scientific | 11995065 |
| RMPI 1640 | Thermo Fisher Scientific | A1049101 |
| FBS | Gibco | A5256701 |
| PS | Gibco | 15140122 |
| Tryple express | Gibco | 12604021 |
| DNase I | Beyotime | D7073 |
| RNase A | Sigma-Aldrich | DN25 |
| NP-40 | Abcam | ab142227 |
| NTA-Ni$^{2+}$ Sepharose | Cytia | 17371202 |
| **Software** | | |
| ImageJ | NIH | https://imagej.nih.gov/ij/ |
| Imais | Bitplane | https://imaris.oxinst.com/ |
| GraphPad prism | GraphPad Software | https://www.graphpad.com/ |
| **Other** | | |
| Human IL-2 ELISA kit | eBioscience | BMS221 |
| Mouse IL-2 ELISA kit | eBioscience | BMD601 |
| Mouse IL-10 ELISA kit | eBioscience | BMS614/2 |
| Mouse IFN-gamma ELISA kit | eBioscience | BMS606 |
| Biotin-protein ligase kit | GeneCopoeia | B0101A |
| Streptavidin beads | Spherotech | SVP-30-5 |
| Glass beads | Duke Standards | 9002 |

## Gene cloning and protein expression, purification and labeling

The extracellular domain of PD-1 and PD-L1 genes were obtained from Generay Biotech and subcloned into the pET28a vector with a C-terminal AVI-Tag. All wildtype and mutated proteins for BFP

experiments were expressed in Rosetta (DE3) *Escherichia coli* cells (TsingKe, China), purified and refolded according to published protocols (Cheng et al, 2013). For wildtype and mutated glycosylated PD-L1/PD-1, genes were subcloned into the Phage vector. Recombinant vectors were transduced into 293F cells for protein expression, and the proteins were then purified using Ni-IDA Sefinose resin (BBI Life Science). The biotinylation of AVI-Tag was performed in vitro using the Biotin-protein ligase kit (GeneCopoeia) and purified by gel filtration chromatography.

## Biomembrane force probe (BFP) experiments

Human red blood cells (RBCs) were biotinylated according to published protocols (Liu et al, 2014). Briefly, fresh human RBCs were isolated from the whole blood of healthy volunteers by finger prick. RBCs were covalently linked with biotin-PEG3500-SGA (Jenkem) by incubating at room temperature (RT) for 30 min in coating buffer (0.1 M NaHCO$_3$, 0.1 M Na$_2$CO$_3$, pH 8.5, ~180 mOsm). The biotinylated RBCs were then incubated with different concentrations of nystatin in N2 buffer (265.2 mM KCl, 38.8 mM NaCl, 0.94 mM KH$_2$PO$_4$, 4.74 mM Na$_2$HPO$_4$, 27 mM sucrose; pH 7.2, 588 mOsm) for 1 h at 0 °C, washed twice, and stored in N2 buffer for later BFP experiments.

To prepare protein-coated beads, biotinylated proteins were ligated with SA beads (SVP-30-5, Spherotech) by incubating at RT for 30 min.

The protein-coated SA bead was attached to the apex of biotinylated RBC aspirated by a stationary micropipette. An hPD1-expressing Jurkat cell was aspirated by another micropipette driven by a piezoelectric translator. In each typical measurement cycle, the PD-1 expressing cell was brought into contact with the protein-coated bead with a 20 pN impingement force for 0.2 s, and then retracted and held at a desired force to wait for bond dissociation. Bond lifetime was measured from the time the bond was sustained at the desired force. The cells were used over various forces; the magnitude of the applied force order is manually chosen from three scenarios: a low-high-low sequence, a high-low-high sequence, or a random order. Also, the T cell will be replaced as soon as it begins to change shape. The average binned bond lifetime and standard error of the mean (SEM) was plotted as a function of force (Chen et al, 2010).

For the adhesion frequency assay, the cell was brought into contact with the bead for a different duration time (t$_c$), then retracted till rupture. The adhesion frequency (P$_a$) was calculated from the force-time curves.

## Flow cytometry

For the PD-1 binding assay, 1 × 10$^6$ EL4 cells were washed three times with PBS buffer, and incubated with biotinylated WT or mutated mPDL1 for 30 min at 4 °C. These cells were then further incubated with SA-FITC (eBioscience) for 30 min at 4 °C. After washing with PBS three times, cells were analyzed on a FACSCalibur (Becton Dickinson) (Wang et al, 2003). Biotinylated BSA and unlabeled mPDL1 were used as a negative control, and biotin-anti-mPD-1 antibody (eBioscience) was used as a positive control.

The density of PD-1 and PD-L1 was quantified using Quantibrite™ PE beads (BD Biosciences) as recommended by the manufacturer. Samples were analyzed on a FACSCalibur.

For the quantification of PD-1 expression on Jurkat cells, 1 × 10$^6$ stimulated or unstimulated Jurkat cells were incubated with PE-conjugated hPD1 antibody (J105, eBioscience) for 30 min at 4 °C. Cells were then washed with PBS and analyzed on a FACSCalibur.

To detect CD80 expression of Jurkat cells, 1 × 10$^6$ Jurkat cells were incubated with biotinylated CD80 antibody (2D10, Biolegend) for 30 min at 4 °C. PBS were used as a negative control. After washing three times with PBS, cells were incubated with SA-APC (eBioscience) for 30 min at 4 °C and then analyzed on a FACSCalibur. Raji cells were used as a positive control.

## Beads preparation for Jurkat cell co-stimulation experiments

Indicated concentrations of biotinylated anti-hCD3 (UCHT1, Abcam), hPD-L1, hPD-L2, anti-hCD28 (CD28.2, Biolegend), or anti-hPD1 (EH12.2H7, Biolegend) were incubated with SA beads for 30 min at RT. These beads were washed three times with PBS and can be directly used for co-stimulated experiments. For preparing CO-beads, the SA-α-CD3 beads were further incubated with serial concentrations of biotinylated PD-L1, PD-L2, or anti-PD1 at RT for 30 min. αCD3/αCD28 and αCD3/PD-L1 beads were made by incubating SA-α-CD3 beads with 2.5 μg/ml biotinylated anti-CD28 or PDL1 respectively at RT for 30 min. αCD28/PD-L1 beads were made by incubating SA-α-CD28 beads with 15 μg/ml biotinylated hPD-L1 at RT for 30 min.

## Fabrication of a DNA-based tension gauge tether on glass beads

Glass beads were washed with nanopure water three times, and then incubated with methanol for 10 min. Subsequently, the cleaned beads were suspended with 1% APTES solution (5% H$_2$O, 0.5% HAC, 93.5% Methanol) and incubated for 1 h. Silanized glass beads were washed three times with water and then incubated with 2.5% glutaric dialdehyde in water for 1 h. The glass beads were incubated with 50 μmol/L NH$_2$-labeled ssDNA for 1 h. The beads can be stored at 4 °C for up to 1 month. Biotinylated anti-CD3 (or PD-L1) and biotin-labeled ssDNA were ligated with streptavidin at a 1:1:1 ratio by incubating for 30 min. Anti-CD3-DNA chimera were annealed with the prepared glass beads by incubating for 40 min (Fig. EV4A). These beads can be used in a cell coculture experiment after three times washing. For the PD-1 tension gauge tether assembly, these beads were further incubated with PD-L1-DNA chimera for 40 min. All incubation steps were conducted at room temperature on a rotator to ensure proper mixing and uniform coating.

Notes: The rupture force for TGTs used here (Low-TT, Medium-TT, High-TT) have been reported in a previous study as approximately 12, 23, and 56 pN, respectively. However, it is important to note that factors such as temperature, salt concentration (particularly Mg²⁺), and loading rate significantly influence the actual rupture force of DNA-based tension sensors. Since our experiments are conducted in a cell culture environment, the precise rupture force under our specific conditions remains uncertain. Therefore, the TGT system is employed in this study primarily to enable comparative analysis across different tension thresholds, thereby providing meaningful qualitative distinctions.

## Jurkat cell co-stimulation

For the co-stimulation experiments, $2 \times 10^5$ Jurkat cells were seeded in triplicate wells of a 96-well flat-bottomed plate. Subsequently, $1 \times 10^6$ indicated beads were added to each well, either in the absence or presence of anti-hCD28 (CD28.2, eBioscience). The supernatants from each well were collected at 18–20 h. The concentration of human IL-2 (hIL-2) secretion was quantified by a human IL-2 ELISA kit (Abcam) according to the manufacturer's instructions (Freeman et al, 2000; Hui et al, 2017).

## Jurakt-hPD1-mGFP stimulation with Raji cells

For IL-2 secretion assays, Raji-hPD-L1-mCherry cells were preloaded with 30 ng/ml SEE superantigen (Signalway Antibody) for 1 h at 37 °C. About 40 µl of Raji-hPD-L1-mCherry cells ($5 \times 10^4$ in total) were seeded in a 96-well U-bottom plate and cocultured with 40 µl Jurkat-hPD1-mGFP cells ($2 \times 10^5$ in total). Then 20 µl of either PBS control or various concentrations of soluble hPD-L1 (s-PDL1) were added into each well. The supernatants were collected at 18–20 h and quantified by ELISA as described above.

For the Jurkat-Raji conjugates imaging assay, Jurkat-hPD1-mGFP cells and SEE preloaded Raji-hPD-L1-mCherry cells were plated in an Optical bottom 96-well plate. Soluble hPD-L1 and human PD-1 antibody were added to the solution, corresponding volume of PBS were used as control. Cell conjugates were observed with an Olympus FV1000 confocal microscope, and the images were processed using Olympus Fluoview.

## Mouse primary CD8+ T cell co-stimulation

Single cell suspensions were obtained from the lymph nodes of C57BL6 mice according to published protocol (Kruisbeek, 2001). The cells were stained with anti-mCD3-APC (17A2, eBioscience) and anti-mCD8-FITC (YTS 169AG 101HL, Abcam) at 4 °C for 30 min. CD3+CD8+ T cells were then separated by Flow Cytometry (BD FACS Aria IIIu). Beads for mouse primary CD8+ T cell co-stimulation were prepared as described above.

For the co-stimulation experiments, $2 \times 10^5$ CD3+CD8+ T cells and $1 \times 10^6$ indicated beads were incubated at 37 °C, 5% $CO_2$ for 3 days in the absence or presence of anti-mCD28(37.51, eBioscience). Supernatants were then collected for the quantification of IFN-γ secretion using an ELISA kit (eBioscience). To detect PD-1 expression, the stimulated cells were first incubated with APC-mPD-1, followed by PBS washing, and then quantified by flow cytometry (Becton Dickinson).

For cell proliferation assays, cells were labeled with 1 µM CFSE (eBioscience) for 10 min at RT in the dark. The labeled cells were further cocultured with the indicated beads for 3 days, and subsequently analyzed by flow cytometry.

## TIRF-SIM

Biotinylated PD-L1 was attached to the glass surface via streptavidin. Jurkat cells expressing PD1-mGFP and SHP2^SH2-SH2-mCherry were added to the surface and incubated at 37 °C for 40 min. After incubation, cells were fixed with paraformaldehyde (PFA), and stained with APC-labeled CD45 antibody. Imaging was performed using a Multi-SIM microscopy equipped with a 100× oil immersion objective (Qiao et al, 2021). Image processing and data analyses were carried out using ImageJ.

## Tumor growth and treatment

About $2 \times 10^5$ MC38 tumor cells were injected subcutaneously into the flank of C57BL/6 mice. 50 µl (1 mg/ml) PD-L1, PD-L1-CR, αPD-L1 mAb, or PBS were injected intratumorally at days 6, 9, 12, and 16 post-tumor inoculation. Tumor volumes were measured using a caliper and calculated according to the formula: length × width × height/2 as previously described (Liu et al, 2016). All mouse maintenance and procedures were approved by the Biomedical Research Ethics Committee of the Institute of Biophysics, Chinese Academy of Sciences, and were performed according to the relevant ethical regulations regarding animal research.

## Molecular dynamics (MD) simulation

The complex structures of the human PD1 Ig-like V-type (IgV) domain complexed with PD-L1 IgV domain, and complexed with PD-L2 IgV domain, were built using the PDB structures (PDB code: 4ZQK(Zak et al, 2015) & 6UMT) as initial models, respectively. The missing residues of PD-1 (Asp85-Asp92) and PD-L2 (D65-S67) were built using the Modloop server (Fiser and Sali, 2003; Tang and Kim, 2019). The protein complexes were solvated in a rectangular box of TIP3P water molecules. Na+ and Cl− ions were added to neutralize the whole system (~0.15 M). Both systems were processed using the VMD program and the CHARM36m force field for proteins (Huang et al, 2017; Humphrey et al, 1996).

The resulting systems were first pre-equilibrated to relax the added missing region of protein, counter ions and water molecules. Subsequently, the production simulations lasted ~100 ns with a 2-fs time step. Four independent repeating simulations were performed. During these simulations, the temperature of the systems was maintained at 310 K with Langevin dynamics, and the pressure was controlled at 1 atm with the Nosé-Hoover Langevin piston method. Particle Mesh Ewald summation was used for electrostatic calculation, and a 12 Å cutoff was used for short-range non-bounded interactions. The simulation trajectories were recorded every 20 ps.

Representative snapshots of each production run for the PD-1/PD-L1 and PD-1/PD-L2 complex systems were extracted as initial conformations for steered molecular dynamics (SMD) simulations. Before the forces were applied, these snapshots were first simulated under temperature and pressure control with a 1 fs time-step for 1 ns and underwent free dynamics simulations for another 1 ns for relaxation. The final configurations were used for the following SMD simulations.

Both constant-velocity (cv-SMD) and constant-force SMD (cf-SMD) simulations were performed to investigate the dissociation process of PD-1 and PD-L1/L2. In each CV-SMD simulation, the C-terminal Cα atom of the PD-1 IgV domain was constrained at its initial position with a spring of spring constant ~1400 pN/nm, and the C-terminal Cα atom of the PD-L1/L2 IgV domain was pulled with a dummy spring of spring constant ~70 pN/nm, which moved at a speed ~0.1 nm/ns. In cf-SMD simulations, the applied force was held at 100 or 50 pN until the complex dissociated. For each system, multiple SMD trajectories were obtained for statistical

analysis. The Energy minimizations and MD simulations in this study were performed with NAMD (Phillips et al, 2005) under periodic boundary conditions.

The inter-domain angle between PD-1 and PD-L1/L2 was used to depict their relative orientation, which was defined as the angle between the vector on PD-1 residues (a vector connecting centroids of S57-S62, L100-R104, L128-A132 and T45-N49, S71-Q75, V111-S118 backbone atoms) and that on PD-L1/L2 (a vector connecting centroids of F42-L53, Q91-G95, Y118-D122 and E31-N35, S80-R84, I101-A109 backbone atoms for PD-L1; of D45-I55, L80-G83, Y106-A109 and E33-N37, E63-S67, P90-G98 backbone atoms for PD-L2). The formation of salt bridge interaction was defined as a distance smaller than 3.5 Å between the heavy atoms of the donor and the acceptor residue.

## Statistical analysis

All statistical analyses were performed using GraphPad Prism. Unless otherwise stated, data were presented as means ± SD or scatter plots. Calculated $P$ values were reported. Differences between the means of the experimental groups were calculated using Student's $t$-test. Comparison between tumor sizes was done using two-way ANOVA.

# Data availability

No large primary datasets have been generated and deposited for this study.

The source data of this paper are collected in the following database record: biostudies:S-SCDT-10_1038-S44319-026-00715-6.

# Peer review information

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

## Acknowledgements

We would like to thank Profs. Shengdian Wang and Yan Qin, and members of their group, for assistance with mouse experiments. The computational resources in this study were provided by the Harbin Supercomputer Center and HPC-Service Station at the Center for Biological Imaging of the Institute of Biophysics. We thank J. Jia and S. Meng (Core Facility, Institute of Biophysics, CAS) for technical support in the flow cytometry analysis. This work was supported by grants from the National Natural Science Foundation of China (T2394512 and 32200549 to HC, 32090044 and 11672317 to JL, T2394511 to WC, 12172371 to Yong Zhang and 32301035 to JF), the Strategic Priority

Research Program of the Chinese Academy of Sciences (XDB37020102 to JL), Beijing Medical Award Foundation (YZTZ-2022-0080-0015 to CL), and Henan Provincial Science and Technology Research Project (242102310348 to JF).

## Author contributions

**Hui Chen**: Resources; Data curation; Formal analysis; Supervision; Funding acquisition; Validation; Visualization; Methodology; Writing—original draft; Writing—review and editing. **Yong Zhang**: Data curation; Software; Funding acquisition; Methodology; Writing—original draft; Writing—review and editing. **Lei Cui**: Resources; Data curation; Software; Methodology; Writing—original draft. **Juan Fan**: Data curation; Funding acquisition; Validation; Methodology. **Huaying Zhu**: Data curation; Methodology. **Songfang Wu**: Resources; Validation. **Hang Zhou**: Resources; Data curation; Formal analysis; Investigation; Methodology. **Yanruo Zhang**: Resources; Validation. **Guangtao Song**: Resources; Project administration. **Ning Jiang**: Writing—review and editing. **Mingzhao Zhu**: Methodology; Project administration. **Changjie Lou**: Resources; Supervision; Funding acquisition; Writing—original draft; Project administration; Writing—review and editing. **Wei Chen**: Resources; Supervision; Funding acquisition; Writing—original draft; Project administration; Writing—review and editing. **Jizhong Lou**: Conceptualization; Resources; Formal analysis; Supervision; Funding acquisition; Validation; Investigation; Methodology; Writing—original draft; Project administration; Writing—review and editing.

Source data underlying figure panels in this paper may have individual authorship assigned. Where available, figure panel/source data authorship is listed in the following database record: biostudies:S-SCDT-10_1038-S44319-026-00715-6.

## Disclosure and competing interests statement

The authors declare no competing interests.

# Expanded View Figures

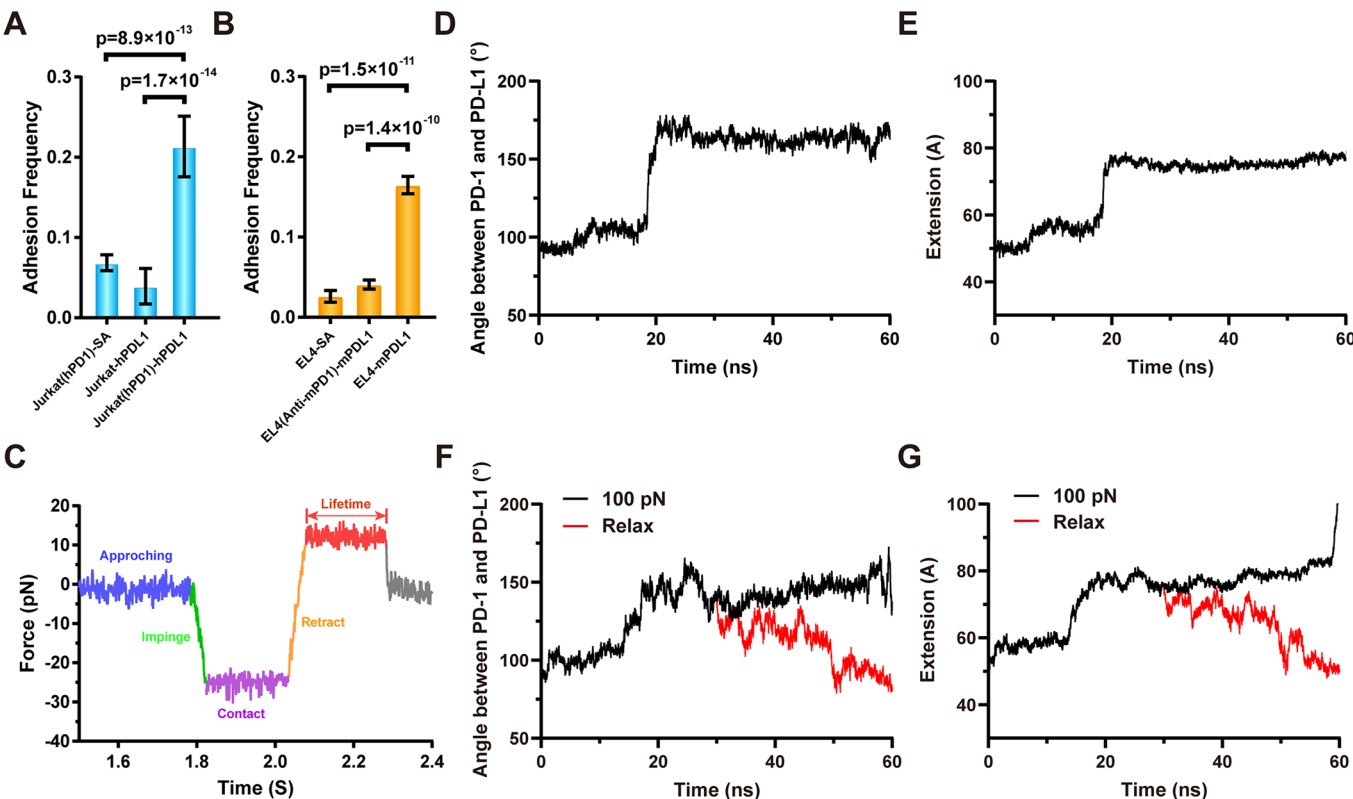

**Figure EV1. Characterization of PD1/ligand interactions.**

(A, B) The adhesion frequency of human (A) and mouse (B) PD-1/PD-L1 interactions, compared to the indicated control, data were shown as mean ± SD, $n = 10$ (A), $n = 9$ (B); (C) Typical force-clamp curve from BFP experiments, different phases were shown in differed colors and indicated; (D, E) Representative time-course of the inter-domain angle (D) and CT-CT distance (E) between PD-1 and PD-L2 in cf-SMD simulations; (F, G). Representative time-course of the inter-domain angle (F) and CT-CT distance (G) between PD-1 and PD-L2 in cf-SMD simulations (black) and relax simulations (red). Data information: one-way ANOVA is used to produce *p* value in Fig. EV1A, B.

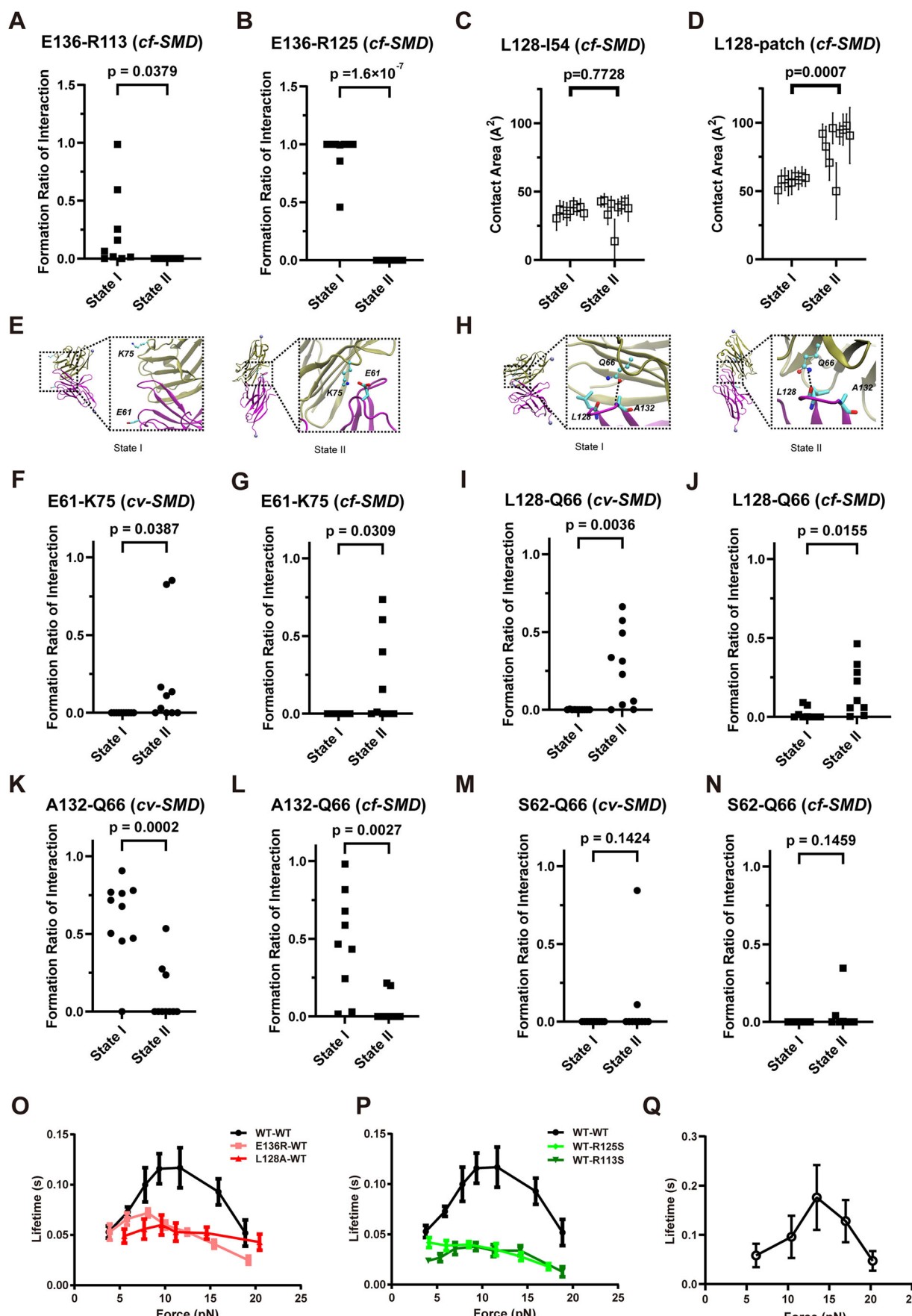

**Figure EV2. Key residue pairs identified from MD simulations.**

(A, B) Probabilities of the formation of E136/R113 (A) and E136/R125 (B) from PD-1 and PD-L1 in cf-SMD simulations ($n = 9$); (C, D) Analysis of contact area between L128 (PD-1) and I54 (C) or hydrophobic patch (D) of PD-L1 in cf-SMD simulations ($n = 9$); (E) Representative snapshots showing the interaction between E61 (PD-1) and K75 (PD-L1) in the two different binding states; (F, G) Probabilities of salt bridge formation between E61 (PD-1) and K75 (PD-L1) in cv-SMD simulations (F), $n = 10$ and cf-SMD simulations (G), $n = 9$; (H) Representative snapshots depicting interaction between L128/A132 backbone (PD1) and Q66 sidechain (PD-L1) in the two different binding states; (I, J) Probabilities of hydrogen bond formation between L128 backbone and Q66 sidechain in cv-SMD simulations (I), $n = 10$ and cf-SMD simulations (J), $n = 9$; (K, L) Probabilities of hydrogen bond formation between A132 backbone and Q66 sidechain in cv-SMD simulations (K), $n = 10$ and cf-SMD simulations (L), $n = 9$; (M, N). Probabilities of salt bridge formation of S62/Q66 in cv-SMD simulations (M), $n = 10$ and cf-SMD simulations (N), $n = 9$; (O) Mean bond lifetime dependence on force for WT or mutated (as indicated) human PD-1 interacting with WT human PD-L1 (purified from 293F), data were shown as mean ± SEM; (P) Mean bond lifetime dependence on force for WT PD-1 interacting with WT or mutated (as indicated) PD-L1 (purified from 293F), data were shown as mean ± SEM. (Q) Mean bond lifetime dependence on force for WT PD-1 interacting with WT PD-L1 (both PD-1 and PD-L1 were purified from 293F), data were shown as mean ± SEM.

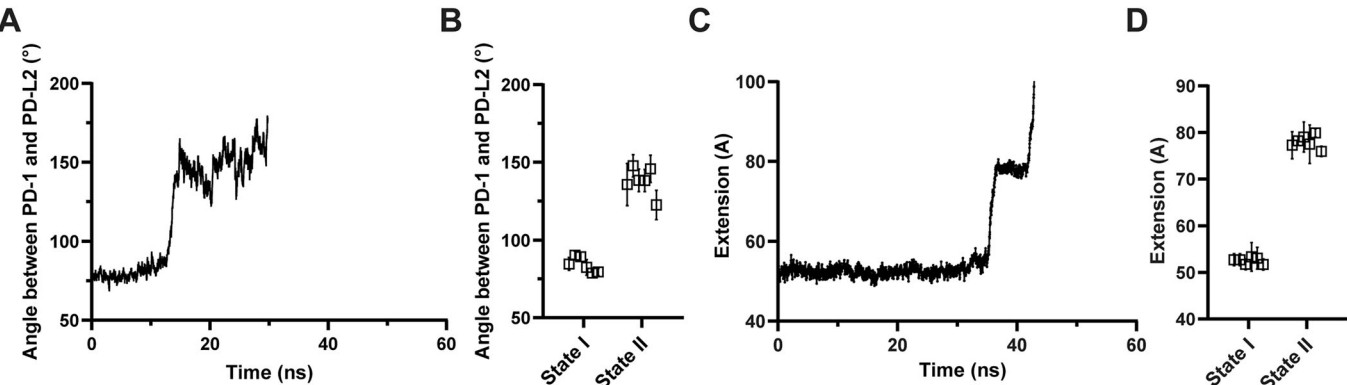

**Figure EV3. PD-1/PD-L2 dissociation by cf-SMD simulations.**

(A) Time-course of the inter-domain angle between PD1 and PD-L2 in one representative cf-SMD simulations, exhibiting two different bending sates; (B) Statistics of the inter-domain angle between PD-1 and PD-L2 for the two binding states in cf-SMD simulations ($n = 6$); (C) Time-course of the CT-CT distance between PD-1 and PD-L2 in the representative cf-SMD simulation shown in (A); (D) Statistics of the CT-CT distance between PD-1 and PD-L2 for the two different binding states in cf-SMD simulations ($n = 6$).

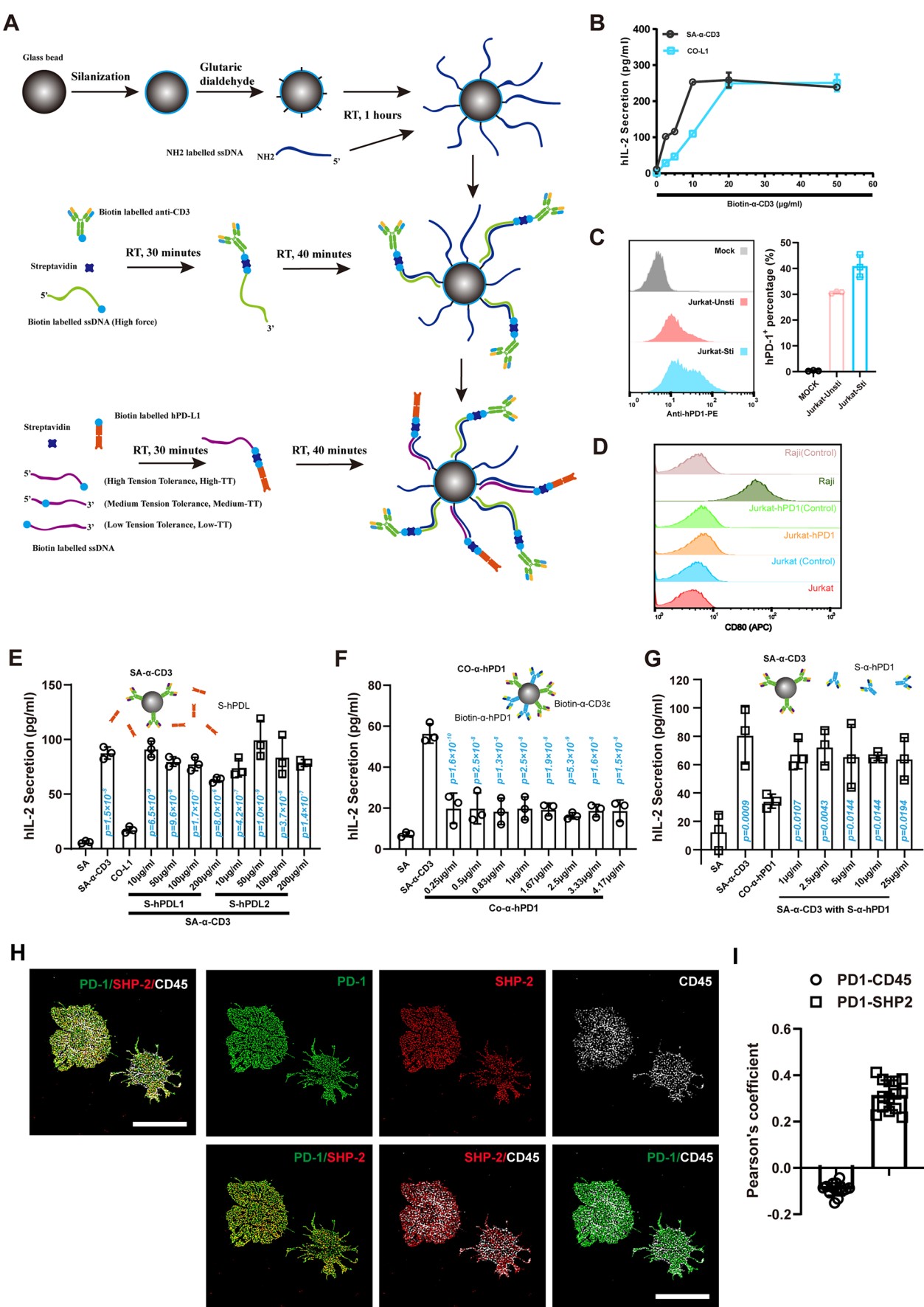

◀  **Figure EV4.  Ligand immobilization is essential for PD-1-mediated T cell inhibition.**

(A) Fabrication of DNA-based tension gauge tether on glass beads (details see Methods); (B) IL-2 secretion of Jurkat cells cocultured with or without PD-L1 (15 µg/ml), data were shown as mean ± SEM, $n = 3$; (C) Cell surface expression of PD-1 in stimulated and unstimulated Jurkat cells, data are shown as mean ± SD, $n = 3$; (D) The expression of CD80 in two Jurkat cell lines (Jurkat-wt and human PD-1 overexpressed Jurkat cell) and Raji cell; (E–G) IL-2 secretion of Jurkat cells stimulated with SA-α-CD3 beads in the presence of soluble PD-L1/PD-L2 (E), CO-α-hPD1 beads alone (F), or SA-α-CD3 beads with soluble PD-L1 antibody (G), data were shown as mean ± SD, $n = 3$; (H, I). The distribution of CD45 and SHP2 in PD-L1 stimulated Jurkat cells (H), the Pearson's coefficient between PD1/CD45 and PD1/SHP2 were quantified (I). Scale bar: 20 µm, $n = 14$. Data information: In panels (B, E–G), 2.5 µg/ml soluble anti-CD28 was used in the experiments, and all the experiments were technically repeated three times. One-way ANOVA is used to produce $p$ value in Fig. EV4D–F.

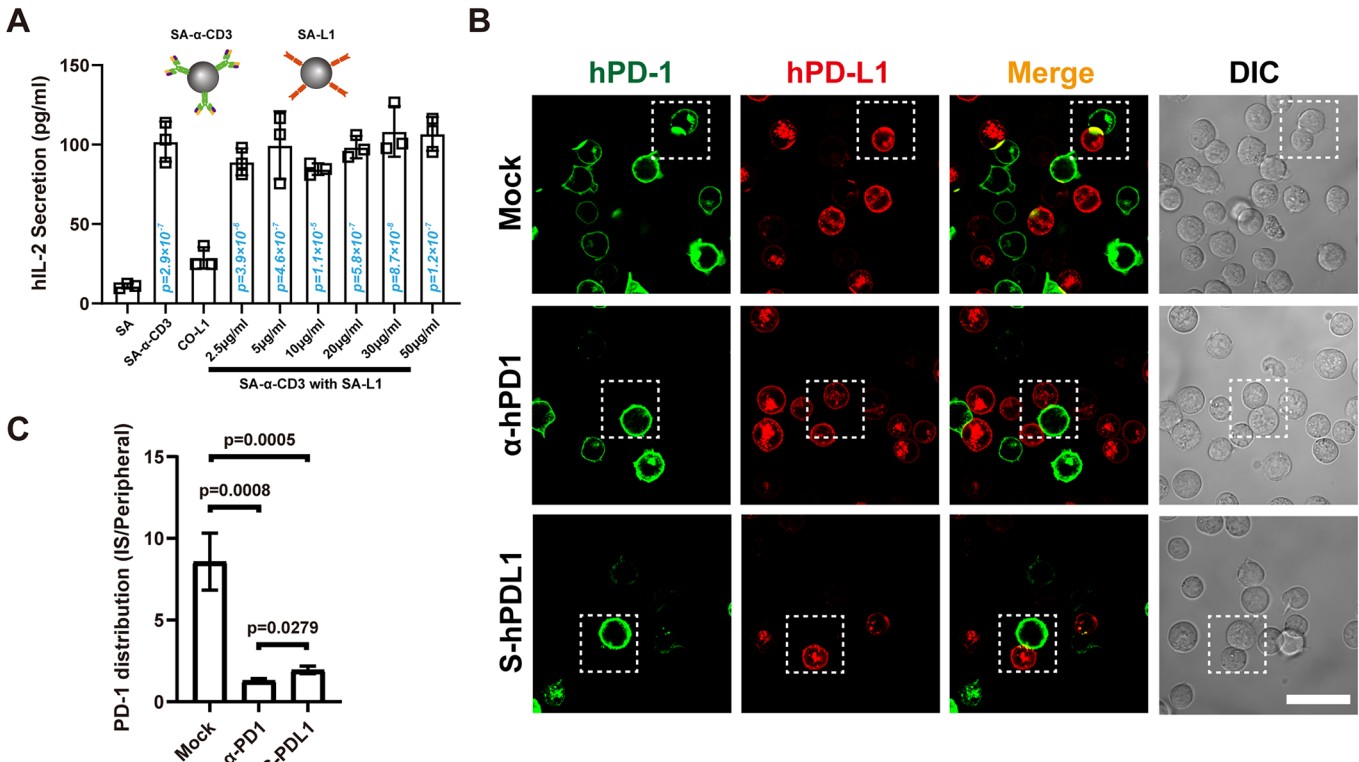

**Figure EV5. PD-1-mediated inhibitory function depends on its IS localization.**

(A) IL-2 secretion of Jurkat cells stimulated with anti-CD3 in the presence of co-immobilized PD-L1 (CO-L1) or PD-L1 immobilized on separated beads with indicated coating concentration, data were shown as mean ± SD, $n = 3$; (B) Representative confocal image of Jurkat (hPD1$^{+}$)-Raji (WT or hPD-L1$^{+}$) conjugates. Cells were imaged to visualize the interaction between Jurkat cells expressing human PD-1 (green) and Raji cells expressing PD-L1 (red); (C) Quantification of PD-1 engagement on cell–cell conjugates. Data were shown as mean ± SEM, Mock: $n = 30$, αPD-1: $n = 22$, S-PDL1: $n = 29$. Data information: One-way ANOVA is used to produce $p$ value in Fig. EV5A, $t$-test is used to quantify the difference between groups in Fig. EV5C.

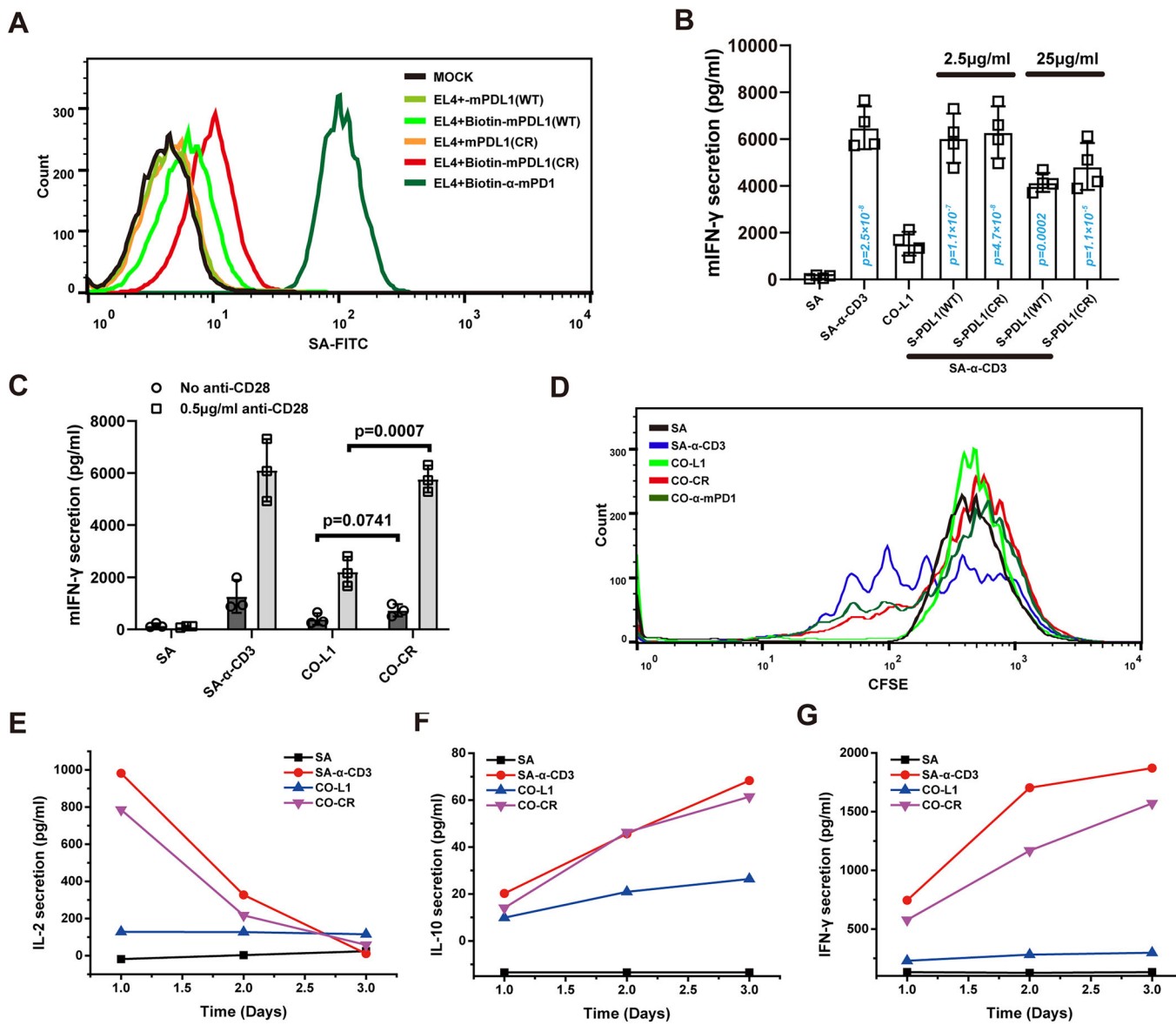

**Figure EV6. Soluble mPDL1-CR mutant blocks the inhibitory function of PD-1 in mice.**

(A) Binding characteristics of wildtype and CR-mutated mouse PD-L1; (B) IFN-γ secretion of mouse primary CD8+ T cells cocultured with anti-CD3 beads in the presence of mouse S-mPDL1(WT) or mouse S-mPDL1(CR), data were shown as mean ± SD, $n = 4$; (C) IFN-γ secretion of mouse primary CD8+ T cells stimulated with indicated beads in the presence or absence of mouse CD28 antibody (0.5 μg/ml), data were shown as mean ± SD, $n = 3$; (D) Proliferation of primary CD8+ T cells stimulated with indicated beads; (E–G) Cytokine secretion of mouse CD8+ T cells activated with indicated beads. Data information: One-way ANOVA is used to produce $p$ value in Fig. EV6B, $t$-test is used to quantify the difference between groups in Fig. EV6C.

