## [Peer Review File · EMBO Reports]

Mechanical force regulates the inhibitory function of PD-1

Hui Chen, Yong Zhang, Lei Cui, Juan Fan, Huaying Zhu, Songfang Wu, Hang Zhou, Yanruo Zhang, Guangtao Song, Ning Jiang, Mingzhao Zhu, Changjie Lou, Wei Chen, and Jizhong Lou

Corresponding author(s): Jizhong Lou (jlou@ibp.ac.cn) , Wei Chen (jackweichen@zju.edu.cn), Hui Chen (cdchenhui@ibp.ac.cn), Changjie Lou (601245@hrbmu.edu.cn)

Review Timeline:

Submission Date:	11th Feb 25
Editorial Decision:	19th Mar 25
Revision Received:	21st Jul 25
Editorial Decision:	20th Aug 25
Revision Received:	26th Sep 25
Editorial Decision:	31st Oct 25
Revision Received:	17th Dec 25
Accepted:	29th Jan 26

Editor: Achim Breiling

Transaction Report:

Dear Prof. Lou,

Thank you for the transfer of your manuscript to EMBO reports. I have now received the reports from the three referees that were asked to evaluate your study, which can be found at the end of this email.

As you will see, the referees think that these findings are of interest. However, they have several comments, concerns, and suggestions, indicating that a major revision of the manuscript is necessary to allow publication of the study in EMBO reports. As the reports are below, and all the referee concerns need to be addressed, I will not detail them here.

Given the constructive referee comments, I would like to invite you to revise your manuscript with the understanding that the concerns of the referees must be addressed in the revised manuscript and in a detailed point-by-point response. Acceptance of your manuscript will depend on a positive outcome of a second round of review. It is EMBO reports policy to allow a single round of revision only and acceptance of the manuscript will therefore depend on the completeness of your responses included in the next, final version of the manuscript.

- 1) a .docx formatted version of the final manuscript text (including legends for main figures, EV figures and tables), but without the figures included. Figure legends should be compiled at the end of the manuscript text.
- 2) individual production quality figure files as .eps, .tif, .jpg (one file per figure), of main figures and EV figures. Please upload these as separate, individual files upon re-submission.

- 4) a complete author checklist, which you can download from our author guidelines (<https://www.embopress.org/page/journal/14693178/authorguide>). Please insert page numbers in the checklist to indicate where the requested information can be found in the manuscript. The completed author checklist will also be part of the RPF.

- 5) that primary datasets produced in this study (e.g. RNA-seq, ChIP-seq, structural and array data) are deposited in an

appropriate public database. If no primary datasets have been deposited, please also state this in a dedicated section (e.g. 'No primary datasets have been generated and deposited'), see below.

The accession numbers and database should be listed in a formal "Data Availability" section that follows the model below. This is now mandatory (like the COI statement). Please note that the Data Availability Section is restricted to new primary data that are part of this study. This section is mandatory. As indicated above, if no primary datasets have been deposited, please state this in this section

Data availability

8) Regarding data quantification and statistics, please make sure that the number "n" for how many independent experiments were performed, their nature (biological versus technical replicates), the bars and error bars (e.g. SEM, SD) and the test used to calculate p-values is indicated in the respective figure legends (also for EV and Appendix figures). Please also check that all the p-values are explained in the legend, and that these fit to those shown in the figure. Please provide statistical testing where applicable. Please avoid the phrase 'independent experiment', but clearly state if these were biological or technical replicates. Please also indicate (e.g. with n.s.) if testing was performed, but the differences are not significant. In case n=2, please show the data as separate datapoints without error bars and statistics. See also: <http://www.embopress.org/page/journal/14693178/authorguide#statisticalanalysis>

9) Please add scale bars of similar style and thickness to microscopic images, using clearly visible black or white bars (depending on the background). Please place these in the lower right corner of the images themselves. Please do not write on or near the bars in the image but define the size in the respective figure legend.

10) Please also note our reference format:

12) We now use CRedit to specify the contributions of each author in the journal submission system. CRedit replaces the author contribution section. Please use the free text box to provide more detailed descriptions and do NOT provide your final manuscript text file with an author contributions section. See also our guide to authors: <https://www.embopress.org/page/journal/14693178/authorguide#authorshippinguidelines>

13) All Materials and Methods need to be described in the main text using our 'Structured Methods' format, which is required for

all research articles. According to this format, the Methods section should include a Reagents and Tools Table (listing key reagents, experimental models, software, and relevant equipment and including their sources and relevant identifiers), uploaded as separate file, and a Methods section in which we encourage the authors to describe their methods using a step-by-step protocol format with bullet points, to facilitate the adoption of the methodologies across labs. More information on how to adhere to this format as well as downloadable templates (.doc) for the Reagents and Tools Table can be found in our author guidelines (section 'Structured Methods'):

14) Please order the sections like this, using these names:

Title page - Abstract - Keywords - Introduction - Results - Discussion - Methods - Data availability section - Acknowledgements (including the funding information) - Disclosure and Competing Interests Statement - References - Figure legends - Expanded View Figure legends

15) Please make sure that all the funding information is also entered into the online submission system and that it is complete and similar to the one in the acknowledgement section of the manuscript text file.

Finally, please note that all corresponding authors are required to supply an ORCID ID upon submission of a revised manuscript and an institutional e-mail address. Please find instructions on how to link the ORCID ID to the account in our manuscript tracking system in our Author guidelines: <http://www.embopress.org/page/journal/14693178/authorguide#authorshipguidelines>

I look forward to seeing a revised form of your manuscript when it is ready.

Yours sincerely,

Referee #1:

Chen et al. presented data that PD-1 forms catch/slip-bonds with its ligands, PD-L1 and PD-L2, highlighting the importance of force in PD-1 function. Using a biomembrane force probe (BFP), they showed that human PD-1 forms catch-bonds with its ligands at forces below 7 pN. Steered molecular dynamics (SMD) simulations predicted force-induced PD-1:PD-L1/2 interfaces and key amino acid interactions, where mutations attenuated PD-1's inhibitory function. A Tension Gauge Tether (TGT) sensor assay revealed that PD-L1 with stronger tethering forces (23 and 56 pN) elicited greater PD-1 inhibitory activity than PD-L1 with weaker tethering force (12 pN).

Notably, the authors discovered that human PD-1 forms stronger catch-bonds with PD-L1 than with PD-L2. This finding is striking given that (1) in the absence of force, human PD-1 binds PD-L2 more strongly than PD-L1 (PMID: 39752535), and (2) in mice, PD-1 forms stronger catch-bonds with PD-L2 than with PD-L1 (PMID: 39333505).

Additionally, the study showed that soluble PD-L1/2 competes with membrane-anchored PD-L1/2 for PD-1 binding, reducing tumor growth in a mouse MC38 tumor model. A key highlight is the development of a gain-of-function soluble PD-L1 mutant, based on the force dependent structure, that more effectively suppresses MC38 tumor growth than PD-L1 WT protein.

In summary, this is a very interesting study offering new insight into the physiological ligand-binding properties of human PD-1 and the across-species divergence of PD-1 biology. The biophysical aspect of this study is strong; however, we do have a few concerns regarding the functional and imaging assays.

Major comments:

1. Figures 1 and 2: Very nice. These are unique biophysical assays that reveal the force dependent nature of the PD-1:PD-L interactions, and also deduce the force-dependent structure, allowing the author to test the functional relevance using point mutations.
2. Figure 3: The differences between PD-L1 and PD-L2 is novel and interesting and can help explain why PD-L2 is a less important ligand despite its higher solution affinity to PD-1.
3. Figure 4. We have several concerns. First, in Figure 4A, PD-1 inhibitory function appears stronger at higher forces (23 and 56 pN) than at 12 pN. However, BFP data in Figure 1 show that PD-1:PD-L1 bond lifetime peaks at 7 pN and declines at higher forces, which seems contradictory. Could the authors clarify this? Second, the claim that force enhances PD-1 localization at immune synapses (Fig. 4C-G) is not well supported. A stronger test would involve PD-1 or PD-L1 mutants with different catch-bonding abilities, but this would require significant effort. How essential is this aspect to the study's main message? Third, Figure 4F and its corresponding paragraph are also unclear. The authors suggest that a PD-1 blocking antibody inhibits PD-1 phosphorylation, yet no such condition is included. Additionally, the data lack key controls, such as PD-1 tyrosine or tailless mutants, and SHP2 recruitment quantification. The rationale for this experiment is unclear, and the claim that "T cell activation without PD-1 stimulation leads to SHP2 recruitment" seems peripheral to the study. We suggest either strengthening this experiment with proper controls and quantification or removing it.
4. Figure 5: the development of the gain of function soluble PD-L1 based on the novel force-dependent structure is exciting. This should be better highlighted in the abstract, since it wouldn't be possible without knowing the catch bond structure. However, we do have concerns on Fig. 5E, which showed that the WT PD-L1 (CO-L1) more strongly inhibited the IFN γ production in mouse primary CD8 T cells compared to both the binding-enhanced mutant PD-L1 (CO-CR) and anti-mouse PD1 (CO-a-mPD1). CO-a-mPD1 condition even upregulated the IFN γ production compared to SA-a-CD3 condition. These results are counter-intuitive - Given that the mutant PD-L1 and anti-PD-1 have higher affinities compared to wild-type PD-L1, shouldn't we expect them to induce greater PD-1 inhibitory functions?

Minor comments:

1. Figure 3: Discussing the across-species divergence of PD-1 catch-bond would provide valuable context for understanding the new finding of this paper.
2. Figure 4A: Adding an illustration depicting the Tension Gauge Tether (TGT) sensor assay would help understanding the data.
3. There are several typos and grammatical errors that need to be corrected.

Referee #2:

In this paper, the authors investigate the mechanical properties of human PD1 binding its ligands, which has recently been demonstrated to form catch bonds in BFP assays. A true understanding of all the factors influencing PD1 activation are necessary for the design of therapeutic agents targeting this receptor, and the authors investigate the importance of force as a requirement for PD1 activation. The paper uses a combination of experimental setups to interrogate the response of PD1-ligand interactions to imposed force, and complements this with point mutations to support evidence of catch bond behaviour. The authors conclude by constructing a high affinity mouse PD-L1 mutant which demonstrates efficacy in a tumour model. The paper is well organised, and the figures are clear, although at times the grammar caused difficulties in comprehension. While this paper goes some way in supporting the catch bond nature of the PD1-ligand interaction, we provide major and minor comments which we hope would further support this finding:

Major Comments:

- While the BFP assays throughout this paper (figures 1-3) clearly indicate catch bond formation, this system has previously been shown to display memory when the same T cell is repeatedly stimulated (<https://www.pnas.org/doi/10.1073/pnas.0704811104>). Could the authors clarify whether the same T cell was re-used over multiple forces, and if so, what order the forces were applied? In other words, are the lifetimes displayed throughout the manuscript truly independent measures? To conclude robustly that force increases bond lifetime, the same results should be achieved when T cells are not re-used, or when lifetimes are measured in response to forces in random order using the same T cell.
- Throughout this paper, caution should be taken when concluding whether force is required for productive signalling or is simply observed across a bond when two cells interact. This is clearest in the TGT-based assays, where the authors state in the discussion: "TGT-based T cell co-stimulation experiments further demonstrate that mechanical loading on the PD-1/PD-L1 interaction is essential for PD-1 function". The TGT results shown in figure 4 use probes which compromise the function PD-L1 beyond discrete force intervals. Because of this, it cannot be concluded that mechanical force is required for PD1 activation, only that it is exerted on the PD1-PDL1 bond. If the force had no functional effect on PD1 activation, the same results would be observed. To show that force is required for signalling, you would need a native PD1-PDL1 interaction without force and show that this prevents PD-1 signalling/inhibition.
- There is not much emphasis on why force might be important for PD1 activation. In the case of TCR-pMHC interactions, catch bonds have been proposed to increase the ability of T cells to discriminate antigens based on affinity/lifetime. In the discussion (page 13), two possible mechanisms of PD-1 activation are proposed by the authors, however neither of these discussed in

sufficient detail prior to this suggestion

- o The first proposal (CD45 exclusion) bears no relevance to this paper, and is instead a reiteration of existing literature. Care should be taken when referring to these hypotheses as the authors' own proposals.
- o The second suggestion is much more relevant to the ideas presented in this paper, however this is the first time that molecular mechanisms of force-induced PD-1-activation are discussed. If the paper presented evidence to support this hypothesis, then this conclusion would bear more weight. For example, further simulations of the entire PD-1 receptor under force, or point mutations intended to disrupt the suggested auto-inhibitory conformation would support this hypothesis.
- While the first half of this paper (figures 1-3) examines the mechanical properties of PD1-PDL1, the following figures examine the development of a soluble ligand for the inhibition of PD1. It is not clear how these lead on from one another, making the narrative feel disjointed. This could be summarised in the concluding paragraph: "Our current study highlights the importance of mechanical force in PD-1 signaling by inducing catch bond between PD-1 and its ligands, and soluble PD-L1, where mechanical force application is disabled, could be a potential agent for cancer immunotherapy". While the authors' suggestion that soluble ligands cannot mediate force-induced activation is valid, it is not clear how this can be utilised to inform the design of improved soluble antagonists of PD1, where on this occasion the improved ligand was simply designed to have a higher solution affinity. Additionally, this suggestion does not account for the observation/design of soluble PD1 agonists which rely on the aggregation of PD1 to activate signalling e.g. <https://www.science.org/doi/10.1126/sciimmunol.add4947> and <https://pmc.ncbi.nlm.nih.gov/articles/PMC8307378/>
- Although the mechanism of CD45 exclusion is stated in the discussion, the remainder of the paper appears to lack alternative non-mechanical explanations for the results observed, namely that surface-immobilisation of PD1 ligands will also facilitate the kinetic segregation mechanism of activation.
- o The following statement in the introduction demonstrates this: "Given the force dependence of these interactions, soluble and immobilized ligands behave differently on PD-1's inhibitory function, as soluble ligands can bind PD-1 but unable to provide an external force, and therefore cannot trigger its inhibitory function."
- o The results also state: "To determine whether mechanical force alone is sufficient to trigger PD-1 signaling, we stimulated Jurkat cell with CO- α -hPD1 beads (streptavidin beads simultaneously ligating human PD-1 antibody and CD3 antibody) in the presence of CD28 antibody", but it is unclear how this isolates the effect of mechanical force alone, given that the ligand is still surface-immobilised and aggregation/kinetic segregation can still occur.
- o Note that elongated PD-1 reduces its function in immobilised form yet may not have an impact on force. This was shown by two studies <https://pubmed.ncbi.nlm.nih.gov/22641383/> and <https://www.biorxiv.org/content/10.1101/2025.01.06.631424v1> (Fig 6)
- o When beads are used to present PD1 ligands (Figure 4D/4E/5A), it is also unclear whether these are intended to mimic soluble or surface-immobilised ligand.
- While the force-lifetime curves displayed (Figures 1-3) are representative of catch bonds, the molecular mechanisms of the two-state model of catch bond formation could be clarified. The second state must have a longer lifetime, thus increasing the observed lifetime as the bond shifts from state I to state II, and this is not made clear in the following sections:
- o The simulations in figure 1 only display the angle and extension of the two states, but it is not clear what separates this observation from that of a slip bond dissociating over multiple states. It should be indicated that the second state has a lower energy in these simulations - e.g. how does the total number of salt bridges/H bonds compare, especially given specific interactions are then mutated in figure 2.
- o In figure 2D, the residues mutated are suggested to disrupt only state I and not state II. Given that increasing the force would shift the bond to state II, why is the peak lifetime of the mutants at higher forces significantly lower than the wildtype? By decreasing the lifetime of state I, you should observe an enhanced catch bond due to a greater difference in lifetime between the two states.
- o The double mutant is stated to have rescued its catch bond in figure 2D, but this is not clear from the results, as the total increase in lifetime with force is very similar to the single mutants. To prove that the double mutant has regained wildtype catch bonding, further tests such as survival frequencies and F1/2 calculations using force ramps could demonstrate this.
- o For all mutants assessed in figure 2, it is important to demonstrate that solution affinity is unchanged with SPR, and the mutations therefore only affect the mechanical response to force.

Minor Comments:

- In the abstract, the following phrase is used: "we discovered that interactions between PD-1 and PD-L1/PD-L2 exhibit catch-slip bond behavior under force". Please note that this finding has recently been published by Li et al., and therefore using the word 'discovered' may appear misleading.
- For the following sentence in the results, "We controlled the site density of PD-L1 on the beads to maintain an adhesion frequency below 20%, ensuring predominantly single-bond interactions between PD-L1 and PD-1 on the cell surface (Figure. S1A and S1B)":
 - o Could the authors please clarify why a frequency below 20% is associated with predominantly single-bond interactions, especially when the association rates without PDL1 range from ~2-7%
 - o We also note that Figure S1A displays a value of >20% which contradicts the statement in the main text
- For the following comment in the results, "While previous reports suggest that PD-L1 can bind CD80 on T cells (27), recent studies increasingly support PD-L1 primarily binding CD80 in cis on the same cell surface (28-30). Our data aligns with these findings", it is unclear how the data here aligns with these findings, and it has not been demonstrated that PD-L1 in cis or trans impacts results.
- For the following comment in the results, "Interestingly, removing force from state II snapshots reverted the conformation back

to state I (Figure. S1F, S1G), suggesting reversible conformational changes between states depending on force application. This switching likely enhances the stability of PD-1/PD-L1 interactions under low force, resulting in catch bonds", it is unclear from this explanation why enhanced stability of state I interactions under low force results in catch bonds.

- In figure S3, it is not clear how the BFP assay has been altered to allow PD1 to be recombinantly expressed, as opposed to presented natively on T cells. In addition, the main text explains that PD-L1 was expressed by E. Coli, whereas the caption of Figure S3 states that all proteins were expressed by 293F cells.
 - It would be helpful to see a figure depicting the TGT assay in figure 4
 - In figure 5C, could the authors provide multiple examples of cell-cell conjugates, as providing a single example may not be representative of the effect of S-hPDL1. No uncropped images have been provided in the supplementary figures.
 - In the introduction and discussion, it is stated that the TCR utilises mechanical force to enhance discrimination and that TCRs form catch bonds with antigens:
 - o "It has been demonstrated that TCR functions of distinguish non-self from self antigens with the aid of mechanical force (21, 22)." - Introduction
 - o "Mechanical force plays a vital role in T cell activation, with the TCR functioning as a mechano-sensor by forming catch bonds with non-self antigens and slip bonds with self antigens (21, 22, 41)." - Discussion
- There is evidence against this model that the authors should acknowledge.
- In the discussion it is stated "These results are in good agreement with our findings" with respect to the recent Li et al. paper. Although this is true for the results of mouse PD1-PDL1, it should be noted that the results for human PD-L1 vs PD-L2 display the opposite trend to the results of mouse PD-L1 vs PD-L2 in the Li et al paper, where it is shown PD-L2 forms a stronger catch bond with PD1 than PD-L1.

Referee #3:

The paper by Chen et al. is a tour-de-force characterization of PD-1 interactions with PD-L1 and PD-L2. The wealth of data provided here is impressive, and contains numerous potentially highly interesting insights. Still, I think it would be important for the authors to clarify the following points:

- 1.) The authors are apparently aware of a similar article by Li et al (<https://doi.org/10.1038/s41467-024-52565-2>), which is also cited here as reference 42. However, the reference to Li et al. is only given in the discussion section of this paper. Given that the catch bond nature of PD-1 with PD-L1 and with PD-L2 has also been found in Li et al., the novelty of large parts of Fig.1, 2, and 3 is compromised. This should be stated openly. Results that are of confirmatory nature should be presented as such.
- 2.) I didn't fully understand how to interpret Fig. 4A. Using TGTs with characteristic rupture force of 12pN don't allow for PD-L1 mediated signal suppression, but at 23pN signals do get suppressed. But isn't 12pN already in the slip bond regime of the PD-1/PD-L1 interaction? In other words, the longest bond duration should occur if cells pulled at the peak of the curve shown in Fig. 1B, around 7pN. In contrast 12pN and 23pN would give fairly similar lifetimes.

We thank the reviewers for their positive feedback, insightful comments, and constructive suggestions, which have significantly enhanced the quality of our work. We have thoroughly addressed their concerns to further substantiate our findings. Below are our point-by-point responses to the referees' comments. The original comments are presented in black, and our replies are in blue.

Referee #1:

Chen et al. presented data that PD-1 forms catch/slip-bonds with its ligands, PD-L1 and PD-L2, highlighting the importance of force in PD-1 function. Using a biomembrane force probe (BFP), they showed that human PD-1 forms catch-bonds with its ligands at forces below 7 pN. Steered molecular dynamics (SMD) simulations predicted force-induced PD-1:PD-L1/2 interfaces and key amino acid interactions, where mutations attenuated PD-1's inhibitory function. A Tension Gauge Tether (TGT) sensor assay revealed that PD-L1 with stronger tethering forces (23 and 56 pN) elicited greater PD-1 inhibitory activity than PD-L1 with weaker tethering force (12 pN).

Notably, the authors discovered that human PD-1 forms stronger catch-bonds with PD-L1 than with PD-L2. This finding is striking given that (1) in the absence of force, human PD-1 binds PD-L2 more strongly than PD-L1 (PMID: 39752535), and (2) in mice, PD-1 forms stronger catch-bonds with PD-L2 than with PD-L1 (PMID: 39333505).

Additionally, the study showed that soluble PD-L1/2 competes with membrane-anchored PD-L1/2 for PD-1 binding, reducing tumor growth in a mouse MC38 tumor model. A key highlight is the development of a gain-of-function soluble PD-L1 mutant, based on the force dependent structure, that more effectively suppresses MC38 tumor growth than PD-L1 WT protein.

In summary, this is a very interesting study offering new insight into the physiological ligand-binding properties of human PD-1 and the across-species divergence of PD-1 biology. The biophysical aspect of this study is strong; however, we do have a few concerns regarding the functional and imaging assays.

Response: We thank the reviewer for the positive evaluation of our study. We have added discussion (*Line 341-352*) of the difference in PD-1 binding to PD-L1 and PD-L2 in our revised manuscript. See also the response to minor comment 1.

Major comments:

1. Figures 1 and 2: Very nice. These are unique biophysical assays that reveal the force dependent nature of the PD-1:PD-L interactions, and also deduce the force-dependent structure, allowing the author to test the functional relevance using point mutations.

Response: We thank the reviewer for the positive comments.

2. Figure 3: The differences between PD-L1 and PD-L2 is novel and interesting and can help explain why PD-L2 is a less important ligand despite its higher solution affinity to PD-1.

Response: We thank the reviewer for the positive comments.

3. Figure 4. We have several concerns. First, in Figure 4A, PD-1 inhibitory function appears stronger at higher forces (23 and 56 pN) than at 12 pN. However, BFP data in Figure 1 show that PD-1:PD-L1 bond lifetime peaks at 7 pN and declines at higher forces, which seems contradictory. Could the authors clarify this?

Response: We appreciate the reviewers' concern about the DNA Tension Gauge Tether (TGT) approach. Actually, the temperature, salt concentration (especially Mg²⁺) and loading rate are all contributing to the rupture force of DNA TGTs. This design was firstly developed by Taekjip Ha group (Science, 2013), and the rupture forces of these TGT were estimated with an optimized equation (Equation (1), Science, 2013),

$$F = 2f_c \left[\chi^{-1} \tanh\left(\chi \frac{L}{2}\right) + 1 \right] \quad (1)$$

the key parameters f_c and χ^{-1} were measured with magnetic tweezers at room temperature in PBS buffer (Hatch K., *et.al.*, Physical Review E, 2008).

In our study, we ligated protein/antibody with SA beads by biotin-SA interaction, and then incubated these beads with Jurkat cells in DMEM medium at 37 °C for 24 hours. That is, the temperature used in our experiments is much higher than that calibrating the rupture force of DNA TGTs in magnetic tweezers experiment. Higher temperature will destabilize DNA TGTs. Thus, the rupture force of the "12pN" TGT in our study could be much lower than 12pN.

Second, the claim that force enhances PD-1 localization at immune synapses (Fig. 4C-G) is not well supported. A stronger test would involve PD-1 or PD-L1 mutants with different catch-bonding abilities, but this would require significant effort. How essential is this aspect to the study's main message?

Response: We thank the reviewer for this insightful suggestion and concur that the inclusion of PD-1 or PD-L1 mutants would offer more robust support for our conclusion. To address this, we immobilized either wildtype or mutated PD-L1 on glass surface by SA-biotin interaction. Jurkat cells expressing human PD-1 were then added and incubated for 30 min, after which the cells were fixed for TIRF-SIM imaging. As expected, wildtype PD-L1 recruited PD-1 to the glass-cell interacting surface, and SHP-2_{SH2-SH2} was also recruited. In contrast, all PD-L1 mutants (R113E/R125E, I54A, Q66A and K75A) show impaired recruitment of both PD-1 and SHP-2_{SH2-SH2} (Figure R1). Here, we used PD-1 fused with mEGFP to quantify PD-1 recruitment. We don't test PD-1 mutants, because it's hard to ensure that the mean density of PD-1 mutants on Jurkat cells is the same as that of wildtype PD-1. We have also added these data in the revised manuscript (Figure 4F)

Figure R1. PD-1 phosphorylation in Jurkat cells.

Jurkat cells (PD-1-mEGFP and SHP-2_{SH2-SH2}-mCherry overexpressed) were stimulated with indicated SLB-ligated wildtype or mutated PD-L1 for 40 min. Typical images were shown. Scale bar: 10 μ m.

Third, Figure 4F and its corresponding paragraph are also unclear. The authors suggest that a PD-1 blocking antibody inhibits PD-1 phosphorylation, yet no such condition is included.

Response: We thank the reviewer for bringing this point to our attention. While it is established that PD-1 inhibits T cell activation through phosphorylation-dependent recruitment of SHP-2, which in turn dephosphorylates CD28, CD3 and PD-1 itself. PD-1 blocking antibodies could restore T cell function by preventing ligand binding and subsequent SHP-2 recruitment. We recognize that our previous assertion about inhibition of PD-1 phosphorylation as the primary mechanism was speculative and unsupported by direct evidence in the context of our study. Thus, we have removed the statement from the manuscript to maintain the integrity of our findings. We appreciate the reviewer's guidance in strengthening the rigor and clarity of our work.

Additionally, the data lack key controls, such as PD-1 tyrosine or tailless mutants, and SHP2 recruitment quantification. The rationale for this experiment is unclear, and the claim that "T cell activation without PD-1 stimulation leads to SHP2 recruitment" seems peripheral to the study. We suggest either strengthening this experiment with proper controls and quantification or removing it.

Response: We sincerely appreciate the reviewer's constructive feedback and acknowledge the need to strengthen this section with appropriate controls and

quantification. Our aim was to characterize the basal phosphorylation state of PD-1's intracellular domain to inform our model of PD-1 triggering. Two possible models for PD-1 triggering were proposed in the Discussion section of our original manuscript, a phosphorylation equilibrium model and a ligand-dependent conformational change model.

Our data showed that PD-1 can indeed undergo phosphorylation in the absence of ligand binding, which partially supports the notion that PD-1 exists in a dynamic phosphorylation/dephosphorylation equilibrium in the resting state. In order to verify the relationship between PD-1 activation and CD45 exclusion, we stimulated PD-1-expressing Jurkat cells with surface-bound PD-L1 and examined the distribution of PD-1, SHP-2, and CD45 on the surface by TIRF-SIM (Figure R2A and R2B), and the results showed that the distribution of PD-1 was negatively correlated with the distribution of CD45 and positively correlated with that of SHP-2 (Figure R2C). The result indicates that CD45 exclusion may promote the activation of PD-1. We have also added these data in the revised manuscript (Figure EV6G and EV6I)

Figure R2. The distribution of CD45 and SHP-2 in PD-L1 stimulated Jurkat cells (A), single channel of PD-1, SHP-2 and CD45 (upper panel) and two channel superimposed figure were shown in (B), the Pearson's coefficient between PD-1/CD45 and PD-1/SHP-2 were quantified (C). Scale bar: 20 μm, n=14.

However, these findings may still be insufficient to definitively distinguish between the proposed catch-bond mechanism (stabilizing ligand binding, excluding CD45, and altering PD-1 intracellular domain phosphorylation equilibrium) and the model of PD-1 autoinhibition. Specifically, these results are also compatible with an alternative scenario: phosphorylation of CD3 and CD28 may modulate the state of PD-1's intracellular domain, rendering it more susceptible to Lck-mediated phosphorylation, with CD45 exclusion playing a facilitative rather than decisive role in receptor triggering. Given this ambiguity and the existence of similar prior reports, we agree with the reviewers' recommendation to remove this section (the original Figure 4F and its corresponding paragraph) from the manuscript.

To strengthen the mechanistic rigor of our work, we have performed additional experiments characterizing PD-L1 mutants' capacity to mediate PD-1 engagement and SHP-2 recruitment. Statistical analyses confirm that wild-type PD-L1 exhibits superior binding to PD-1 (Figure R3A) and drives more efficient SHP-2 recruitment (Figure R3B) compared to mutants, providing clearer evidence for the functional relevance of PD-1/PD-L1 interaction dynamics. We have also added these data in the revised manuscript (Figure 4G and 4H).

Figure R3. Quantification of PD-1 intensity (A) and SHP-2 intensity (B) in contact area (Typical images were shown in Figure R1).

4. Figure 5: the development of the gain of function soluble PD-L1 based on the novel force-dependent structure is exciting. This should be better highlighted in the abstract, since it wouldn't be possible without knowing the catch bond structure. However, we do have concerns on Fig. 5E, which showed that the WT PD-L1 (CO-L1) more strongly inhibited the IFN γ production in mouse primary CD8 T cells compared to both the binding-enhanced mutant PD-L1 (CO-CR) and anti-mouse PD1 (CO-a-mPD1). CO-a-mPD1 condition even upregulated the IFN γ production compared to SA-a-CD3 condition. These results are counter-intuitive - Given that the mutant PD-L1 and anti-PD-1 have higher affinities compared to wild-type PD-L1, shouldn't we expect them to induce greater PD-1 inhibitory functions?

Response: We really appreciate the reviewer's evaluation of our work, and have revised the abstract to highlight the gain-of-function soluble PD-L1 mutant.

Regarding Figure 5E, Flow cytometry (Figure EV8A) confirmed that mouse PD-L1 mutants (mPDL1-CR) exhibited stronger PD-1 binding in the absence of force. Additionally, our BFP experiments revealed that the CR mutant had a longer lifetime than wildtype at lower forces (2-6 pN) but a shorter lifetime at higher force (> 6 pN) (Figure 5D).

This biphasic behavior may partially explain why CO-CR shows weaker inhibition of IFN- γ compared to CO-L1. We chose the CR mutant for mouse PD-1 blockade therapy due to its stronger PD-1 binding in solution and under low force regime.

We observed that immobilized PD-1 antibody in mice failed to suppress T-cell

activation and instead enhanced it, contradicting our expectations and findings in human. We hypothesize that this discrepancy arises from differential epitope binding and steric effects. Specifically, the large size of the antibody may induce a spatial configuration where the PD-1/antibody complex exceeds the spacing of TCR-antibody complexes. This misalignment could physically segregate PD-1 from TCR signaling microclusters, abrogating its inhibitory function. Alternatively, PD-1-antibody interactions might strengthen cell adhesion to beads, inadvertently triggering co-stimulatory pathways that override PD-1's inhibitory function.

Minor comments:

1. Figure 3: Discussing the across-species divergence of PD-1 catch-bond would provide valuable context for understanding the new finding of this paper.

Response: Our results indicated that for human, hPD-1 form stronger catch-bond with PD-L1 than PD-L2; While for mouse, mPD-1 form weaker catch bond with PD-L1 than PD-L2 [PMID: 39333505]. This highlighted the across-species divergence of PD-1 revealed by a recent study [PMID: 39752535]. We have added some discussions in our revised manuscript (*Line 337-340*).

2. Figure 4A: Adding an illustration depicting the Tension Gauge Tether (TGT) sensor assay would help understanding the data.

Response: We thank the reviewer for the suggestion. We have added an illustration figure to depicting the assay to construct the TGT sensor (Figure R4, also Figure EV5 in the revised manuscript).

Figure R4 Fabrication of DNA-based tension gauge tether on glass beads. (Details see methods)

3. There are several typos and grammatical errors that need to be corrected.

Response: We are sorry for these typos and grammatical errors, we have revised the manuscript carefully and corrected all the typos and grammatical errors we found.

Referee #2:

In this paper, the authors investigate the mechanical properties of human PD1 binding its ligands, which has recently been demonstrated to form catch bonds in BFP assays. A true understanding of all the factors influencing PD1 activation are necessary for the design of therapeutic agents targeting this receptor, and the authors investigate the importance of force as a requirement for PD1 activation. The paper uses a combination of experimental setups to interrogate the response of PD1-ligand interactions to imposed force, and complements this with point mutations to support evidence of catch bond behaviour. The authors conclude by constructing a high affinity mouse PD-L1 mutant which demonstrates efficacy in a tumour model. The paper is well organised, and the figures are clear, although at times the grammar caused difficulties in comprehension.

While this paper goes some way in supporting the catch bond nature of the PD1-ligand interaction, we provide major and minor comments which we hope would further support this finding:

Major Comments:

- While the BFP assays throughout this paper (figures 1-3) clearly indicate catch bond formation, this system has previously been shown to display memory when the same T cell is repeatedly stimulated (<https://www.pnas.org/doi/10.1073/pnas.0704811104>). Could the authors clarify whether the same T cell was re-used over multiple forces, and if so, what order the forces were applied? In other words, are the lifetimes displayed throughout the manuscript truly independent measures? To conclude robustly that force increases bond lifetime, the same results should be achieved when T cells are not re-used, or when lifetimes are measured in response to forces in random order using the same T cell.

Response: We thank the reviewer to pointing this issue out. We totally agree that this is a very important issue, also tried to avoid the potential memory affect. We are sorry that we did not include methods' detail in the original manuscript. In our BFP studies, the T cells are reused over various forces, also, lifetime for each force level are from various T-cells. For each T-cell used, the magnitude of the applied force order is manually chosen from three scenario: a low-high-low sequence, a high-low-high sequence, or a random order. Also, the T cell will be replaced as soon as it begins to change shape. We did not perform the experiment with T cells not re-used because it is really time consuming.

- Throughout this paper, caution should be taken when concluding whether force is required for productive signalling or is simply observed across a bond when two cells interact. This is clearest in the TGT-based assays, where the authors state in the discussion: "TGT-based T cell co-stimulation experiments further demonstrate that mechanical loading on the PD-1/PD-L1 interaction is essential for PD-1 function". The TGT results shown in figure 4 use probes which compromise the function PD-L1 beyond discrete force intervals. Because of this, it cannot be concluded that mechanical force is required for PD1 activation, only that it is exerted on the PD1-PDL1 bond. If the force had no functional effect on PD1 activation, the same results would be observed. To show that

force is required for signalling, you would need a native PD1-PDL1 interaction without force and show that this prevents PD-1 signalling/inhibition.

Response: We thank the reviewer for the insightful comment. It is true that catch-bond behavior does not mean the across membrane signaling is force dependent. Thus, we didn't draw conclusions based solely on BFP or TGT data. To test the correlation between force and signaling, we compared assays in which immobilized PD-L1s (where force can be exerted) and soluble PD-L1s (where force cannot be generated) were used (Figure 4 and Figure 5A). The study with soluble PD-L1s (without force) do show that it prevents PD-1's signaling (loss of inhibition effect) (Figure 4B). Moreover, soluble PD-L1 can compete with immobilized PD-L1 and the inhibition effect by immobilized PD-L1 can be diminished as the concentration of soluble PD-L1 was increased (Figure 5A). From these data, we conclude that the signaling of PD-1 is force dependent.

- There is not much emphasis on why force might be important for PD1 activation. In the case of TCR-pMHC interactions, catch bonds have been proposed to increase the ability of T cells to discriminate antigens based on affinity/lifetime. In the discussion (page 13), two possible mechanisms of PD-1 activation are proposed by the authors, however neither of these discussed in sufficient detail prior to this suggestion
 - o The first proposal (CD45 exclusion) bears no relevance to this paper, and is instead a reiteration of existing literature. Care should be taken when referring to these hypotheses as the authors' own proposals.

Response: We thank the reviewer for this insightful comment. In our opinion, PD-1/PD-L1 interaction occur in the cell-cell interface where force can be generated from multiple origins. In this mechanical environment, catch-bond may help PD-1 to transmit signal more efficiently (by using longer time), and make PD-1 stay longer within the vicinity of TCR, CD28 and maybe other co-activating molecules. Then SHP-2 molecule which PD-1 recruited can also take affect longer. We believe further in-depth investigations are required to dissect why force might be important for PD-1 activation.

The exclusion of CD45 has been investigated in TCR-pMHC, CD2-CD58 and PD1-PDL1 interaction pairs. These results are important for studying the mechanism of inhibitory function of CD45 as well as the activation mechanism of many receptors. We introduce CD45 exclusion here because the force-induced stabilization of catch bond facilitates this process, and the size of PD-1/PD-L1 complex is smaller than that of the ecto-domain of CD45. Thus, PD-1/PD-L1 complexes, may assist TCR-pMHC complexes to bring the two cells closer and maintain the exclusion of the large-size CD45. Forming catch bond allows for a more stable binding of PD-1 to its ligands in a force-loading environment, which would contribute to an increase in the dwell time of PD-1/PD-L1 at the T cell-target cell interaction interface as well as the exclusion of CD45. Of course, as we mentioned in our reply to reviewer 1 (Major-3-3), our current results do not prove whether this model is correct or not, but only provide a possible explanation, further in-depth studies are required to reveal the physiological significance of the catch bond phenomenon in the process of receptor activation.

We agree that the discussion on the potential mechanism of PD-1 inhibition function is kind of reiteration of existing literature. We have rephrased the corresponding statement to make clear that it is not our own proposals.

o The second suggestion is much more relevant to the ideas presented in this paper, however this is the first time that molecular mechanisms of force-induced PD-1-activation are discussed. If the paper presented evidence to support this hypothesis, then this conclusion would bare more weight. For example, further simulations of the entire PD-1 receptor under force, or point mutations intended to disrupt the suggested auto-inhibitory conformation would support this hypothesis.

Response: We sincerely appreciate the reviewer's insightful suggestion, which aligns closely with our ongoing investigations. We indeed have already performed MD simulations to characterize the conformational dynamics of full-length PD-1 embedded in a lipid bilayer (Figure R5). MD simulations began with an extended PD-1 conformation (Figure R5A, left) and were repeated 10 times with distinct starting configurations, each running for ~200 ns without external forces. Analysis of the final ~100 ns trajectories revealed that the intracellular domain (ICD) consistently curled toward the inner leaflet (Figure R5A, right). Using contact area calculations (Figure R5B-C), we identified key interaction hotspots: R203-R204 (juxtamembrane region), Y248, M257, and P271-R272, with the strongest contacts with lipids. These findings indicate that PD-1 ICD-lipid interactions are moderate yet specific, suggesting a regulatory role in basal states.

Building on this, we hypothesize that mechanical forces applied to the extracellular domain (ECD) of PD-1 could modulate ICD's conformation, facilitating transmembrane signaling. While preliminary MD simulations under applied force are in progress, we acknowledge that this speculation requires further validation. Given the complexity of force-dependent effects, we plan to comprehensively address this mechanism in a subsequent study.

Figure for referee with unpublished data and its description has been removed upon request by the authors.

- While the first half of this paper (figures 1-3) examines the mechanical properties of PD1-PDL1, the following figures examine the development of a soluble ligand for the inhibition of PD1. It is not clear how these lead on from one another, making the narrative feel disjointed. This could be summarised in the concluding paragraph: "Our current study highlights the importance of mechanical force in PD-1 signaling by inducing catch bond between PD-1 and its ligands, and soluble PD-L1, where mechanical force application is disabled, could be a potential agent for cancer immunotherapy". While the authors' suggestion that soluble ligands cannot mediate force-induced activation is valid, it is not clear how this can be utilised to inform the design of improved soluble antagonists of PD1, where on this occasion the improved ligand was simply designed to have a higher solution affinity. Additionally, this suggestion does not account for the observation/design of soluble PD1 agonists which rely on the aggregation of PD1 to activate signalling:

e.g. <https://www.science.org/doi/10.1126/sciimmunol.add4947> and <https://pmc.ncbi.nlm.nih.gov/articles/PMC8307378/>

Response: Thank the reviewer for the suggestion of a concluding paragraph, in the revised manuscript, we have added a summary paragraph to conclude our findings (*Line 399-408*).

Our study underscores the critical role of mechanical force in PD-1 activation via catch bond formation with monomeric PD-L1, a mechanism requiring force-dependent conformational stabilization. However, as noted in our original manuscript and supported by recent work (Phillips et al., *Sci. Immunol.* 2024), PD-L1 dimers/oligomers may initiate signaling through alternative mechanisms, including PD-1 clustering mediated by transmembrane domain dimerization. This duality may suggest at least two non-mutually exclusive models for PD-1 activation: (i) force-dependent conformational switching induced by monomeric ligands, (ii) force-independent clustering driven by multivalent interactions. Of course, further studies are required to prove/disprove both models.

Regarding PD-1 agonist antibodies, their mechanism likely involves spatial reorganization and/or allosteric regulation. Bivalent binding may crosslink preform PD-1 dimers or induce clustering, mimicking oligomeric PD-L1 and excluding CD45 phosphatase. Immobilized monomeric binding could stabilize the active conformations of extracellular region, propagating through the transmembrane domain to prime intracellular phosphorylation. These hypotheses, while plausible, remain speculative and beyond the scope of our current study. We emphasize that mechanical force is essential for monomeric ligand-mediated activation but may be dispensable for multivalent interactions. Future work should delineate context-specific contributions of force and oligomerization, particularly for therapeutic antibodies targeting distinct PD-1 epitopes.

- Although the mechanism of CD45 exclusion is stated in the discussion, the remainder of the paper appears to lack alternative non-mechanical explanations for the results

observed, namely that surface-immobilisation of PD1 ligands will also facilitate the kinetic segregation mechanism of activation.

o The following statement in the introduction demonstrates this: "Given the force dependence of these interactions, soluble and immobilized ligands behave differently on PD-1's inhibitory function, as soluble ligands can bind PD-1 but unable to provide an external force, and therefore cannot trigger its inhibitory function."

Response: Thank the review for this comment, we are sorry that we did not explain our opinion clearly in our statement referred. As our reply to the previous comment, there may exist at least two non-mutually exclusive models for PD-1 activation: (i) force-dependent conformational switching induced by monomeric ligands, (ii) force-independent clustering driven by multivalent interactions.

In this statement, "ligands" specifically denote monomeric PD-1 ligands. In contrast, dimeric or multimeric PD-1 ligands/antibodies can activate PD-1 via a mechanism that does not rely on mechanical force. Nevertheless, we agree with the reviewer that other mechanism can not be ruled out. So we have softened our statement in the revised manuscript.

o The results also state: "To determine whether mechanical force alone is sufficient to trigger PD-1 signaling, we stimulated Jurkat cell with CO- α -hPD1 beads (streptavidin beads simultaneously ligating human PD-1 antibody and CD3 antibody) in the presence of CD28 antibody", but it is unclear how this isolates the effect of mechanical force alone, given that the ligand is still surface-immobilised and aggregation/kinetic segregation can still occur.

Response: We sincerely appreciate the reviewer for highlighting this important concern. In our experiments, we utilized PD-1 antibodies that can block PD-L1 binding to PD-1 to serve as a functional mimic of PD-L1. These antibodies exhibit stronger binding affinity for PD-1 but do not form catch bonds, a characteristic that allowed us to partially assess the role of mechanical force in T cell signaling.

We fully agree with the reviewer that this experimental setup is not sufficient to definitively determine whether mechanical force alone can initiate PD-1 signaling, and we recognize the potential inaccuracy in such an interpretation. To address this, we have revised the original paragraph to clarify that these results support the importance of mechanical force in PD-1-mediated T cell regulation but not suggest that force alone is sufficient for complete PD-1 activation.

o Note that elongated PD-1 reduces its function in immobilised form yet may not have an impact on force. This was shown by two studies

<https://pubmed.ncbi.nlm.nih.gov/22641383/>

and <https://www.biorxiv.org/content/10.1101/2025.01.06.631424v1> (Fig 6)

Response: We sincerely appreciate the reviewer's constructive comment, which has prompted us to clarify our perspective on CD45 exclusion and the impact of PD-1

extracellular domain length on its function.

In our view, two key conditions should be met for PD-1 to effectively exclude CD45 and exert its inhibitory function: (1) the PD-1/PD-L1 interaction must have a sufficiently long lifetime to stabilize the signaling complex, and (2) the axial length of the PD-1/PD-L1 interaction must be shorter than that of the CD45 extracellular domain. If the PD-1/PD-L1 axis exceeds CD45's extracellular length, even a strong interaction between PD-1 and PD-L1 would fail to exclude CD45, as CD45 could still access and dephosphorylate nearby signaling molecules.

Regarding the effect of lengthening PD-1's extracellular domain on force transmission, force on PD-1 might remain unaffected in isolated PD-1/PD-L1 interactions. However, in the physiological context—where T cell-target cell interactions involve multiple molecular pairs (e.g., TCR/pMHC, CD28/CD80)—an excessively long extracellular domain would likely reduce PD-1's ability to sense and transmit relevant mechanical forces. Critically, PD-1's inhibitory function relies on its proximity to CD28 and TCR, as SHP-2 (recruited by activated PD-1) must access these targets to dephosphorylate them. Thus, extending the extracellular domain could either directly reduce force on PD-1 or, even if force is unaffected, separate PD-1 from CD28/TCR—both scenarios impairing its inhibitory capacity.

o When beads are used to present PD1 ligands (Figure 4D/4E/5A), it is also unclear whether these are intended to mimic soluble or surface-immobilised ligand.

Response: In these experiments, beads were employed to mimic surface-immobilized ligands, recapitulating the physiological context of ligand presentation on target cells. In Figure 5A specifically, we included both immobilized and soluble agents to dissect their distinct roles. Immobilized PD-L1 (on beads) was designed to engage PD-1 and trigger its inhibitory signaling, thereby suppressing IL-2 secretion in PD-1-expressing Jurkat cells. Soluble PD-L1 or anti-PD-1 antibodies were used as competitive antagonists: by binding PD-1 without initiating downstream inhibitory signaling (due to their soluble nature), they block the interaction between PD-1 and immobilized PD-L1. This competition relieves PD-1-mediated suppression, rescuing IL-2 secretion in the Jurkat cells.

• While the force-lifetime curves displayed (Figures 1-3) are representative of catch bonds, the molecular mechanisms of the two-state model of catch bond formation could be clarified. The second state must have a longer lifetime, thus increasing the observed lifetime as the bond shifts from state I to state II, and this is not made clear in the following sections:

o The simulations in figure 1 only display the angle and extension of the two states, but it is not clear what separates this observation from that of a slip bond dissociating over multiple states. It should be indicated that the second state has a lower energy in these simulations - e.g. how does the total number of salt bridges/H bonds compare, especially given specific interactions are then mutated in figure 2.

Response: We sincerely appreciate the reviewer's insightful suggestion, which has helped

refine our analysis of PD-1/PD-L1 conformational dynamics under force.

In our study, we defined distinct conformational states of the PD-1/PD-L1 complex using geometric parameters (angle and extension) derived from molecular dynamics (MD) simulations. Our results show that mechanical force induces a transition from a V-shaped structure of PD-1/PD-L1 complex (state I) to an extended conformation (state II), with each state exhibiting unique interaction modes that imply different mechanical stability.

In constant-force steered MD (SMD) simulations, state II exhibited longer lifetimes than state I under ~100 pN force (Fig. EV1 D-G), but the lack of statistical significance and the arbitrary nature of the applied force prevent us from definitively assigning superior mechanical stability to either state. Importantly, we propose that catch-bond formation does not depend solely on one state having a longer lifetime in a two-state model; instead, it arises from the combined effects of the distinct mechanical properties of each state and the occurrence probability of state II under force.

Following the reviewer's suggestion, we quantified hydrogen (H)-bond and salt bridge interactions between PD-1 and PD-L1 in both states (Figure R6). State II contains fewer H-bonds than state I, and while state I has two salt bridges (E136-R113/R125; Fig. 2 B-C), state II has only one (E61-K75; Fig. EV2F), indicating weakened polar interactions in state II. Conversely, hydrophobic interactions are stronger in state II (Fig. 2I), highlighting that mechanical stability depends on a balance of polar and hydrophobic forces.

Another critical consideration is the mechanism of force-induced dissociation: when force is applied to state I, the V-shaped complex undergoes stepwise unzipping, leaving some interactions intact during the transition. In contrast, disrupting the extended state II requires breaking all remaining interactions over a shorter timescale. This indicates absolute counts of H-bonds or salt bridges in isolated states cannot fully predict the mechanical stability of the complex, as the dynamics of dissociation (stepwise vs. concerted) also play a key role.

Figure R6. The number of H bonds between PD-1 and PD-L1 in cv-SMD simulations.

o In figure 2D, the residues mutated are suggested to disrupt only state I and not state II. Given that increasing the force would shift the bond to state II, why is the peak lifetime of the mutants at higher forces significantly lower than the wildtype? By decreasing the lifetime of state I, you should observe an enhanced catch bond due to a greater difference in lifetime between the two states.

Response: We sincerely appreciate the reviewer's astute observation regarding the

PD-L1 R113E/R125E mutant and its potential impact on conformational transitions. Our imaging and flow-cyto data clearly showed that R113E/R125E mutation impaired PD-L1 binding to PD-1 (Figure 4 F-G and Figure R7). Although state II may not be affected by the mutation, disruption of state I may also abrogate the conformation transition from state I to state II, and result in the reduced lifetime over the entire range of forces tested (2-20 pN).

Figure R7. Binding characteristics of wild type and mutated PD-L1, typical FACS histograms (A) and quantification (B) of PD-L1 binding were shown.

o The double mutant is stated to have rescued its catch bond in figure 2D, but this is not clear from the results, as the total increase in lifetime with force is very similar to the single mutants. To prove that the double mutant has regained wildtype catch bonding, further tests such as survival frequencies and F1/2 calculations using force ramps could demonstrate this.

Response: We sincerely appreciate the reviewer's insightful comments, which prompted us to refine our analysis of bond lifetime differences between PD-1/PD-L1 single mutations and the swap double mutant. To better delineate these distinctions, we focused on quantifying bond lifetimes specifically within the 5–10 pN force range—where mechanical regulation of the interaction is prominent.

As shown in Figure R8, the results clearly demonstrate that the swap-double mutant exhibits significantly longer bond lifetimes than all single mutants within this 5–10 pN range.

Figure R8. Mean lifetime of indicated wildtype or mutated PD-1/PD-L1 interactions within the 5–10 pN.

o For all mutants assessed in figure 2, it is important to demonstrate that solution affinity is unchanged with SPR, and the mutations therefore only affect the mechanical response to force.

Response: We sincerely appreciate the reviewer's valuable suggestion, which prompted us to further characterize the binding affinities of PD-L1 mutants targeting distinct conformational states. The mutants in Figure 2 were designed to probe two key states: the initial bound conformation (state I) and the force-induced conformation (state II). As hypothesized, mutations specific to state II might not significantly alter solution-phase affinity, while those mutations affecting interactions existed in both states could impair binding.

To address this, we performed flow cytometry experiments using biotinylated PD-L1 mutants: these were incubated with PD-1-expressing Jurkat cells, stained with streptavidin-APC, and analyzed for binding (Figure R7). The results showed that the binding ability of the I54A mutant was similar to that of wild-type PD-L1, whereas the K75A mutant weakened the ability of PD-L1 to bind PD-1, and the mutant with R113E/R125E, which should be against state I, almost completely disrupted the binding of PD-L1 to PD-1. These findings correlate well with our BFP data, reinforcing that state-specific mutations differentially affect binding based on their role in stabilizing either the initial or force-induced conformation.

Minor Comments:

- In the abstract, the following phrase is used: "we discovered that interactions between PD-1 and PD-L1/PD-L2 exhibit catch-slip bond behavior under force". Please note that this finding has recently been published by Li et al., and therefore using the word 'discovered' may appear misleading.

Response: Thank the reviewer to mention this point, as it provides an opportunity to clarify the timeline of our work and its relationship to the study by Li et al.

Our study was conducted between 2014 and 2018, with the first draft completed in

2019; the current manuscript was finalized in 2023. This timeline explains the initial use of the term “discovered” in our text.

Notably, the origins of this project stemmed from a proposed collaboration between one of our corresponding authors (Dr. Jizhong Lou) and Dr. Cheng Zhu (corresponding author of Li et al.). The initial plan was for Dr. Lou’s group to focus on MD simulations and Dr. Zhu’s group to perform BFP experiments. Regrettably, due to miscommunication, both groups ultimately pursued the full scope of the work independently and decided to publish separately.

In May 2023, shortly before we planned to submit our manuscript, Dr. Lou met Dr. Zhu at a conference and learned that their manuscript was under review. As a result, we chose to postpone our submission until after Li et al.’s paper was published.

Nevertheless, claiming “discovered” in this work may not be proper, and we have revised this language in the revised manuscript.

• For the following sentence in the results, "We controlled the site density of PD-L1 on the beads to maintain an adhesion frequency below 20%, ensuring predominantly single-bond interactions between PD-L1 and PD-1 on the cell surface (Figure. S1A and S1B)":

o Could the authors please clarify why a frequency below 20% is associated with predominantly single-bond interactions, especially when the association rates without PDL1 range from ~2-7%

o We also note that Figure S1A displays a value of >20% which contradicts the statement in the main text

Response: In the classical single molecule force spectroscopy experiments, each contact between the bead and the cell is viewed as an independent event, the number of molecular pairs formed during each contact follows a Poisson distribution (Eq.2):

$$P\{X = k\} = \frac{\lambda^k}{k!} e^{-\lambda} \quad (k = 0, 1, 2, \dots) \quad \dots \dots \dots (2)$$

The probability distribution was used to calculate the likelihood of single-molecule interactions. When the adhesion rate (i.e., the frequency of detectable bead-cell interactions) is 20%, the probability that these interactions arise from a single molecular pair accounts for 89.25% of all observed events. Even at a slightly higher adhesion rate of 25%, the probability of single-molecule interactions remains high at 86.32%. Keeping the adhesion frequency low ensures that the vast majority of interactions reflect single-molecule behavior.

To reflect the fact of our study, we have revised our statement.

• For the following comment in the results, "While previous reports suggest that PD-L1 can bind CD80 on T cells (27), recent studies increasingly support PD-L1 primarily binding CD80 in cis on the same cell surface (28-30). Our data aligns with these findings", it is unclear how the data here aligns with these findings, and it has not been demonstrated that PD-L1 in cis or trans impacts results.

Response: We want to determine the interaction of PD-L1 with PD-1 expressed on Jurkat cells, however Jurkat cells also express CD80. If PD-L1 can interact with CD80 *in trans*, then it would affect our results. For our control experiment (Figure EV1B), we showed that the adhesion rate of Jurkat-hPD-L1 was less than 5%, thus suggesting that PD-L1 and CD80 could not interact *in trans*.

- For the following comment in the results, "Interestingly, removing force from state II snapshots reverted the conformation back to state I (Figure. S1F, S1G), suggesting reversible conformational changes between states depending on force application. This switching likely enhances the stability of PD-1/PD-L1 interactions under low force, resulting in catch bonds", it is unclear from this explanation why enhanced stability of state I interactions under low force results in catch bonds.

Response: Thank for pointing this out. This is a typo, should be "enhances interactions under high force", we have corrected this typo in the revised manuscript.

- In figure S3, it is not clear how the BFP assay has been altered to allow PD1 to be recombinantly expressed, as opposed to presented natively on T cells. In addition, the main text explains that PD-L1 was expressed by *E. Coli*, whereas the caption of Figure S3 states that all proteins were expressed by 293F cells.

Response: We apologize for the lack of clarity that led to the reviewer's misunderstanding. PD-1 was over-expressed on Jurkat cells by lentivirus infection, and only the PD-L1 in Figure S3 was purified from 293F, all other PD-L1s were purified from *E. coli*.

- It would be helpful to see a figure depicting the TGT assay in figure 4

Response: We have added a supporting figure (Figure EV5) to depicting the TGT assay.

- In figure 5C, could the authors provide multiple examples of cell-cell conjugates, as providing a single example may not be representative of the effect of S-hPDL1. No uncropped images have been provided in the supplementary figures.

Response: We have added a few representative images and calculated the ratio of PD-1 in the IS region to the peripheral region by Image J (Figure R9A and Figure 5C). The results showed that both the PD-1 antibody (α -PD1) and PD-L1 (S-PDL1) were effective in decreasing the aggregation of PD-1 in the IS (Figure R9B and Figure EV7C).

Figure R9. Assessment of Jurkat (hPD1+)-Raji (hPD-L1+) conjugates in the presence of S-hPDL1 or α-hPD1. Representative confocal images (A) and quantification (B) of PD-1 engagement on cell-cell conjugates. Mock: n=30, αPD-1: n=22, S-PDL1: n=29, Scale bar: 10 μm.

We have added the uncropped images of original Figure 5C here as a supplementary figure (Figure R10 and Figure EV7).

Figure R10. Representative confocal image of Jurkat (hPD1+)-Raji (WT or hPD-L1+) conjugates. Cells were imaged to visualize the interaction between Jurkat cells expressing human PD-1 (green) and Raji cells expressing PD-L1 (red).

- In the introduction and discussion, it is stated that the TCR utilises mechanical force to enhance discrimination and that TCRs form catch bonds with antigens:
 - o "It has been demonstrated that TCR functions of distinguish non-self from self antigens

with the aid of mechanical force (21, 22)." - Introduction

o "Mechanical force plays a vital role in T cell activation, with the TCR functioning as a mechano-sensor by forming catch bonds with non-self antigens and slip bonds with self antigens (21, 22, 41)." - Discussion

There is evidence against this model that the authors should acknowledge.

Response: We fully respect the reviewer's opinion. For TCR discriminating antigen, there are quite a few different biological models or hypotheses, and these models have contributed great to revealing the mechanism of TCR triggering. However, this paper mainly focused on the mechano-regulation mechanism of PD-1/PD-L signaling, so we still want to keep the scope of the discussion.

- In the discussion it is stated "These results are in good agreement with our findings" with respect to the recent Li et al. paper. Although this is true for the results of mouse PD1-PDL1, it should be noted that the results for human PD-L1 vs PD-L2 display the opposite trend to the results of mouse PD-L1 vs PD-L2 in the Li et al paper, where it is shown PD-L2 forms a stronger catch bond with PD1 than PD-L1.

Response: We have added discussion on the results of mouse PD-L1 vs PD-L2 in the revised manuscript (*Line 341-352*).

Referee #3:

The paper by Chen et al. is a tour-de-force characterization of PD-1 interactions with PD-L1 and PD-L2. The wealth of data provided here is impressive, and contains numerous potentially highly interesting insights. Still, I think it would be important for the authors to clarify the following points:

1.) The authors are apparently aware of a similar article by Li et al (<https://doi.org/10.1038/s41467-024-52565-2>), which is also cited here as reference 42. However, the reference to Li et al. is only given in the discussion section of this paper. Given that the catch bond nature of PD-1 with PD-L1 and with PD-L2 has also been found in Li et al., the novelty of large parts of Fig.1, 2, and 3 is compromised. This should be stated openly. Results that are of confirmatory nature should be presented as such.

Response: We thank the reviewer for this comment. As mentioned in reply to one of reviewer 2's comment, our work was carried out between 2014-2018, and current manuscript is finalized in 2023. We know the manuscript of Li et al from the very beginning, wait and submit our manuscript after their manuscript is published. We are sorry that we only added the citation of their article in the discussion section of our manuscript before our submission.

In this revised manuscript, we have added more discussions of the article by Li et al (*Line 331-334*). From a publication perspective, Li *et.al.* proposed for the first time a catch-bond behavior for the interaction of PD-1 and PD-L in the mouse-derived system, and provided a detailed study of the molecular mechanism of the interaction between PD-1 and PD-L2 to form a catch-bond.

Our work, by contrast, focuses primarily on the human PD-1/PD-L1 system, with a specific emphasis on elucidating the catch-bond mechanism underlying their interaction. Additionally, we include comparative analyses of mouse PD-1/PD-L interactions, which reveal striking species-specific differences. For example, in the human system, PD-L1 binds PD-1 with higher affinity than PD-L2, whereas the opposite is true in the mouse system (Figure 1B-C, Figure R11). These distinctions highlight the importance of validating mechanistic findings across species, particularly for therapeutic development, where translational relevance to human biology is critical.

Figure R11. Mean bond lifetime dependence on mechanical force of human and mouse

PD-1/PD-L2 interactions, data are shown as Mean \pm SEM.

2.) I didn't fully understand how to interpret Fig. 4A. Using TGTs with characteristic rupture force of 12pN don't allow for PD-L1 mediated signal suppression, but at 23pN signals do get suppressed. But isn't 12pN already in the slip bond regime of the PD-1/PD-L1 interaction? In other words, the longest bond duration should occur if cells pulled at the peak of the curve shown in Fig. 1B, around 7pN. In contrast 12pN and 23pN would give fairly similar lifetimes.

Response: Thank the reviewer for this important comment. Reviewer 1 also raised a similar comment (Major comment 3 of Reviewer 1). The 12pN, 23pN notation of TGT are based on their rupture force measured with magnetic tweezers at room temperature in PBS buffer. It is known that temperature, salt concentration (especially Mg²⁺) and force loading rate are all contributing to the rupture force of DNA TGTs. In our study, we ligated protein/antibody with SA beads by biotin-SA interaction, and then incubated these beads with Jurkat cells in DMEM medium at 37 °C for 24 hours. That is, the temperature used in our experiments is much higher than that calibrating the rupture force of DNA TGTs in magnetic tweezers experiment. Higher temperature will destabilize DNA TGTs. Thus, in our study, the actual force required to rupture TGTs will be overestimated. Thus, the rupture force of the "12pN" TGT in our study could be much lower than 12pN.

Dear Prof. Lou,

Thank you for the submission of your revised manuscript to our editorial offices. I have now received the reports from the three referees that I asked to re-evaluate the study, you will find below. As you will see, the referees now feel that in the revised manuscript most referee points from the previous round have been addressed. However, all three referees have remaining concerns, further comments and suggestions to improve the manuscript, I ask you to address in a final revised manuscript as indicated by the reports (see below). Please also provide a detailed final p-b-p-response to these points and my editorial requests below.

Editorial requests:

- Please provide the abstract in present tense throughout.
- Please have your final manuscript text and the figure legends carefully proofread by a native speaker.
- Please provide individual production quality figure files as .eps, .tif, .jpg (one file per figure), of main figures and EV figures. Please upload these as separate, individual files upon re-submission. Presently, several main figures have few panels. Thus, please re-arrange the figures in a way that we have 6 final main figures and not more than 6 final EV figures. Please adjust all the callouts. Please use as callouts "Figure X" or "Fig. X" (not "Figure. X") for main figures and "Figure EVx" or Fig. EVx" for EV figures.
- Table EV1 needs also to be uploaded as a separate file (file type: Expanded View) and its legend needs to be removed from the main manuscript text file. Please add a callout for this table to the main manuscript text.
- Please also make sure that each figure panel (main and EV figures) is called out separately and sequentially. Presently, there seem to be no separate callouts for panels 3F and 3G. Please check.
- Please check that the number "n" for how many independent experiments were performed, their nature (biological versus technical replicates), the bars and error bars (e.g. SEM, SD) and the test used to calculate p-values is indicated in the respective figure legends. Please also check that all the p-values are explained in the legend, that exact p-values are listed and that these fit to those shown in the figure. Please provide statistical testing where applicable. Please avoid the phrase 'independent experiment' but clearly state if these were biological or technical replicates. Please also indicate (e.g. with n.s.) if testing was performed, but the differences are not significant. In case n=2, please show the data as separate datapoints without error bars and statistics. See also:
<http://www.embopress.org/page/journal/14693178/authorguide#statisticalanalysis>

If $n < 5$, please show single datapoints for diagrams. Presently, it seems that no statistical testing was performed for most of the diagrams shown in the main or EV or Appendix figures. Please do that. Moreover, our data editors have these additional requests:

- Please note that the exact p values are not provided in the legend of figure 2C
- Please indicate the statistical test used for data analysis in the legends of figures 2B, C, H, I; 5F; EV2 A-D, F, G, I, J, K-L
- Please note that information related to n is missing in the legends of figures 1B, C; 2D, E, F, H, I, J, K, L; 3A, B; 4A, B, C, D, E, G, H, I; 5A, B, D, E; EV3 A, B; EV6 A, B, D, E, F
- Please note that the error bars are not defined in the legends of figures 1G, H, I, J; 2B, C, F, H, I, L; 3B, E, G; 4A, B, C, D, E, G, H, I; 5A, B, D, E, F; EV1 A, B; EV2 A-D, F, G, I, J, K-L; EV4 B, D; EV6 A, B, D, E, F, I
- Please add to each legend (main, EV figures and Appendix Figures, where applicable) a 'Data Information' section (or name the provided 'notes' section like this) explaining the statistics used or providing information regarding replicates and scales. See:

- Please order the manuscript sections like this, using (only) these names:
Title page - Abstract - Keywords - Introduction - Results - Discussion - Methods - Data availability section - Acknowledgements - Disclosure and Competing Interests Statement - References - Figure legends - Expanded View Figure legends.
- Please move all the funding information to the acknowledgements.
- The data availability section (DAS) is restricted to externally deposited large datasets generated in a study. If no primary datasets have been deposited, please state this in this section ("No large primary datasets have been generated and deposited for this study"). Please remove all other information from this section.
- Please remove the section 'Supporting Information' from the manuscript.

- We updated our journal's competing interests policy in January 2022 and request authors to consider both actual and perceived competing interests. Please review the policy <https://www.embopress.org/competing-interests> and update your competing interests if necessary. Please name this section 'Disclosure and Competing Interests Statement' and put it after the Acknowledgements section.

- We now use CRediT to specify the contributions of each author in the journal submission system. CRediT replaces the author contribution section. Please use the free text box to provide more detailed descriptions and do NOT provide your final manuscript text file with an author contributions section. See also our guide to authors: <https://www.embopress.org/page/journal/14693178/authorguide#authorshipguidelines>

- All Materials and Methods need to be described in the main text using our 'Structured Methods' format, which is required for all research articles. According to this format, the Methods section should include a Reagents and Tools Table (listing key reagents, experimental models, software, and relevant equipment and including their sources and relevant identifiers), uploaded as separate file, and a Methods section in which we encourage the authors to describe their methods using a step-by-step protocol format with bullet points, to facilitate the adoption of the methodologies across labs. More information on how to adhere to this format as well as downloadable templates (.doc) for the Reagents and Tools Table can be found in our author guidelines (section 'Structured Methods');

- Thanks for providing the requested source data. However, please upload the source data as one folder per main figure, grouping together all the files for this figure (and ZIPed together). Please also add any source data for new main figure panels (see above). Please provide the numerical data as excel files that most interested readers can open (not as .pzfx). Moreover, it seems source data for panel 2K is missing. Please check.

Moreover, please note that corresponding authors are required to supply an ORCID ID upon submission of a revised manuscript and an institutional e-mail address. Please do this for co-corresponding author Changjie Lou. Please find instructions on how to link the ORCID ID to the account in our manuscript tracking system in our Author guidelines: <http://www.embopress.org/page/journal/14693178/authorguide#authorshipguidelines>

In addition, I would need from you uploaded separately:

Please let me know if you have questions regarding the revision.

Yours sincerely,

Referee #1:

I thank the authors for addressing my previous concerns. I am largely satisfied with the revisions, but three points remain:

First, regarding the discrepancy in optimal forces between Figures 1 and 4: the authors suggest that differences in experimental conditions-specifically, room temperature for Figure 1 versus 37 {degree sign}C for Figure 4-may explain the variation. This reminded me that the pN values in Figure 4 were originally determined by the Ha group at room temperature, and such rupture

forces can differ substantially at physiological temperature. It would be very helpful if the authors could directly measure the rupture forces of the DNA probes at 37 {degree sign}C, to better relate the functional results with the TGT data.

Second, I appreciate the inclusion of new cell imaging data comparing PD-L1 catch-bond mutants (Figure 4F). However, the observed effects may reflect differences in PD-1 binding affinities under no-force conditions. In Figure 2, at the lowest force (~2 pN), the I54A, Q66A, and K75A mutants exhibit shorter bond lifetimes than PD-L1 WT, suggesting weaker baseline affinities. Have the authors measured the binding affinities of these soluble PD-L1 mutants to PD-1?

Finally, the file titled "Revision_with_tracked_changes" does not appear to contain visible tracked changes, making it difficult to identify where revisions were made.

Referee #2:

We acknowledge that the authors have put effort into revising the manuscript as a response to our previous comments, however we believe there are several critical points that they have not appropriately addressed. These are listed below as responses to the author's comments:

1) Original comment: While the BFP assays throughout this paper (figures 1-3) clearly indicate catch bond formation, this system has previously been shown to display memory when the same T cell is repeatedly stimulated (<https://www.pnas.org/doi/10.1073/pnas.0704811104>). Could the authors clarify whether the same T cell was re-used over multiple forces, and if so, what order the forces were applied? In other words, are the lifetimes displayed throughout the manuscript truly independent measures? To conclude robustly that force increases bond lifetime, the same results should be achieved when T cells are not re-used, or when lifetimes are measured in response to forces in random order using the same T cell.

Authors' response: We thank the reviewer to pointing this issue out. We totally agree that this is a very important issue, also tried to avoid the potential memory affect. We are sorry that we did not include methods' detail in the original manuscript. In our BFP studies, the T cells are reused over various forces, also, lifetime for each force level are from various T-cells. For each T-cell used, the magnitude of the applied force order is manually chosen from three scenario: a low-high-low sequence, a high-low-high sequence, or a random order. Also, the T cell will be replaced as soon as it begins to change shape. We did not perform the experiment with T cells not re-used because it is really time consuming.

Reviewer's response: While they authors imply that this information is now available in their revised manuscript, we could not identify it in the main text or methods. They have not explained in their manuscript that lifetime measurements are not independent measures and that T cells are re-used, and they have not provided information on how they re-use T cells in their manuscript.

Since the lifetimes shown throughout the manuscript are not all independent (i.e. they rely on re-using the same T cell), we are concerned that statistics were performed assuming each measurement is independent. The authors need to provide the lifetime measurements they have performed explicitly identifying which came from each T cell and the sequence of force used. They need to then perform statistical analysis to show that the lifetime is not dependent on the order the force was applied.

Without this information, it can be argued that catch bonds may simply be the result of repeated measurements using the same T cell for example.

2) Original comment: Throughout this paper, caution should be taken when concluding whether force is required for productive signalling or is simply observed across a bond when two cells interact. This is clearest in the TGT-based assays, where the authors state in the discussion: "TGT-based T cell co-stimulation experiments further demonstrate that mechanical loading on the PD-1/PD-L1 interaction is essential for PD-1 function". The TGT results shown in figure 4 use probes which compromise the function PD-L1 beyond discrete force intervals. Because of this, it cannot be concluded that mechanical force is required for PD1 activation, only that it is exerted on the PD1-PDL1 bond. If the force had no functional effect on PD1 activation, the same results would be observed. To show that force is required for signalling, you would need a native PD1-PDL1 interaction without force and show that this prevents PD-1 signalling/inhibition.

Authors' response: We thank the reviewer for the insightful comment. It is true that catch-bond behavior does not mean the across membrane signaling is force dependent. Thus, we didn't draw conclusions based solely on BFP or TGT data. To test the correlation between force and signaling, we compared assays in which immobilized PD-L1s (where force can be exerted) and soluble PD-L1s (where force cannot be generated) were used (Figure 4 and Figure 5A). The study with soluble PD-L1s (without force) do show that it prevents PD-1's singling (loss of inhibition effect) (Figure 4B). Moreover, soluble PD-L1 can compete with immobilized PD-L1 and the inhibition effect by immobilized PD-L1 can be diminished as the concentration of soluble PD-L1 was increased (Figure 5A). From these data, we conclude that the signaling of PD-1 is force dependent.

Reviewer's response: The authors suggest that force is required because soluble PD-L1 cannot activate. While we agree that soluble PD-L1 is unlikely to be able to induce forces, it also cannot segregate phosphatases such as CD45. In other words, explanations that do not require force can also explain the authors results. As a result, the authors cannot claim that force is required because a soluble molecule cannot signal. The authors should revise their manuscript throughout to explain that the observation that soluble PD-L1 cannot signal is consistent with their force model but it does not prove it.

3) Original comment: There is not much emphasis on why force might be important for PD1 activation. In the case of TCR-pMHC interactions, catch bonds have been proposed to increase the ability of T cells to discriminate antigens based on affinity/lifetime. In the discussion (page 13), two possible mechanisms of PD-1 activation are proposed by the authors, however neither of these discussed in sufficient detail prior to this suggestion.

The first proposal (CD45 exclusion) bears no relevance to this paper, and is instead a reiteration of existing literature. Care should be taken when referring to these hypotheses as the authors' own proposals.

Authors' response: We thank the reviewer for this insightful comment. In our opinion, PD-1/PD-L1 interaction occur in the cell-cell interface where force can be generated from multiple origins. In this mechanical environment, catch-bond may help PD-1 to transmit signal more efficiently (by using longer time), and make PD-1 stay longer within the vicinity of TCR, CD28 and maybe other co-activating molecules. Then SHP-2 molecule which PD-1 recruited can also take affect longer. We believe further in-depth investigations are required to dissect why force might be important for PD-1 activation.

Reviewer's response: It's important to note that this can be achieved by simply having a higher affinity. It doesn't necessarily answer why a catch bond in particular is required.

4) Original comment: While the force-lifetime curves displayed (Figures 1-3) are representative of catch bonds, the molecular mechanisms of the two-state model of catch bond formation could be clarified. The second state must have a longer lifetime, thus increasing the observed lifetime as the bond shifts from state I to state II, and this is not made clear in the following sections:

- The simulations in figure 1 only display the angle and extension of the two states, but it is not clear what separates this observation from that of a slip bond dissociating over multiple states. It should be indicated that the second state has a lower energy in these simulations - e.g. how does the total number of salt bridges/H bonds compare, especially given specific interactions are then mutated in figure 2.

Authors' response: We sincerely appreciate the reviewer's insightful suggestion, which has helped refine our analysis of PD-1/PD-L1 conformational dynamics under force. In our study, we defined distinct conformational states of the PD-1/PD-L1 complex using geometric parameters (angle and extension) derived from molecular dynamics (MD) simulations. Our results show that mechanical force induces a transition from a V-shaped structure of PD-1/PD-L1 complex (state I) to an extended conformation (state II), with each state exhibiting unique interaction modes that imply different mechanical stability. In constant-force steered MD (SMD) simulations, state II exhibited longer lifetimes than state I under ~100 pN force (Fig. EV1 D-G), but the lack of statistical significance and the arbitrary nature of the applied force prevent us from definitively assigning superior mechanical stability to either state. Importantly, we propose that catch-bond formation does not depend solely on one state having a longer lifetime in a two-state model; instead, it arises from the combined effects of the distinct mechanical properties of each state and the occurrence probability of state II under force.

Reviewer's response: This appears to be a deviation from the previously defined two-state catch bond model (see Chakrabarti et al <https://doi.org/10.1016/j.jsb.2016.03.022>), in which catch bond properties occur from the transition between a slip bond of shorter lifetime to one of longer lifetime under force (thus, state II must always be of longer lifetime). If the authors are assuming a separate model, then this needs to be explained to clarify how these mutations are responsible for altering catch bonds.

5) Original comment: In figure 2D, the residues mutated are suggested to disrupt only state I and not state II. Given that increasing the force would shift the bond to state II, why is the peak lifetime of the mutants at higher forces significantly lower than the wildtype? By decreasing the lifetime of state I, you should observe an enhanced catch bond due to a greater difference in lifetime between the two states.

Authors' response: We sincerely appreciate the reviewer's astute observation regarding the PD-L1 R113E/R125E mutant and its potential impact on conformational transitions. Our imaging and flow-cyto data clearly showed that R113E/R125E mutation impaired PD-L1 binding to PD-1 (Figure 4 F-G and Figure R7). Although state II may not be affected by the mutation, disruption of state I may also abrogate the conformation transition from state I to state II, and result in the reduced lifetime over the entire range of forces tested (2-20 pN).

Reviewer's response: As per the previous response, this does not fully explain why a mutation to state I can disrupt the catch bond unless another model is being used. In theory, every mutation that reduces solution affinity could disrupt the transition between states, but this does not mean that those residues are involved in catch bond formation. Lowering the state I affinity may simply prevent state II from occurring under force because bond dissociation happens before this transition.

6) Original comment: For all mutants assessed in figure 2, it is important to demonstrate that solution affinity is unchanged with

SPR, and the mutations therefore only affect the mechanical response to force.

Authors' response: We sincerely appreciate the reviewer's valuable suggestion, which prompted us to further characterize the binding affinities of PD-L1 mutants targeting distinct conformational states. The mutants in Figure 2 were designed to probe two key states: the initial bound conformation (state I) and the force-induced conformation (state II). As hypothesized, mutations specific to state II might not significantly alter solution-phase affinity, while those mutations affecting interactions existed in both states could impair binding. To address this, we performed flow cytometry experiments using biotinylated PD-L1 mutants: these were incubated with PD-1-expressing Jurkat cells, stained with streptavidin-APC, and analyzed for binding (Figure R7). The results showed that the binding ability of the I54A mutant was similar to that of wild-type PD-L1, whereas the K75A mutant weakened the ability of PD-L1 to bind PD-1, and the mutant with R113E/R125E, which should be against state I, almost completely disrupted the binding of PD-L1 to PD-1. These findings correlate well with our BFP data, reinforcing that state-specific mutations differentially affect binding based on their role in stabilizing either the initial or force-induced conformation.

Reviewer's response: This data supports the authors' explanation that the R12E mutations disrupt state I (zero-force) binding compared to wild type. Of the two remaining mutants which are suggested to only disrupt state II binding, K75A is clearly also lower affinity under zero force and thus state I must also be affected - this should be explained in the text.

Importantly, measuring binding at a single concentration cannot unfortunately be used to determine an affinity. For example, if the concentration of PD-L1 used is higher than the K_D for both mutants, it will show identical binding even though the K_D is very different. To determine an affinity, the EC_{50} needs to be determined from a titration of PD-L1 concentrations.

7) Original comment: In the introduction and discussion, it is stated that the TCR utilises mechanical force to enhance discrimination and that TCRs form catch bonds with antigens:

- "It has been demonstrated that TCR functions of distinguish non-self from self antigens with the aid of mechanical force (21, 22)." - Introduction
 - "Mechanical force plays a vital role in T cell activation, with the TCR functioning as a mechano-sensor by forming catch bonds with non-self antigens and slip bonds with self antigens (21, 22, 41)." - Discussion
- There is evidence against this model that the authors should acknowledge.

Authors' response: We fully respect the reviewer's opinion. For TCR discriminating antigen, there are quite a few different biological models or hypotheses, and these models have contributed great to revealing the mechanism of TCR triggering. However, this paper mainly focused on the mechano-regulation mechanism of PD-1/PD-L signaling, so we still want to keep the scope of the discussion.

Reviewer's response: While we understand that the manuscript is focused on PD-1, the manuscript text at the moment is misleading the reader to think that it is scientific consensus that the TCR is a mechano-sensor that displays catch bonds. This is not the case and the authors should either remove the sentences concerning the TCR to avoid misleading the reader or at the very least explain to the reader that there is a body of evidence showing that the TCR forms slip bonds and that force on the TCR is not required for T cell activation. Here are some papers that support these other models:

<https://www.nature.com/articles/s41467-021-22775-z>

<https://www.pnas.org/doi/10.1073/pnas.1902141116>

<https://www.biorxiv.org/content/10.1101/2024.12.18.629139v1>

<https://www.embopress.org/doi/full/10.15252/emboj.2022111841>

[https://www.cell.com/biophysj/fulltext/S0006-3495\(11\)05402-6](https://www.cell.com/biophysj/fulltext/S0006-3495(11)05402-6)

Collectively, these studies are supporting a model whereby adhesion receptors can shield the TCR and other surface receptors from molecular forces. This force-shielding model was initially proposed by Pettmann et al (2023) EMBO Journal and has been discussed in the following reviews:

<https://pubmed.ncbi.nlm.nih.gov/40312550/>

<https://pubmed.ncbi.nlm.nih.gov/38794795/>

<https://pubmed.ncbi.nlm.nih.gov/36808636/>

Referee #3:

I appreciate the new discussion on the paper by Li et al. Still, I would recommend adding also a sentence in the introduction, and clearly stating the novelty of the current approach with respect to the previous one.

In addition, I don't feel that the answer to my previous second question is sufficient. If the force regimes of the used TGTs is unknown, what is the reason of using the TGTs at all? Further, if the argumentation of the authors is correct, terming the TGTs 12pN/23pN/56pN is misleading and should be changed.

We thank the reviewers for their additional comments and suggestions, which have further enhanced the quality of our work. We have thoroughly addressed their concerns to substantiate our findings. Below are our point-by-point responses to the referees' comments. The original comments are presented in black, and our replies are in blue.

Referee #1:

I thank the authors for addressing my previous concerns. I am largely satisfied with the revisions, but three points remain:

First, regarding the discrepancy in optimal forces between Figures 1 and 4: the authors suggest that differences in experimental conditions—specifically, room temperature for Figure 1 versus 37 {degree sign}C for Figure 4—may explain the variation. This reminded me that the pN values in Figure 4 were originally determined by the Ha group at room temperature, and such rupture forces can differ substantially at physiological temperature. It would be very helpful if the authors could directly measure the rupture forces of the DNA probes at 37 {degree sign}C, to better relate the functional results with the TGT data.

Response: We really appreciate reviewer's suggestion to directly measure the rupture forces of DNA probes at physiological-related condition, which would strengthen the mechanistic link between our single molecule force data and the TGT-based functional

results. However, measuring DNA unzipping forces under physiological conditions presents certain technical challenges. Moreover, as mentioned in our previously response, temperature, ion (especially Mg²⁺) concentration, loading rate are all contributing to the exact level of the rupture force.

Previously, Prentiss et.al. demonstrated that DNA unzipping forces decrease substantially above 35 degree (10.1103/PhysRevLett.96.248101, 10.1103/PhysRevLett.93.078101), with deviations from the theoretical prediction by Peyrard-Bishop-Dauxois model. Additional work has further established that such forces are exquisitely sensitive to experimental variables, including Mg²⁺ concentration (10.1038/ncomms11026) and loading rate (10.1073ypnas.151257598) or both (10.1038/s41467-022-34212-w).

In our specific setup, precise temperature control in the BFP instrument remains technically challenging, and the dynamic loading rate exerted on cells during TGT experiments cannot be precisely quantified. These limitations collectively make accurate measurement of DNA unzipping forces under cell culture conditions unfeasible within the current timeline.

We recognize the value of this measurement for future work and hope to address it as technical capabilities improve. Thank you again for emphasizing this important experimental connection, which underscores the need for rigorous condition-matching in single-molecule and functional assays.

Second, I appreciate the inclusion of new cell imaging data comparing PD-L1 catch-bond

mutants (Figure 4F). However, the observed effects may reflect differences in PD-1 binding affinities under no-force conditions. In Figure 2, at the lowest force (~2 pN), the I54A, Q66A, and K75A mutants exhibit shorter bond lifetimes than PD-L1 WT, suggesting weaker baseline affinities. Have the authors measured the binding affinities of these soluble PD-L1 mutants to PD-1?

Response: We sincerely appreciate the reviewer's insightful observation, which prompted us to directly assess whether the functional effects of PD-L1 mutants (Figure 4F) stem from altered differences in baseline (no-force) binding affinity to PD-1.

To address this, we performed flow cytometry experiments comparing the binding of soluble PD-L1 mutants (I54A, Q66A, K75A) to PD-1-expressing Jurkat cells across a concentration range (0.1–5 μ M) (Figure R1A). The results confirm the reviewer's speculation that some mutants exhibit altered baseline affinity. Among these mutants, K75A showed binding comparable to that of the wild-type at concentrations up to 5 μ M, whereas Q66A remains lower binding affinity. Molecular dynamics simulations revealed that, in the absence of force, Q66 directly interacts with A132 of PD-1, therefore, the mutation also destabilizes State 1 and result in shorter lifetime at low force (~2 pN). On the other hand, K75 forms a hydrogen bond network with D73 and Q66, which indirectly modulates the interaction between Q66 and A132 (Figure R1B). We did not observe significant changes in PD-1 binding for the I54A mutant compared to wild-type PD-L1 from 0.1 μ M to 5 μ M, which may due to the concentration of PD-L1 we used is still too high. Alternatively, the reduced bond lifetime in the low-force regime could also result from

weakened State 2 interactions by I54 mutation.

These findings clarify that the functional differences observed in Figure 4F arise from a combination of baseline affinity effects (Q66A) and force-specific interaction defects (I54A, K75A), reinforcing the context-dependent nature of these mutations' impacts.

Figure R1. PD-L1 mutations affect PD-1 binding.

A. PD-1 binding of PD-L1 mutations quantification by flow-cyto, biotinylated PD-L1 (wildtype and mutations) were incubated with PD-1⁺ Jurkat cells followed by SA-APC staining; B. Snap-shot of PD-1/PD-L1 complex from simulation.

Finally, the file titled "Revision_with_tracked_changes" does not appear to contain visible tracked changes, making it difficult to identify where revisions were made.

Response: We are sorry for this mistake. During the submission of the current revision, we have regenerated a new PDF file which include all the tracks.

Referee #2:

We acknowledge that the authors have put effort into revising the manuscript as a response to our previous comments, however we believe there are several critical points that they have not appropriately addressed. These are listed below as responses to the author's comments:

1) Original comment: While the BFP assays throughout this paper (figures 1-3) clearly indicate catch bond formation, this system has previously been shown to display memory when the same T cell is repeatedly stimulated (<https://www.pnas.org/doi/10.1073/pnas.0704811104>). Could the authors clarify whether the same T cell was re-used over multiple forces, and if so, what order the forces were applied? In other words, are the lifetimes displayed throughout the manuscript truly independent measures? To conclude robustly that force increases bond lifetime, the same results should be achieved when T cells are not re-used, or when lifetimes are measured in response to forces in random order using the same T cell.

Authors' response: We thank the reviewer to pointing this issue out. We totally agree that this is a very important issue, also tried to avoid the potential memory affect. We are sorry that we did not include methods' detail in the original manuscript. In our BFP studies, the T cells are reused over various forces, also, lifetime for each force level are from various T-cells. For each T-cell used, the magnitude of the applied force order is manually chosen

from three scenarios: a low-high-low sequence, a high-low-high sequence, or a random order. Also, the T cell will be replaced as soon as it begins to change shape. We did not perform the experiment with T cells not re-used because it is really time consuming.

Reviewer's response: While the authors imply that this information is now available in their revised manuscript, we could not identify it in the main text or methods. They have not explained in their manuscript that lifetime measurements are not independent measures and that T cells are re-used, and they have not provided information on how they re-use T cells in their manuscript.

Response: Thank the reviewer for the suggestion, we have added the information into the BFP method in our revised manuscript.

Since the lifetimes shown throughout the manuscript are not all independent (i.e. they rely on re-using the same T cell), we are concerned that statistics were performed assuming each measurement is independent. The authors need to provide the lifetime measurements they have performed explicitly identifying which came from each T cell and the sequence of force used. They need to then perform statistical analysis to show that the lifetime is not dependent on the order the force was applied.

Without this information, it can be argued that catch bonds may simply be the result of repeated measurements using the same T cell for example.

Response: We sincerely appreciate the reviewer's critical concern regarding the independence of lifetime measurements and potential order effects from reusing the same T cell. This is a crucial consideration for ensuring the robustness of single-molecule force spectroscopy data, and we have addressed it with detailed analyses as follows:

We structured our analysis into two distinct datasets:

Firstly, the Cross-sectional dataset, 18 cells in total, each tested exclusively within a narrow force range (within several pN, as indicated in **Figure R2A**) with no repeated force applications. This design avoids order effects entirely, as each cell contributes measurements to only one force regime. Raw lifetime measurements from these cells, labeled by their source cell, are presented as scatter plots in Figure R2A, with clear separation by cell identifier.

Secondly, the Longitudinal dataset, 4 cells in total, each subjected to complete force cycles. Individual measurements from these cells are plotted in **Figure R2B**, with color coding to indicate cell identity.

Notably, both datasets exhibited consistent catch bond behavior, with mean lifetime profiles peaking at intermediate forces (**Figure R2C**). This convergence confirms that the observed catch bond phenomenon is not an artifact of repeated measurements or force application order but reflects inherent mechanical properties of the PD-1/PD-L1 interaction.

Thank the reviewer again for emphasizing this important validation, which strengthens the statistical rigor of our findings.

Figure R2. Raw data from different strategies of cell using in BFP assay.

A and B. Scatter plot of different cells: Cross-sectional (A) and Longitudinal (B), C. The mean lifetime curves from Cross-sectional (A) and Longitudinal (B).

2) Original comment: Throughout this paper, caution should be taken when concluding whether force is required for productive signalling or is simply observed across a bond when two cells interact. This is clearest in the TGT-based assays, where the authors state in the discussion: "TGT-based T cell co-stimulation experiments further demonstrate that mechanical loading on the PD-1/PD-L1 interaction is essential for PD-1 function". The TGT results shown in figure 4 use probes which compromise the function PD-L1 beyond discrete force intervals. Because of this, it cannot be concluded that mechanical force is required for PD1 activation, only that it is exerted on the PD1-PDL1 bond. If the force had no functional effect on PD1 activation, the same results would be observed. To show that

force is required for signalling, you would need a native PD1-PDL1 interaction without force and show that this prevents PD-1 signalling/inhibition.

Authors' response: We thank the reviewer for the insightful comment. It is true that catch-bond behavior does not mean the across membrane signaling is force dependent. Thus, we didn't draw conclusions based solely on BFP or TGT data. To test the correlation between force and signaling, we compared assays in which immobilized PD-L1s (where force can be exerted) and soluble PD-L1s (where force cannot be generated) were used (Figure 4 and Figure 5A). The study with soluble PD-L1s (without force) do show that it prevents PD-1's signaling (loss of inhibition effect) (Figure 4B). Moreover, soluble PD-L1 can compete with immobilized PD-L1 and the inhibition effect by immobilized PD-L1 can be diminished as the concentration of soluble PD-L1 was increased (Figure 5A). From these data, we conclude that the signaling of PD-1 is force dependent.

Reviewer's response: The authors suggest that force is required because soluble PD-L1 cannot activate. While we agree that soluble PD-L1 is unlikely to be able to induce forces, it also cannot segregate phosphatases such as CD45. In other words, explanations that do not require force can also explain the authors results. As a result, the authors cannot claim that force is required because a soluble molecule cannot signal. The authors should revise their manuscript throughout to explain that the observation that soluble PD-L1 cannot signal is consistent with their force model but it does not prove it.

Response: We thank the reviewer's suggestion. As indicated in our previous response, both mechanical force and size are important for PD-1 function. In our opinion, the exclusion of CD45 by the PD-1/PD-L1 axis should be accompanied by an increase in tension (mechanical force) along the interaction interface, which prolongs the bond lifetime and ultimately leads to complete CD45 exclusion. Nevertheless, demonstrating this is really challenging, so we have toned down the emphasis on the role of force in PD-1 activation in our revised manuscript as suggested.

3) Original comment: There is not much emphasis on why force might be important for PD1 activation. In the case of TCR-pMHC interactions, catch bonds have been proposed to increase the ability of T cells to discriminate antigens based on affinity/lifetime. In the discussion (page 13), two possible mechanisms of PD-1 activation are proposed by the authors, however neither of these discussed in sufficient detail prior to this suggestion.

The first proposal (CD45 exclusion) bears no relevance to this paper, and is instead a reiteration of existing literature. Care should be taken when referring to these hypotheses as the authors' own proposals.

Authors' response: We thank the reviewer for this insightful comment. In our opinion, PD-1/PD-L1 interaction occur in the cell-cell interface where force can be generated from multiple origins. In this mechanical environment, catch-bond may help PD-1 to transmit signal more efficiently (by using longer time), and make PD-1 stay longer within the

vicinity of TCR, CD28 and maybe other co-activating molecules. Then SHP-2 molecule which PD-1 recruited can also take affect longer. We believe further in-depth investigations are required to dissect why force might be important for PD-1 activation.

Reviewer's response: It's important to note that this can be achieved by simply having a higher affinity. It doesn't necessarily answer why a catch bond in particular is required.

Response: We sincerely agree with the reviewer's critical point. High-affinity interactions alone indeed can drive PD-1 signaling, e.g., as seen with immobilized PD-1 antibodies.

Our manuscript focuses primarily on characterizing the catch bond behavior of PD-1/PD-L interactions and linking this mechano-property to function, but we appreciate the opportunity to elaborate on the distinct advantages of catch bonds beyond affinity. These advantages center on force-dependent, context-specific regulation of signaling - a capability high-affinity slip bonds lack:

Taking PD-1 as an example, high-affinity slip bonds can also lead to CD45 exclusion, PD-1 phosphorylation, and subsequent downstream signaling—which may explain why immobilized PD-1 antibodies are capable of activating PD-1 in vitro. Moreover, several studies have shown that replacing the extracellular domain of PD-1 with other domains and stimulating with the corresponding ligands can still mediate PD-1 activation (doi: 10.4049/jimmunol.173.2.945.).

However, for high-affinity slip bonds, dissociation typically requires application of greater force or more destructive mechanisms like trogocytosis. In contrast, catch bond

behavior aligns with the force-shielding model (Pettmann et. al., 2023, The EMBO Journal): once stable T-cell-target cell conjugates form, the force on individual receptor–ligand pairs (such as PD-1 with PD-Ls) decreases. This may trigger a decrease in catch bond lifetime, enabling regulated dissociation of the complex and precise termination of signaling—avoiding the collateral effects of forced dissociation with slip bonds.

Moreover, the catch-bond mechanism may effectively prevent unnecessary receptor activation. Under low-force conditions (e.g., incidental receptor-ligand encounters), high-affinity slip bonds exhibit long lifetimes (low k_{off} , **Figure R3**), promoting stable engagement and unintended activation. Catch bonds, by contrast, have shorter lifetimes (high k_{off} , Figure R3) at low forces, reducing the probability of non-specific signaling. Mechanical force thus acts as a “gatekeeper”: only when meaningful cell-cell contact generates sufficient force do catch bonds stabilize (prolonged lifetime) to initiate signaling.

Furthermore, cells can modulate the mechanical microenvironment experienced by receptors (such as PD-1) via cytoskeletal organization or the interaction of other receptor–ligand pairs. This allows dynamic control over receptor (such as PD-1) activation - a level of regulation unattainable with static high-affinity slip bonds, which lack force-dependent responsiveness.

Figure R3. Different lifetime of catch bond and slip bond at low-force regime.

4) Original comment: While the force-lifetime curves displayed (Figures 1-3) are representative of catch bonds, the molecular mechanisms of the two-state model of catch bond formation could be clarified. The second state must have a longer lifetime, thus increasing the observed lifetime as the bond shifts from state I to state II, and this is not made clear in the following sections:

- The simulations in figure 1 only display the angle and extension of the two states, but it is not clear what separates this observation from that of a slip bond dissociating over multiple states. It should be indicated that the second state has a lower energy in these simulations - e.g. how does the total number of salt bridges/H bonds compare, especially given specific interactions are then mutated in figure 2.

Authors' response: We sincerely appreciate the reviewer's insightful suggestion, which has helped refine our analysis of PD-1/PD-L1 conformational dynamics under force. In our study, we defined distinct conformational states of the PD-1/PD-L1 complex using geometric parameters (angle and extension) derived from molecular dynamics (MD) simulations. Our results show that mechanical force induces a transition from a V-shaped structure of PD-1/PD-L1 complex (state I) to an extended conformation (state II), with each state exhibiting unique interaction modes that imply different mechanical stability. In constant-force steered MD (SMD) simulations, state II exhibited longer lifetimes than state I under ~100 pN force (Fig. EV1 D-G), but the lack of statistical significance and the

arbitrary nature of the applied force prevent us from definitively assigning superior mechanical stability to either state. Importantly, we propose that catch-bond formation does not depend solely on one state having a longer lifetime in a two-state model; instead, it arises from the combined effects of the distinct mechanical properties of each state and the occurrence probability of state II under force.

Reviewer's response: This appears to be a deviation from the previously defined two-state catch bond model (see *Chakrabarti et al* <https://doi.org/10.1016/j.jsb.2016.03.022>), in which catch bond properties occur from the transition between a slip bond of shorter lifetime to one of longer lifetime under force (thus, state II must always be of longer lifetime). If the authors are assuming a separate model, then this needs to be explained to clarify how these mutations are responsible for altering catch bonds.

Response: Thank the reviewer for this point. We are sorry that we did not explain this clearly in our manuscript. We believe that the two-pathway catch bond model still hold in PD-1/PD-L1 interaction. But we should emphasize here that under force, the dissociation is also starting from state I, and go through state II.

Thus, the dissociation pathway without force includes only state I; while the dissociation pathway under force includes state I plus state II. Then, the state II found in our study is an intermediate state and is not required to be stronger than state I. What is required is that “state I plus state II” is stronger than “state I only”.

5) Original comment: In figure 2D, the residues mutated are suggested to disrupt only

state I and not state II. Given that increasing the force would shift the bond to state II, why is the peak lifetime of the mutants at higher forces significantly lower than the wildtype? By decreasing the lifetime of state I, you should observe an enhanced catch bond due to a greater difference in lifetime between the two states.

Authors' response: We sincerely appreciate the reviewer's astute observation regarding the PD-L1 R113E/R125E mutant and its potential impact on conformational transitions. Our imaging and flow-cyto data clearly showed that R113E/R125E mutation impaired PD-L1 binding to PD-1 (Figure 4 F-G and Figure R7). Although state II may not be affected by the mutation, disruption of state I may also abrogate the conformation transition from state I to state II, and result in the reduced lifetime over the entire range of forces tested (2-20 pN).

Reviewer's response: As per the previous response, this does not fully explain why a mutation to state I can disrupt the catch bond unless another model is being used. In theory, every mutation that reduces solution affinity could disrupt the transition between states, but this does not mean that those residues are involved in catch bond formation. Lowering the state I affinity may simply prevent state II from occurring under force because bond dissociation happens before this transition.

Response: We thank the reviewer for the comment. As mentioned in our previous response, without force, the dissociation pathway includes only state I; Under force

application, the dissociation pathway includes both state I and state II. Thus, mutation disrupting state I, state II or the transition of the two states will all affect catch bond behavior. We don't think lowering state I affinity will completely prevent state II from occurring, of course, when the affinity of state I is lowered, the occurring possibility of state II may be reduced due to the shortened time for the transition to occur.

6) Original comment: For all mutants assessed in figure 2, it is important to demonstrate that solution affinity is unchanged with SPR, and the mutations therefore only affect the mechanical response to force.

Authors' response: We sincerely appreciate the reviewer's valuable suggestion, which prompted us to further characterize the binding affinities of PD-L1 mutants targeting distinct conformational states. The mutants in Figure 2 were designed to probe two key states: the initial bound conformation (state I) and the force-induced conformation (state II). As hypothesized, mutations specific to state II might not significantly alter solution-phase affinity, while those mutations affecting interactions existed in both states could impair binding. To address this, we performed flow cytometry experiments using biotinylated PD-L1 mutants: these were incubated with PD-1-expressing Jurkat cells, stained with streptavidin-APC, and analyzed for binding (Figure R7). The results showed that the binding ability of the I54A mutant was similar to that of wild-type PD-L1, whereas the K75A mutant weakened the ability of PD-L1 to bind PD-1, and the mutant with R113E/R125E, which should be against state I, almost completely disrupted the binding of

PD-L1 to PD-1. These findings correlate well with our BFP data, reinforcing that state-specific mutations differentially affect binding based on their role in stabilizing either the initial or force-induced conformation.

Reviewer's response: This data supports the authors' explanation that the R12E mutations disrupt state I (zero-force) binding compared to wild type. Of the two remaining mutants which are suggested to only disrupt state II binding, K75A is clearly also lower affinity under zero force and thus state I must also be affected - this should be explained in the text.

Response: We sincerely appreciate the reviewer's perceptive observation. Analysis of the PD-1/PD-L1 crystal structure (PDB: 4ZQK) and our state I (zero-force) simulation snapshots confirms K75 in PD-L1 does not form direct contacts with PD-1 residues. However, trajectory analyses reveal K75 is part of a critical intramolecular hydrogen bond network within PD-L1 that indirectly stabilizes state I: K75 interacts with D73 (PD-L1), which in turn interacts with Q66 (PD-L1); Q66 then forms a direct hydrogen bond with the backbone atom of A132 (PD-1) - a key interaction for state I stability (Fig. S2 K-L, **Figure R4**). The K75A mutation weakens the K75-D73 interaction, destabilizing the D73-Q66 contact and ultimately reducing the stability of the Q66-A132 bond with PD-1. We have incorporated this mechanistic explanation into the revised manuscript (*Line 186-190*).

Figure R4. A snapshot of PD-1/PD-L1 complex from our MD simulation.

Importantly, measuring binding at a single concentration cannot unfortunately be used to determine an affinity. For example, if the concentration of PD-L1 used is higher than the K_D for both mutants, it will show identical binding even though the K_D is very different. To determine an affinity, the EC_{50} needs to be determined from a titration of PD-L1 concentrations.

Response: We sincerely appreciate the reviewer for this suggestion, we fully agree that binding measurements at a single concentration are insufficient to assess affinity.

To address this, we have additionally performed this binding assay again with serial PD-L1 concentration (Figure R5A). The results show that I54A mutant showed a comparable binding with wt PD-L1 at all tested concentrations (0.1 μM -5 μM), while R12E mutant exhibited severely impaired binding at all concentrations, consistent with its role in destabilizing state I. Interestingly, the Q66A and K75A mutants displayed moderate binding defects relative to wild-type, aligning with our prior analysis.

To further validate the I54A mutant's zero-force binding, we have also complemented

the titration data with BFP-based adhesion frequency measurements between PD-1⁺ Jurkat cells and PD-L1 (wt or I54A) across serial contact times (0.1 s - 2 s) (Figure R5B). No significant difference in binding frequency was observed between wild-type and I54A, reinforcing that I54A does not alter the intrinsic zero-force interaction with PD-1.

These titration and single-molecule binding frequency data collectively confirm the affinity profiles of the mutants, with I54A specifically lacking zero-force defects.

Figure R5. PD-L1 mutations affect PD-1 binding.

A. PD-1 binding of PD-L1 mutations quantification by flow-cyto, biotinylated PD-L1 (wildtype and mutations) were incubated with PD-1⁺ Jurkat cells followed by SA-APC staining; B. Adhesion frequency of I54A mutated PD-L1 interact with PD-1⁺ Jurkat cells, different contact time were use in this BFP assay.

7) Original comment: In the introduction and discussion, it is stated that the TCR utilises mechanical force to enhance discrimination and that TCRs form catch bonds with antigens:

- "It has been demonstrated that TCR functions of distinguish non-self from self antigens with the aid of mechanical force (21, 22)." - Introduction
- "Mechanical force plays a vital role in T cell activation, with the TCR functioning as a

mechano-sensor by forming catch bonds with non-self antigens and slip bonds with self antigens (21, 22, 41)." - Discussion

There is evidence against this model that the authors should acknowledge.

Authors' response: We fully respect the reviewer's opinion. For TCR discriminating antigen, there are quite a few different biological models or hypotheses, and these models have contributed great to revealing the mechanism of TCR triggering. However, this paper mainly focused on the mechano-regulation mechanism of PD-1/PD-L signaling, so we still want to keep the scope of the discussion.

Reviewer's response: While we understand that the manuscript is focused on PD-1, the manuscript text at the moment is misleading the reader to think that it is scientific consensus that the TCR is a mechano-sensor that displays catch bonds. This is not the case and the authors should either remove the sentences concerning the TCR to avoid misleading the reader or at the very least explain to the reader that there is a body of evidence showing that the TCR forms slip bonds and that force on the TCR is not required for T cell activation. Here are some papers that support these other models:

<https://www.nature.com/articles/s41467-021-22775-z>

<https://www.pnas.org/doi/10.1073/pnas.1902141116>

<https://www.biorxiv.org/content/10.1101/2024.12.18.629139v1>

<https://www.embopress.org/doi/full/10.15252/embj.2022111841>

[https://www.cell.com/biophysj/fulltext/S0006-3495\(11\)05402-6](https://www.cell.com/biophysj/fulltext/S0006-3495(11)05402-6)

Collectively, these studies are supporting a model whereby adhesion receptors can shield the TCR and other surface receptors from molecular forces. This force-shielding model was initially proposed by Pettmann et al (2023) EMBO Journal and has been discussed in the following reviews:

<https://pubmed.ncbi.nlm.nih.gov/40312550/>

<https://pubmed.ncbi.nlm.nih.gov/38794795/>

<https://pubmed.ncbi.nlm.nih.gov/36808636/>

Response: Thank the reviewer for the suggestion, we have removed these sentences concerning the TCR catch bond to avoid potential misleading.

Referee #3:

I appreciate the new discussion on the paper by Li et al. Still, I would recommend adding also a sentence in the introduction, and clearly stating the novelty of the current approach with respect to the previous one.

Response: We have added a sentence in the Introduction to explicitly contextualize our work relative to Li et al. and highlight its novelty (*Line 96-97*).

In addition, I don't feel that the answer to my previous second question is sufficient. If the force regimes of the used TGTs is unknown, what is the reason of using the TGTs at all? Further, if the argumentation of the authors is correct, terming the TGTs 12pN/23pN/56pN is misleading and should be changed.

Response: We sincerely appreciate the reviewer's constructive feedback, which has helped us strengthen the clarity of our manuscript's context and methods.

We acknowledge the reviewer's valid concern about the unknown absolute force regimes of TGTs under our experimental conditions and the potential confusion from numerical labeling.

The core value of the TGT system in our study lies not in delivering precise, absolute forces (e.g., 12/23/56 pN) but in creating a controlled gradient of mechanical stability thresholds that allow us to test a key hypothesis: PD-1 activation requires ligand binding

plus the ability to sustain mechanical tension. TGTs uniquely enable this by modulating the force tolerance of the PD-L1 anchor (via distinct DNA duplex designs) without altering PD-L1's intrinsic binding affinity for PD-1. This allows us to disentangle “binding alone” from “binding + mechanical stability” as drivers of PD-1 signaling—something unfeasible with soluble or uniformly immobilized ligands.

While the absolute force transmitted to PD-1/PD-L1 remains uncertain (due to variables like temperature and dynamic loading rates), the TGT gradient provides clear qualitative resolution: only PD-L1 anchored to TGTs with higher mechanical tolerance (previously labeled “23pN” and “56pN”) supported efficient PD-1 activation, whereas those with lower tolerance (“12pN”) or no anchor (soluble) did not. This confirms that mechanical stability—beyond binding—is required for signaling.

To address the misleading numerical labels, we have revised all TGT designations in the manuscript to qualitative terms: “Low Tension Tolerance,” “Medium Tension Tolerance,” and “High Tension Tolerance.” We also included a detailed description of the force calibration and measurement conditions in the Methods section. This will provide readers with clearer information for interpretation and reference.

Dear Prof. Lou,

Thank you for the submission of your further revised manuscript to our editorial offices. I have now received the reports from two of the three referees that I asked to re-evaluate the study, you will find below. As you will see, referees #1 and #3 now support the publication of your manuscript in EMBO reports, whereas referee #2 still has remaining concerns, or states that previous concerns have not been adequately addressed. During cross-commenting, referee #1 and #3 agreed with referee #2 and stated that these points need to be addressed, the paper revised as indicated and that the requested source data needs to be provided.

I thus invite you to revise your manuscript further with the understanding that the remaining concerns of the referee will be addressed in the revised manuscript as indicated above and in the report of referee #2. Please also provide a final detailed point-by-point response to all remaining referee points and the editorial requests below.

Editorial requests:

- Please use as callouts "Figure X" or "Fig. X" (not "Figure. X") for main figures and "Figure EVx" or "Fig. EVx" for EV figures. Please remove the point after the word 'Figure' in all callouts.
- Please add the schematic shown in Appendix Figure S1 to one of the main or EV figures. Then please update the callouts and remove Appendix Figure S1 from the manuscript files.
- Please also remove the pdf files with combined main and EV figures from the submission.
- We replaced Table EV1 with an Excel version. Please check that this is correct and that the added legend is fine. If not, please submit a final excel file for this table (with a legend on the first TAB).
- Please check again that the number "n" for how many independent experiments were performed, their nature (biological versus technical replicates), the bars and error bars (e.g. SEM, SD) and the test used to calculate p-values is indicated in the respective figure legends. Please also check that all the p-values are explained in the legend, and that these fit to those shown in the figure. Please provide statistical testing where applicable. Please avoid the phrase 'independent experiment' but clearly state if these were biological or technical replicates. Please also indicate (e.g. with n.s.) if testing was performed, but the differences are not significant. In case n=2, please show the data as separate datapoints without error bars and statistics. See also:
<http://www.embopress.org/page/journal/14693178/authorguide#statisticalanalysis>
- If n<5, please show single datapoints for diagrams. Presently some diagrams have no statistics (e.g. 5G, 5H left panel, 6G or EV5F) or miss the 'n.s.'. Please check panels 2F, 2L, 3B, 4A-D, 4E, 4I, 5A, 5B, 5E, EV4B, EV4D, EV4E, EV4F, EV4H, EV5A, EV6B and EV6C.
- Please add a 'Data Information' section to the legend of Fig. 1.
- Please remove the instruction text and the example table from the Reagents & Tools Table.
- Please have your final manuscript text file carefully proofread by a native speaker. There are still typos and errors present.

Yours sincerely,

Referee #1:

The authors have fully addressed my concerns.

Referee #2:

The authors have addressed only some of our remaining concerns.

Comment 1:

We explained the following in our response to the authors:

"Since the lifetimes shown throughout the manuscript are not all independent (i.e. they rely on re-using the same T cell), we are concerned that statistics were performed assuming each measurement is independent. The authors need to provide the lifetime measurements they have performed explicitly identifying which came from each T cell and the sequence of force used. They need to then perform statistical analysis to show that the lifetime is not dependent on the order the force was applied.

Without this information, it can be argued that catch bonds may simply be the result of repeated measurements using the same T cell for example."

The authors have responded by providing figure R2 organising the lifetime data based on cells experiencing a narrow force range (figure R2A) or the full force cycle (figure R2B). However, this does not address the key concern that performing repeated measurements on the same cell may affect the lifetime, as previously observed to be the case for TCR/pMHC (<https://www.pnas.org/doi/10.1073/pnas.0704811104>). For example, the cells in Figure R2B, such as the yellow cell, display a variety of bond lifetimes that are short and long but it's not possible to determine whether all the shorter lifetimes took place at earlier test cycles and all the longer cycles took place at later test cycles (i.e. is there a systematic bias in the lifetime with the test cycle?). To show that measurements are independent and later cycles do not tend to give longer or shorter lifetimes, the bond lifetime as a function of test cycle can be presented (as in the paper we cite above). The authors can also provide the source data as part of their manuscript.

The way the data is presented, it is also unclear whether a catch bond can be observed in an individual cell over multiple test cycles or is this a phenomena that only appears when plotting all the data together? Based on the data in Figure R2B, it looks as though individual cells display only slip bonds (possibly with the exception of the red cell) but it is difficult to see all the data since there is overlap. Can the authors plot the mean lifetime over force for each cell individually from Figure R2B?

Based on the raw data presented in figure R2, we are also wondering if the mean lifetime may be heavily skewed by a few outliers with long lifetimes (potentially those from earlier or later test cycles, or the ~10% of events that are not single-molecule involving multiple PD-1/PD-L1/L2 interactions). Can the authors test whether a few long-lived events are skewing their mean lifetimes? Do the authors get a similar lifetime over force curve when plotting the median rather than the mean lifetime for example?

Comment 2:

Despite the authors' response that the role of force has been toned down, we still note the following phrases in the discussion which have not been changed, and make the assumption that force is required, rather than observed:

- o "TGT-based T cell co-stimulation experiments further demonstrate that mechanical loading on the PD-1/PD-L1 interaction is essential for PD-1 function".
- o "This force-enhanced engagement of PD-1 is crucial for its inhibitory functions".

Referee #3:

The authors addressed all my previous points. The paper can now be published as is.

We thank all the reviewers for their valuable comments and suggestions. Since Reviewer #1 and #3 have no further concerns, below we responded the concerns from Reviewer #2 point-by-point. The original comments are presented in black, and our replies are in blue.

Referee #2:

The authors have addressed only some of our remaining concerns.

Comment 1:

The authors have responded by providing figure R2 organising the lifetime data based on cells experiencing a narrow force range (figure R2A) or the full force cycle (figure R2B). However, this does not address the key concern that performing repeated measurements on the same cell may affect the lifetime, as previously observed to be the case for TCR/pMHC (<https://www.pnas.org/doi/10.1073/pnas.0704811104>). For example, the cells in Figure R2B, such as the yellow cell, display a variety of bond lifetimes that are short and long but it's not possible to determine whether all the shorter lifetimes took place at earlier test cycles and all the longer cycles took place at later test cycles (i.e. is there a systematic bias in the lifetime with the test cycle?). To show that measurements are independent and later cycles do not tend to give longer or shorter lifetimes, the bond lifetime as a function of test cycle can be presented (as in the paper we cite above). The authors can also provide the source data as part of their manuscript.

Response: We thank the reviewer for raising this important point regarding potential cycle-dependent effects arising from repeated measurements on the same cell. The study cited indeed reported a memory effect in TCR-pMHC adhesion frequency upon repeated testing. To our knowledge, however, that work did not report or analyze the impact of repeated measurements on bond lifetime, which is the primary readout in our study. Nevertheless, we fully agree that it is important to explicitly demonstrate that repeated force cycles do not introduce a systematic bias in the measured PD-1/ligand bond lifetimes.

To directly address this concern, we have now analyzed bond lifetime as a function of test cycle. Specifically, we extracted the individual lifetime measurement data from a representative cell (082317-3, yellow cell in Fig. R2B of our previous response) and replotted the individual bond lifetime and the running average versus test cycle index (Figure R1A). This analysis shows that both short- and long-lived bonds are distributed throughout the entire testing sequence, with no observable monotonic increase or decrease in lifetime as testing progresses. Importantly, the mean bond lifetime remains stable across cycles, indicating the absence of a systematic cycle-dependent bias (Figure R1B).

These results demonstrate that repeated measurements on the same cell do not measurably affect PD-1/ligand bond lifetime under our experimental conditions,

supporting the independence of individual lifetime measurements.

Figure R1. (A) The sequence of all events from yellow cell (082317-3),

(B) Plot of the running mean force (green) and lifetime (red) calculated over cycles.

The way the data is presented, it is also unclear whether a catch bond can be observed in an individual cell over multiple test cycles or is this a phenomena that only appears when plotting all the data together? Based on the data in Figure R2B, it looks as though individual cells display only slip bonds (possibly with the exception of the red cell) but it is difficult to see all the data since there is overlap. Can the authors plot the mean lifetime over force for each cell individually from Figure R2B?

Response:

We thank the reviewer for this insightful question regarding whether catch-bond behavior can be observed at the level of individual cells rather than only after pooling data across cells.

To directly address this concern, we have now replotted the mean bond lifetime as a function of force for individual cells. Specifically, we selected three representative cells that were tested over a sufficiently broad force range to assess force-dependent behavior. As shown in the new Figure R2, each of these individual cells exhibits a non-monotonic lifetime–force relationship characteristic of catch-bond behavior, with bond lifetimes increasing with force at low forces and decreasing at higher forces.

Regarding the cell 082317-3 (yellow color shown in the Figure R2B of our previous response), this cell was tested only within a narrow, low-force regime and therefore does not span the force range required to resolve catch-bond behavior. As a result, it could not be meaningfully included in the individual-cell lifetime–force analysis.

These analyses demonstrate that catch-bond behavior in PD-1/ligand interactions can be observed at the level of individual cells across multiple test cycles, and is not an artifact arising from pooling data across cells.

Figure R2. The mean lifetime-mean force curves of three individual cells.

Based on the raw data presented in figure R2, we are also wondering if the mean lifetime

may be heavily skewed by a few outliers with long lifetimes (potentially those from earlier or later test cycles, or the ~10% of events that are not single-molecule involving multiple PD-1/PD-L1/L2 interactions). Can the authors test whether a few long-lived events are skewing their mean lifetimes? Do the authors get a similar lifetime over force curve when plotting the median rather than the mean lifetime for example?

Response:

We appreciate the reviewer's concern, giving us the opportunity to dig the experimental results in more depth.

In our BFP measurements, we applied a strict upper cutoff of 2 seconds to exclude excessively long-lifetime events. All events exceeding this threshold were removed prior to analysis, thereby minimizing the contribution of rare, unusually long-lived interactions. As shown in the scatter plot summarizing all PD1-PDL1 interaction events (**Figure R3A**), the observed catch-bond behavior does not arise from a small number of extreme outliers.

To further test the robustness of our conclusions, and as suggested by the reviewer, we have replotted the data using the median bond lifetime rather than the mean. The resulting lifetime-force relationship clearly exhibits non-monotonic dependence on force that is characteristic of catch bond behavior (**Figure R3B**). This demonstrates that the observed catch bond is not driven by a few long-lived events and is robust to the choice of statistical metric.

Figure R3. (A) The scatter plot of PD1-PDL1 interaction,

(B) The median lifetime-mean force curve of PD1-PDL1 interaction.

Finally, to address the possibility multi-molecular interactions or cellular context might contribute to the observed behavior, we performed additional BFP measurements using purified PD-1 and PD-L1 proteins from 293F cells. The purified PD-1/PD-L1 system also exhibits clearly catch-bond behavior (Figure R4; added as Figure EV2Q in the revised manuscript). The results are in agree with a previous study by Cheng Zhu's group demonstrated that purified TCR proteins, when reconstituted onto red blood cells via biotin–streptavidin ligation, still exhibit catch-bond behavior with their ligands (DOI: 10.1002/eji.201445358).

These results indicate that catch-bond formation is an intrinsic property of the PD-1/ligand interaction and is not dependent on cellular complexity or rare multi-molecular events.

Figure R4. The mean lifetime curve of purified PD1-PDL1 interaction.

Comment 2:

Despite the authors' response that the role of force has been toned down, we still note the following phrases in the discussion which have not been changed, and make the assumption that force is required, rather than observed:

- o "TGT-based T cell co-stimulation experiments further demonstrate that mechanical loading on the PD-1/PD-L1 interaction is essential for PD-1 function".
- o "This force-enhanced engagement of PD-1 is crucial for its inhibitory functions".

Response:

In the revised manuscript, we have now explicitly revised this language to avoid causal or essential claims regarding force, to ensure that our conclusions are appropriately framed as observational and mechanistic insights within the context of force-dependent assays, without implying that mechanical force is strictly required for PD-1 function in all physiological contexts.

Prof. Jizhong Lou
Institute of Biophysics, Chinese Academy of Sciences
China

Dear Prof. Lou,

Thank you for the submission of your final revised manuscript to our editorial offices. I have now received the report from the referee that I asked to look into this, you will find below. As you will see, the referee now fully supports a publication of your study in EMBO reports.

I am thus very pleased to accept your manuscript for publication in the next available issue of EMBO reports. Thank you for your contribution to our journal.

You may qualify for financial assistance for your publication charges - either via a Springer Nature fully open access agreement or an EMBO initiative. Check your eligibility: <https://link.springer.com/journal/44319/how-to-publish-with-us>

Yours sincerely,

Referee #2:

We thank the authors for taking the time to reply to our comments.

We have no further comments.

>>> Please note that it is EMBO Reports policy for the transcript of the editorial process (containing referee reports and your response letter) to be published as an online supplement to each paper. If you do NOT want this, you will need to inform the Editorial Office via email immediately. More information is available here: <https://link.springer.com/partners/embo-press/editorial-policies#Peer%20review>